# Document-Level In-Context Few-Shot Relation Extraction via Pre-Trained Language Models

## Abstract

Document-level relation extraction aims at inferring structured human knowledge from textual documents. State-of-the-art methods for this task use pre-trained language models (LMs) via fine-tuning, yet fine-tuning is computationally expensive and cannot adapt to new relation types or new LMs. As a remedy, we leverage the generalization capabilities of pre-trained LMs and present a novel framework for *document-level in-context few-shot relation extraction*. Our framework has three strengths: it eliminates the need (1) for named entity recognition and (2) for human annotations of documents, and (3) it can be updated to new LMs without re-training. We evaluate our framework using DocRED, the largest publicly available dataset for document-level relation extraction, and demonstrate that our framework achieves state-of-the-art performance. We further show that our framework actually performs much better than the original labels from the development set of DocRED. Finally, we conduct an extensive benchmark demonstrating the effectiveness of our framework, achieving state-of-the-art results across six relation extraction datasets and outperforming more than 30 baseline methods. Unlike our framework, the baseline methods have large computational overhead (e.g., from fine-tuning). To the best of our knowledge, we are the first to reformulate the document-level relation extraction task as a tailored in-context few-shot learning paradigm.

## 1 Introduction

Relational facts are widely used to represent human knowledge (Grishman, 2019; Han et al., 2020; Weikum et al., 2021). With the explosion of the web, relational facts have become broadly available through large knowledge bases (KBs) (Auer et al., 2007; Bollacker et al., 2008; Suchanek et al., 2007; Vrandečić & Krötzsch, 2014) and thereby support many downstream tasks. Examples are commonsense reasoning (Lin et al., 2019; Liu et al., 2021a), question answering (Das et al., 2022; Luo et al., 2018; Wang et al., 2022), fact checking (Huynh & Papotti, 2019; Vedula & Parthasarathy, 2021), and product recommendations (Wang et al., 2018; Zhou et al., 2020a;b). However, relational facts are not readily available in structured form but are commonly embedded in unstructured texts. To this end, methods are needed for relation extraction from text.

State-of-the-art methods for relation extraction leverage pre-trained language models (LMs) and fine-tune them using human-annotated documents. These works can be loosely grouped into two streams. One stream requires named entities (e. g., by specifying the entities of interest with a special token) as input (Hu et al., 2023; Tan et al., 2022; Wang et al., 2019; Wang Xu & Zhao, 2022; Xiao et al., 2022; Xu et al., 2021a; 2023; Zhang et al., 2021; Zhou et al., 2021). Another stream avoids the use of named entities as input and, instead, learns the detection of named entities through tailored training (Cabot & Navigli, 2021; Eberts & Ulges, 2021; Giorgi et al., 2022; Lu et al., 2022b; Zhang et al., 2023b).

Yet, state-of-the-art methods based on LMs have three main drawbacks that limit their applicability for relation extraction in practice. (1) State-of-the-art methods for relation extraction typically require the *named entities* to be either given as input or to be inferred via a customized training objective. This can propagate the errors into the relation extraction pipeline and thereby degrade the downstream performance. (2) State-of-the-art methods for relation extraction need large amounts of *human-annotated* documents for training. However, human annotation is costly. (3) State-of-the-art methods

are based on LMs that are *fine-tuned*. As a result, whenever a new type of relation is added to the knowledge base or whenever a better LM is adopted, the entire training process must be repeated. This introduces a huge computational overhead.

There are some recent efforts that use the reasoning abilities of LMs via in-context learning for relation extraction (Li et al., 2023; Wadhwa et al., 2023; Wan et al., 2023). However, these are designed for *sentence-level* relation extraction, meaning for a *small* set of relation types. Due to high computational costs, their scalability to documents is limited (see Table 1). Here, we introduce a novel method to leverage in-context learning for *document-level* relation extraction.

**Our REPLM framework:** We introduce a novel framework called REPLM for *document-level* in-context few-shot **r**elation **e**xtraction via **p**re-trained **l**anguage **m**odels. Our framework leverages the generalization capabilities of pre-trained LMs by reformulating the relation extraction task as a tailored in-context few-shot learning paradigm. Specifically, for a given document, we retrieve sets of the most relevant in-context examples of a corresponding relation and aggregate the outputs in a probabilistic framework.

**Contributions:**[1] ① We present a novel framework called REPLM for in-context few-shot relation extraction via pre-trained LMs. To the best of our knowledge, we are the first to reformulate the *document-level* relation extraction task as a tailored in-context few-shot learning paradigm. ② Our REPLM framework has key advantages for practice: it eliminates the error propagation from named entity recognition, it circumvents the need for human annotations, and it is flexible in that it is directly applicable to new relations and new backbone LMs without re-training. ③ We show that our REPLM achieves state-of-the-art performance across a variety of datasets.

## 2 RELATED WORK

**In-context few-shot learning of LMs:** LMs have achieved superior performance in many downstream tasks (Beltagy et al., 2019; Brown et al., 2020; Devlin et al., 2019; Lewis et al., 2020; Liu et al., 2019; OpenAI, 2023; Radford et al., 2019; Raffel et al., 2020; Wang & Komatsuzaki, 2021; Wang et al., 2023b; Wei et al., 2022a;b; Zhang et al., 2023c). Due to the large computational cost of fine-tuning an LM, Brown et al. (2020) proposed in-context few-shot learning to teach an LM a new task at inference time. We provide an overview of applications in Appendix A. However, we are not aware of any earlier work that leveraged in-context few-shot learning for *document-level* relation extraction.

**Early research on relation extraction:** Early works extracted relations from text via pattern extraction methods (Carlson et al., 2010; Jiang et al., 2017; Nakashole et al., 2012; Pawar et al., 2017; Weikum et al., 2021) and via statistical methods (Jiang & Zhai, 2007; Lin et al., 2015; Nguyen et al., 2007; Sarawagi & Cohen, 2004; Wang, 2008; Wang et al., 2014; Yu & Lam, 2010; Zhang et al., 2006a;b). However, the above methods have only a limited modeling capacity, as compared to neural networks (Adel & Schütze, 2017; Han et al., 2020; Katiyar & Cardie, 2017; Miwa & Bansal, 2016; Zeng et al., 2014; Zheng et al., 2017; Zhou et al., 2016). As shown later, LM-based methods better capture the complex interactions between named entities to classify the relation. A detailed review is in the Appendix A.

**Relation extraction via LM:** State-of-the-art methods for relation extraction are based on fine-tuning pre-trained LMs. Specifically, these methods use pre-trained LMs such as BERT (Devlin et al., 2019), RoBERTa (Liu et al., 2019), and SciBERT (Beltagy et al., 2019) and fine-tuned them for relation extraction. For instance, Wang et al. (2019) fine-tuned BERT to classify the relation between each named entity pair in a sentence. There have been various follow-up works to improve performance by learning complex dependency between named entities (Hu et al., 2023; Paolini et al., 2021; Tan et al., 2022; Wang & Lu, 2020; Wang Xu & Zhao, 2022; Xiao et al., 2022; Xu et al., 2021a; 2023; Zhang et al., 2023a; 2021; Zhou et al., 2021). A detailed review is in Appendix A. However, these works require the named entities to be annotated and provided as input at both training and test time.

Some works relax the requirement of given named entities to facilitate processing the raw documents at test time. As a remedy, these works jointly learn to extract named entities and relations. Examples are SpERT (Eberts & Ulges, 2020), JEREX (Eberts & Ulges, 2021), Seq2Rel (Giorgi et al., 2022), UIE (Lu et al., 2022b), and TaG (Zhang et al., 2023b). Their drawback is that multi-step pipelines with named entities recognition propagate the errors to relation extraction (Cabot & Navigli, 2021). Motivated by this, Cabot & Navigli (2021) developed **REBEL**, an auto-regressive model based on

---

[1]Codes available at https://anonymous.4open.science/r/REPLM_framework and in supplementary material.

BART (Lewis et al., 2020), which is fine-tuned to output relations as sequences of texts. To the best of our knowledge, **REBEL** is the only fine-tuned LM-based method without the need for a named entity recognition pipeline and thus represents one of our main baselines.

Still, the above methods have salient *limitations:* (1) they (with the exception of REBEL) require named entities to be given or infer them, which is a source of noise; (2) they require large amounts of human annotations; and (3) they require re-training to handle new relations.

**In-context learning for *sentence-level* relation extraction:**. There are three recent works that leverage in-context learning for *sentencel-level* relation extraction (see Table 1). GPT-RE (Wan et al., 2023) requires named entities to be provided for each sentence and generates $k$ different chain-of-thought (CoT) reasonings (Wei et al., 2022b) from GPT-3 (Brown et al., 2020) as in-context examples for each named entity pairs to classify their relation. CodeIE (Li et al., 2023) leverages Codex (deprecated) model from OpenAI for structured output. It requires that $k$ code generation examples are provided in-context for each relation type. Similar to GPT-RE, Wadhwa et al. (2023) uses CoT reasonings from GPT-3 for the entire training corpus and fine-tunes Flan-T5 (Chung et al., 2022) based on the generated CoT outputs. At the inference time, all of these works require $\mathcal{O}(k \cdot R)$ examples to be fit in-context for $k$-shot demonstrations of $R$ relation types. As a result, these works are *only* applicable to a *small* set of *sentences* and relation types for two reasons: (1) the *high* computational cost resulting from commercial architectures and (2) the requirement of a *large* number of in-context examples at the inference time. Therefore, these methods work only for the *sentence-level* relation extraction (i. e., they are not scalable to *document-level*).

**Research gap:** To the best of our knowledge, no work has adapted the in-context few-shot learning paradigm for *document-level* relation extraction. This presents our novelty and offers direct benefits in practice (i.e., no need for named entity input, no need for human annotations, and flexible adaptation to new relations without re-training).

Table 1: Comparison of relevant relation extraction methods.

| Method | Scope | Pre-trained LM | Open source | No need for fine-tuning. | No need for named entities. |
|---|---|---|---|---|---|
| GPT-RE Wan et al. (2023) | Sentence | OpenAI's GPT | ✓ | ✗ | ✗ |
| CodeIE Li et al. (2023) | Sentence | OpenAI's Codex | ✗* | ✓ | ✓ |
| Wadhwa et al. (2023) | Sentence | OpenAI's GPT | ✗ | ✗ | ✓ |
| REBEL Cabot & Navigli (2021) | Document | BART-large | ✓ | ✗ | ✓ |
| **REPLM (*ours*)** | Document | GPT-J† | ✓ | ✓ | ✓ |

*Codex models are deprecated at the time of writing.
† Our work can easily be extended to other LMs as shown in Section 8.

## 3 PROBLEM DESCRIPTION

**Relation extraction:** The relation extraction from documents is defined as follows (Eberts & Ulges, 2021; Giorgi et al., 2022; Lu et al., 2022b; Tan et al., 2022; Wang et al., 2019; Wang Xu & Zhao, 2022; Xu et al., 2021a; 2023; Zhang et al., 2021; Zhou et al., 2021). Given is a set of documents $\mathcal{D} = \{d_i\}_{i=1}^{M}$, where $M$ is the number of documents . For each document $d_i$, the aim is to enumerate knowledge triplets $\{(r_{im}, s_{im}, o_{im})\}_{m=1}^{R_i}$, where $r_{im} \in \mathcal{R}$ is a *relation* and $s_{im}$ and $o_{im}$ are the *subject* and *object* of the relation $r_{im}$, and where $R_i$ is the number of relations in $d_i$. For instance, the document "`The Reality Dysfunction` is a `science fiction` novel by British writer `Peter F. Hamilton` ..." yields the knowledge triplets (author, `The Reality Dysfunction`, `Peter F. Hamilton`), (genre, `The Reality Dysfunction`, `science fiction`), etc.

**Difference to earlier works:** Earlier works (see Sec. 2) generally address the above task through a mandatory step for named entities detection. Specifically, the aforementioned works first need to detect the named entities of a document $d_i$, i. e., $\{e_{ij}\}_{j=1}^{N_i}$, where $N_i$ is the number of entities in $d_i$. Then they proceed by predicting the relation(s) between each named entity pair $(e_{ij}, e_{ij'})_{j,j' \in \{1,...,N_i\}, j \neq j'}$ among the $R$ relations, where $e_{ij}$ is the subject and $e_{ij'}$ is the object of predicted relation(s). As such, the number of predictions scales with the number of named entity pairs, i.e., it is in $O(N_i^2)$.

**In-context few-shot learning in REPLM:** Our REPLM framework addresses the above drawbacks and approaches relation extraction as a triplet generation task. In this setup, the LM learns how to generate subject(s) and object(s) of a given relation from its in-context few-shot examples. Therefore, our REPLM framework does *not* require annotations of named entities. Our setup also facilitates the flexibility of adding new relations, simply by leveraging the given context examples. Specifically, for a given document $d_i \in \mathcal{D}$ and relation $r \in \mathcal{R}$, we prompt a pre-trained LM to generate the knowledge triplets of a relation $\{(r_{im}, s_{im}, o_{im}) \mid r_{im} = r\}$ with no further fine-tuning.

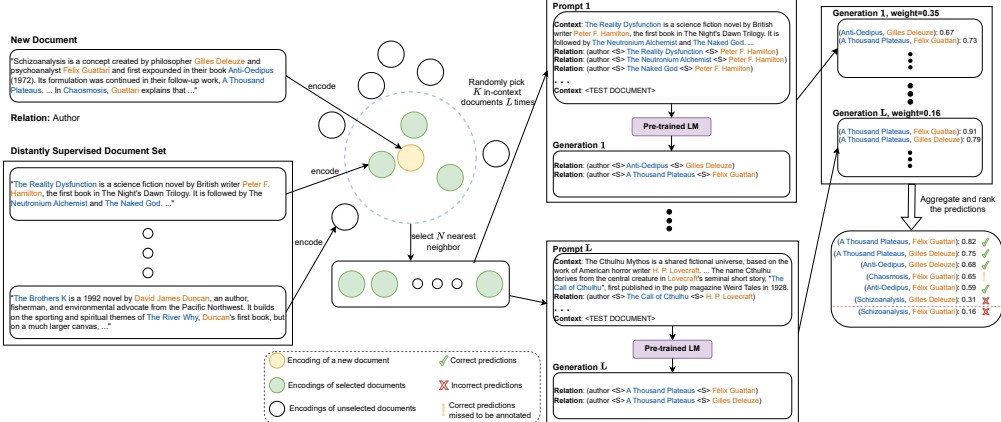

Figure 1: Overview of our REPLM. Our framework takes a new document and relation as input and then proceeds along three steps: (1) selects a candidate pool of $N$ in-context examples; (2) constructs $L$ sets of such in-context examples; and (3) calculates the joint probabilities of subject-object pairs to extract knowledge triplets. *Legend:* subjects and objects are colored in blue and orange, respectively.

In our REPLM framework, we have two sets of documents as input: (1) a distantly-supervised set $\mathcal{D}^{\text{dist}}$ for providing in-context few-shot examples and (2) a training set $\mathcal{D}^{\text{train}}$ for calibrating hyperparameters (which is optional). Details are in the next section.

## 4 PROPOSED REPLM FRAMEWORK

**Approach (see Fig. 1):** At a high level, our framework seeks to infer the correct knowledge triplets $(r, s, o)$ from a given document $d_i$ and for a given relation $r$. To do so, we estimate the joint probability of a subject-object pair $(s, o)$ conditional on $d_i$ and $r$, i.e., $p(s, o \,|\, d_i, r)$. After having estimated the probability, we simply rank the candidate subject-object pairs according to their probabilities and keep the top-ranked pairs as knowledge triplets. In our framework, we follow this approach but, as a main innovation, leverage a pre-trained LM to approximate $p(s, o \,|\, d_i, r)$.

**Learning via in-context few-shot examples:** Pre-trained LMs are not explicitly trained for our relation extraction task, although they generally have the ability to extract information from a given context when guided properly. In our framework, we intentionally avoid the use of fine-tuning a pre-trained LM due to high computational cost and the inability of handling new relations. Instead, we provide guidance for our task via in-context few-shot examples. These examples demonstrate how to extract the subject-object pairs of relation $r$ from the given context. As a result, we can approximate $p(s, o \,|\, d_i, r) \sim p(s, o \,|\, C, d_i, r)$, where $C$ represents the selected set of in-context examples.

However, selecting only a *single* set of in-context examples may lead to a poor approximation of the probability $p(s, o \,|\, d_i, r)$, because the selected in-context examples introduce bias in output generation (e. g., recency bias, label space of the in-context examples) as studied in prior literature (Hongjin et al., 2023; Liu et al., 2022a; Min et al., 2022b; Rubin et al., 2022; Wei et al., 2023). Instead, we mitigate the above bias by considering *multiple sets* of in-context examples. As a result, we calculate the joint probability of a subject-object pair as

$$p(s, o \,|\, d_i, r) = \sum_{l=1}^{L} p(C_l \,|\, d_i, r) \cdot p(s, o \,|\, C_l, d_i, r), \qquad (1)$$

where we aggregate the outputs from $L$ sets of in-context examples. Here, $p(C_l \,|\, d_i, r)$ is the weight of set $C_l$ of in-context examples, which measures how well $C_l$ is a candidate set compared to other sets of in-context examples.

**Steps:** Our REPLM framework proceeds along three steps: (1) it first selects a candidate pool for the in-context examples (Sec. 4.1); (2) it then constructs multiple sets of in-context examples and assigns their weights via a tailored approach (Sec. 4.2); and (3) it calculates the joint probabilities subject-object pairs to extract the knowledge triplets (Sec. 4.3). We describe the steps in the following.

### 4.1 SELECTING CANDIDATES FOR IN-CONTEXT EXAMPLES

We now create a candidate pool of in-context few-shot examples for a given document $d_i$. Crucially, we generate our candidate pool in a way that, on the one hand, it is created via distant supervision

and thus without human annotation, and, on the other hand, it is semantically related to the document $d_i$, thereby providing meaningful guidance.

**Distant supervision:** We create the in-context few-shot examples from the set $\mathcal{D}^{\text{dist}}$ generated by distant supervision. Specifically, $\mathcal{D}^{\text{dist}}$ is a dataset *without* any human annotation. In our implementation, we use the distantly-supervised split of DocRED (Yao et al., 2019), automatically created via an external knowledge base (KB). Reassuringly, we emphasize that this split comprises documents and knowledge triplets but it was created *without* any human annotation.

Distant supervision assumes that, if a document $\tilde{d} \in \mathcal{D}^{\text{dist}}$ contains both the subject and object of a knowledge triplet from a KB, it likely discusses their relationship. This premise allows for the automatic generation of annotated document sets. A key benefit is that distantly supervised documents offer rich insights into label space, textual distributions, and expected output formats, over which the in-context few-shot learning paradigm in our REPLM can generalize.[2]

**Semantic filtering:** (i) We first filter the documents in $\mathcal{D}^{\text{dist}}$ so that we only keep the documents that contain at least one knowledge triplet of a relation $r$. We denote the result by $\mathcal{D}_r^{\text{dist}}$, defined as

$$\mathcal{D}_r^{\text{dist}} = \{d_j \mid \exists\, r', s, o \text{ s.t. } (r', s, o) \in d_j \wedge r' = r \wedge d_j \in \mathcal{D}^{\text{dist}}\}. \tag{2}$$

(ii) We then retrieve $N$ documents from $\mathcal{D}_r^{\text{dist}}$ that are semantically most similar to $d_i$. For this, we leverage the technique from (Liu et al., 2022a) and encode the document $d_i$ and all the documents $\{d_j \mid d_j \in \mathcal{D}_r^{\text{dist}}\}$ into their embeddings via encoder $F_\theta$. (iii) We calculate the cosine similarity between the embeddings of $d_i$ and $d_j$, i.e., $\frac{F_\theta(d_i) \cdot F_\theta(d_j)}{||F_\theta(d_i)||_2 \cdot ||F_\theta(d_j)||_2}$. (iv) We keep the top-$N$ documents in $\mathcal{D}_r^{\text{dist}}$ in terms of cosine similarity to $d_i$ in embedding space. The selected $N$ documents form the context pool $\mathcal{D}_r^{\text{pool}}$, from which we construct multiple sets of in-context examples in the following.

### 4.2 Constructing Multiple Sets of In-Context Examples

For robustness, we create $L$ sets with in-context examples from our candidate pool $\mathcal{D}_r^{\text{pool}}$. We achieve this by random sampling of $K$ documents from $\mathcal{D}_r^{\text{pool}}$ across $L$ repetitions. As a result, we obtain $L$ sets of in-context examples, i.e., $C_1, \ldots, C_L$. Then, we perform weighting at the set level.

**Weighting at set level:** The output from each context set $C_l$ should contribute to the relation extraction task proportional to some weight $p(C_l \mid d_i, r)$. We calculate the weight as follows. First, we get a score for $C_l$ which is the average cosine similarity between the documents in $C_l$ and $d_i$, i.e.,

$$\text{score}(C_l) = \frac{1}{K} \sum_{d_j \in C_l} \frac{F_\theta(d_i) \cdot F_\theta(d_j)}{||F_\theta(d_i)||_2 \cdot ||F_\theta(d_j)||_2}. \tag{3}$$

We then use the score of $C_l$ to calculate the weight via

$$p(C_l \mid d_i, r) = \frac{\exp(\text{score}(C_l)/\tau)}{\sum_{l'=1}^{L} \exp(\text{score}(C_{l'})/\tau)}, \tag{4}$$

where $\tau > 0$ is for temperature scaling. Hence, $p(C_l \mid d_i, r)$ represents how much the final output of REPLM should attend to the output generated from the context set $C_l$.

### 4.3 Computing Knowledge Triplet Probabilities

We now calculate the probabilities for subject-object pairs and then extract the knowledge triplets.

**Prompting:** We prompt our pre-trained LM with both (i) the in-context few-shot examples derived from $C_l$ and (ii) the document $d_i$ at the end of the prompt. For this, we first prepare the in-context demonstrations for each context set $C_l$. That is, we concatenate the documents $d_j$ in $C_l$, where each document $d_j$ is appended with its corresponding knowledge triplets $\{(r_{jm}, s_{jm}, o_{jm}) \mid r_{jm} = r\}$. Each knowledge triplet is added in a new line (see Fig. 1). For the textual prompt, we separate the relation, subject, and object of $(r, s, o)$ with a special separator symbol ``. This facilitates easier parsing of the subjects and objects generated.

**Calculation of joint probability:** We first obtain the log probabilities of both subject and object tokens under our pre-trained LM. We normalize the log probabilities by the length (i. e., number of tokens) of the subject and object. Formally, we compute (here: we directly write the exponent of the

---

[2]We further compare distant supervision with human-annotated data. They have the same performance, confirming the effectiveness of this approach for relation extraction (see Appendix D).

average log. probabilities for the ease of reading):

$$p(s\,|\,C_l, d_i, r) = \sqrt[\mathrm{len}(s)]{\prod_{k=1}^{\mathrm{len}(s)} p(s_k\,|\,s_{<k}, C_l, d_i, r)}, \quad p(o\,|\,s, C_l, d_i, r) = \sqrt[\mathrm{len}(o)]{\prod_{k=1}^{\mathrm{len}(o)} p(o_k\,|\,o_{<k}, s, C_l, d_i, r)}, \quad (5)$$

where $\mathrm{len}(s)$ and $\mathrm{len}(o)$ are the number of tokens of the subject and object, respectively. Afterward, we compute the joint probability $p(s, o\,|\,C_l, d_i, r) = p(s\,|\,C_l, d_i, r) \cdot p(o\,|\,s, C_l, d_i, r)$.

**Ranking:** As the final step, we calculate $p(s, o\,|\,d_i, r)$ by aggregating over the context sets $C_l$, $l = 1, \ldots, L$, as in Eq. (1) and repeat this for all generated subject-object pairs. We keep all generated knowledge triplets whose probability exceeds a certain threshold $\theta$, i.e., $\{(r, s, o)\,|\,p(s, o\,|\,d_i, r) > \theta\}$. Of note, if a subject-object pair is not generated from a context set $C_l$, then $p(s, o\,|\,C_l, d_i, r) = 0$.

Note that the latter step is different from state-of-the-art methods as these methods must enumerate over all possible subject-object pairs. Further, as can be seen here, our framework does not require named entities as input, which is another salient difference to many of the existing works.

## 5 EXPERIMENTAL SETUP

We perform an extensive evaluation of our framework using the DocRED (Yao et al., 2019), the largest *document-level* relation extraction dataset publicly available. DocRED includes 96 relation types and comprises three sets: (1) a distantly-supervised set with 101,873 documents, (2) a human-annotated train set with 3053 documents, and (3) a human-annotated dev set with 998 documents. We provide further details about the DocRED dataset in Appendix B.1. Importantly, in our experimental setup, we use a distantly-supervised set for in-context few-shot learning ($\mathcal{D}^{\mathrm{dist}}$) and evaluate the performance on the development set. Thereby, we ensure that our framework is solely trained without human annotation. To better understand the performance of our REPLM, we thus later also perform additional experiments (see Sec. 7) using *sentence-level* relation extraction datasets.

**Baselines.** We evaluate our framework against state-of-the-art methods for relation extraction that scale to *document-level* and, for comparability, do *not* require named entity recognition pipelines (see Table 1). These are: (1) **REBEL** (Cabot & Navigli, 2021), applying triplet linearization to extract relations from the document. (2) **REBEL-sent**, extracting relations in a sentence-by-sentence manner. We include this variant because REBEL is originally trained at sentence level and, as shown later, is the best-performing baseline for sentence-level relation extraction. Note that REBEL is the *only* baseline from the literature not requiring a named entity recognition pipeline for *document-level* relation extraction.

In our experiments, we use REBEL-large from Hugging Face[3], which is pre-trained by a tailored REBEL dataset[4]. We note that REBEL-large is further fine-tuned on the human-annotated training set of DocRED, which may give it an (unfair) advantage. Hyperparameter selection and early stopping are based on the development set, which is again, an advantage not needed by our framework.

**REPLM variants:** We compare two variants of our framework: (1) **REPLM** is the original variant as described above. Therein, we use fixed parameters. A sensitivity analysis in Appendix J shows that the performance robustness to different parameter choices. (2) **REPLM (params adj)** is a variant for which the hyperparameters (e. g., temperature, threshold) are selected based on the training set.

We assess the contribution of different components in our framework. For this, we run an extensive series of experiments using the following variants: (1) **REPLM (random fixed)** randomly selects a single set of $K$ documents for each relation[5]. However, the set is fixed across all evaluations. (2) **REPLM (random all)** randomly selects a set of $K$ documents for each relation *and* for each evaluation. (3) **REPLM (best context$\ominus$)** selects the top-$K$ documents for each relation and each document according to the cosine similarity. In-context examples are ordered from most similar to the least similar. (4) **REPLM (best context$\oplus$)** similarly selects top $K$ documents for each relation and each document. This time, in-context examples are ordered in reverse order, from least similar to most similar. We include these two alternatives to evaluate the effect (if any) of recency bias (Hongjin et al., 2023; Lu et al., 2022a).

---

[3]https://huggingface.co/Babelscape/rebel-large

[4]https://huggingface.co/datasets/Babelscape/rebel-dataset

[5]We also explored finding the "best" documents (Appendix K) of a relation. It requires evaluation against human-annotation and still performs worse than REPLM. Hence, we exclude its results.

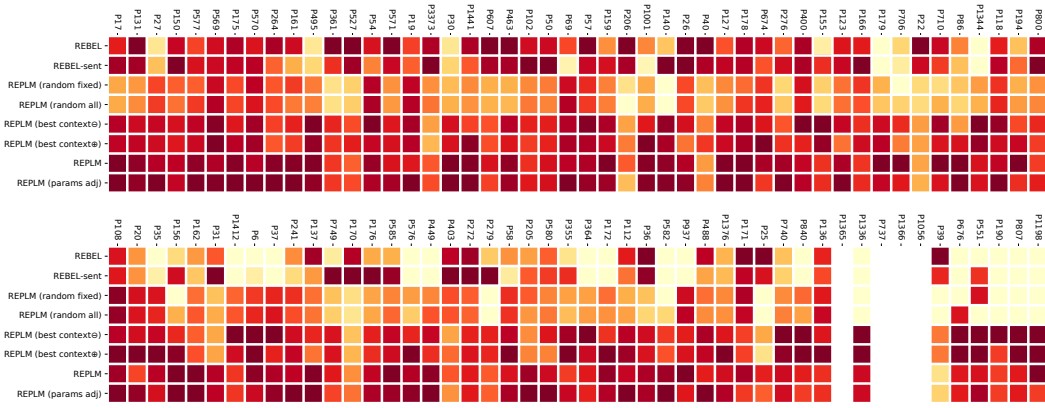

Figure 2: F1 scores per relation type (darker = better). Missing color means that no correct predictions were made for this relation. F1 scores are normalized by the maximum value for each relation. Relations are in decreasing order of their number of knowledge triplets.

**Evaluation.** We calculate the F1 score for each relation, counting an extraction as correct only if the subject and object exactly align with the ground-truth. Thus, extracted relations missing in the development set are false positives, while those in the set but not generated are false negatives.

**Implementation.** We mainly use GPT-JT[6] ($\sim$6B parameters) as our pre-trained LM for in-context few-shot learning. Our additional experiments (Sec. 7 and Sec. 8) show that other LMs can be seamlessly incorporated into our REPLM, such as OpenAI's GPT models or Meta's Llama models. Appendix E provides all details of our framework.

## 6 RESULTS

### 6.1 OVERALL PERFORMANCE

First, we evaluate how accurately our REPLM framework extracts the relations from the given documents by comparing them against human annotations (Fig. 2)[7]. Overall, our REPLM and REPLM (params adj) achieve state-of-the-art performance on most relation types. This pattern is especially pronounced for relations with a large number of knowledge triplets (e.g., P17: `country`, P131: `located in`, P27: `country of citizenship`).

Table 2 reports the overall performance, i.e., the micro F1 score over all relation types. Our REPLM achieves an F1 score of 33.93, and our REPLM (params adj) an F1 score of 35.09. The slight advantage of the latter is expected and can be attributed to the additional hyperparameter tuning. For comparison, the REBEL-sent baseline registers only an F1 score of 27.52. In sum, our framework performs the best and results in an improvement of +27 %. Note that REBEL was even fine-tuned on some samples of the dev set, which again demonstrates the clear superiority of our framework. We observe that REPLM outputs, on average, 20.21 knowledge triplets per document while REBEL outputs only 4.93; we discuss the implications later.

Table 2: Document-level relation extraction results. Shown: Micro F1.

| Method | F1 score |
|---|---|
| REBEL (Cabot et al., 2021) | 26.17 |
| REBEL-sent (Cabot et al., 2021) | 27.52 |
| REPLM (random fixed) | 21.04 $\pm$ 0.17 |
| REPLM (random all) | 21.14 $\pm$ 0.09 |
| REPLM (best context$\ominus$) | 31.31 |
| REPLM (best context$\oplus$) | 31.04 |
| REPLM (*ours*) | 33.93 |
| REPLM (params adj) (*ours*) | **35.09** |

Higher is better. Best value in bold.

We further compare different variants of our REPLM to understand the source of performance gains (see Fig. 2 and Table 2). (1) Retrieving the best in-context examples improves the performance compared to random examples by more than 48 % (REPLM (best context$\ominus$) and REPLM (best context$\oplus$) vs. REPLM (random fixed) and REPLM (random all)). (2) We do not observe that a recency bias plays a decisive role in our results, as both REPLM (best context$\ominus$) and REPLM (best context$\oplus$) reach a similar performance. (3) Our complete framework brings a significant improvement over REPLM (best context$\ominus$) and REPLM (best context$\oplus$) (+18 %) by aggregating multiple sets of most relevant in-context examples, thus establishing the importance of using multiple sets.

---

[6]https://huggingface.co/togethercomputer/GPT-JT-6B-v1

[7]F1 scores on each relation are given in Appendix H.

**Insights.** We conjecture that our REPLM extracts more relations than REBEL, as it further identifies missing annotations in DocRED. In Appendix F, we empirically validate that, for each relation type, some dev documents have no annotation but are semantically similar to those containing at least one knowledge triplet. This suggests these dev documents include the relation but lack the annotation. To confirm, we manually validate cases where our REPLM fails. We find many relations extracted by our method are correct but considered false positives due to missing annotations. For example, REPLM generates the relation (author, Chaosmosis, Félix Guattari) but it is not annotated and thus marked as incorrect (see right part of Fig. 1). Additional examples are in Appendix G.

## 6.2 Comparison Against External Knowledge

The above evaluations were constrained by relying solely on the human annotations on DocRED, potentially penalizing accurate methods due to missing annotations. We now repeat our evaluations using an alternative gold standard for a more comprehensive benchmark.

**Ground-truth via external knowledge:** To locate missing annotations in DocRED, we aggregate all relations extracted from all methods on all documents. We then check the correctness of the extracted relations via an external KB. Specifically,

Table 3: Document-level relation extraction results evaluated via external KB. Shown: Micro F1 scores.

| Method | F1-Score |
|---|---|
| REBEL (Cabot et al., 2021) | 20.30 |
| REBEL-sent (Cabot et al., 2021) | 20.00 |
| REPLM (*ours*) | 32.33 |
| REPLM (params adj) (*ours*) | **36.51** |

Higher is better. Best value in bold.

we leverage the pipeline from HELM (Liang et al., 2022) and check if generated knowledge triplets exist in Wikidata (Vrandečić & Krötzsch, 2014). We add all matched triplets to the existing list of ground-truth triplets from DocRED and repeat the evaluation. As a result, total number of relations in development set increased from 12,212 to 18,592.[8]

**Results:** For DocRED with external ground-truth, our framework outperforms REBEL by a considerable margin across most relation types (Fig. 3)[9] and in the overall performance (Table 3). For example, our REPLM improves F1 score over REBEL by more than 59 % (32.33 vs. 20.30). The improvement for REPLM (params adj) is even larger and amounts to 80 % (36.51 vs 20.30).

## 7 Extensive Benchmarking across Additional Datasets

After showing the effectiveness of our complete REPLM in the largest available document-level relation extraction dataset, we now turn to both smaller document-level datasets and sentence-level datasets, and then now conduct one of the most extensive benchmarking studies in relation extraction. Specifically, we implement our framework with 5 different LLM backbones, and compare them across 6 relation extraction datasets against more than 30 baseline methods. Yet, unlike our framework, the baselines have large computational overhead (e.g., additionally requiring both named-entity recognition and fine-tuning).

**REPLM variants:** The five different LLM backbones of our framework are: GPT-JT, Llama-3.1-8B, Llama-3.1-70B, GPT-3.5-Turbo, and GPT-4o.[10] **Datasets:** On top of DocRED, we consider two additional document-level relation extraction datasets: CDR (Li et al., 2016b) and GDA (Wu et al., 2019).[11] We consider three sentence-level datasets: CONLL04 (Roth & Yih, 2004), NYT (Riedel et al., 2010), and ADE (Gurulingappa et al., 2012). **Baseline methods:** All +30 baseline methods are only listed in Table 4 due to space.

**Performance:** Table 4 presents the micro-F1 scores for all methods across datasets. **(1)** Baseline methods show a clear divide: some can only be used for documents, while others only be used for sentences. This is due to that (a) sentence-level methods classify all named-entity pairs, which does *note* scale to documents, and that (b) document-level methods model inter- and intra-sentence relations explicitly, which does *not* apply to sentence-level datasets. In contrast, our framework handles all datasets and scales easily to larger models. **(2)** Adopting newer, stronger language models significantly boosts performance. For instance, on DocRED, the F1 score increased by +24.57 from GPT-JT to GPT-3.5-Turbo, and by +8.69 from GPT-3.5-Turbo to GPT-4o. This trend

---

[8]We note that the increase in the number of relations does not necessarily imply an improvement in the F1 score for our REPLM. The extracted relations are still filtered by the probability threshold $\theta$, which, in turn, reduces the recall (and possibly the F1 score).

[9]The details of evaluation via external KB are in Appendix I.

[10]For Llama, see Dubey et al. (2024). For GPT, see https://platform.openai.com/docs/models

[11]Details about CDR and GDA are given in Appendix B.2.

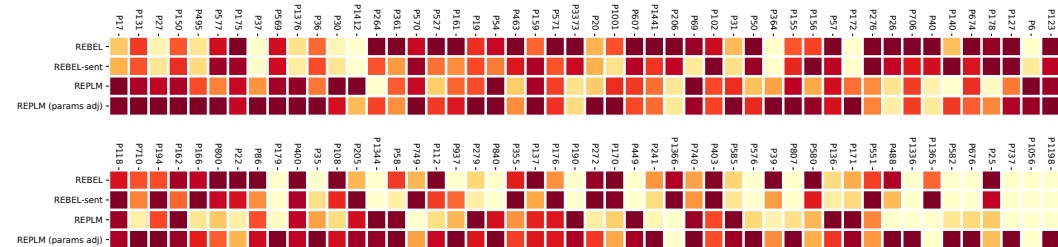

Figure 3: F1 scores per relation type (darker = better). Here, the comparison is made against annotations that additionally make use of external knowledge and should thus more closely reflect the ground-truth. Relation types are arranged in decreasing order of their number of knowledge triplets. For visibility, F1 scores are normalized by the maximum value for each relation.

is consistent across datasets and between Llama-3.1-8B and Llama-3.1-70B. **(3)** Our REPLM with GPT-4o achieves the best performance on DocRED, CoNLL04, and ADE, and near-best results on CDR and NYT.

We further investigated cases where the best variant of our framework, i.e., REPLM (GPT-4o), did not achieve top performance. The issues stem from missing or inconsistent entity annotations in the biomedical datasets CDR and GDA. Baseline methods, trained with these annotations, implicitly overfit to them and avoid the issue. For example, our framework correctly identifies the triplet "Gene-Disease Association, complement receptor 1, insulin-dependent diabetes mellitus" in GDA, but it is marked as a false positive since "complement receptor 1" is only annotated as "CR1", "C3bR", and "CD35". In NYT, noisy relations from the distantly-supervised dataset curation lead baseline methods to memorize triplets from training, thus inflating their performance incorrectly. Detailed analyses are provided in Appendix C.

Table 4: Evaluation of REPLM variants across datasets. Shown: Micro F1.

| | Method | Document-level | | | Sentence-level | | |
| --- | --- | --- | --- | --- | --- | --- | --- |
| | | DocRED | CDR | GDA | CONLL04 | NYT | ADE |
| Document-level methods | SciBERT (Beltagy et al., 2019) | – | 65.1 | 82.5 | – | – | – |
| | LSR (Nan et al., 2020) | – | 64.8 | 82.2 | – | – | – |
| | DHG (Zhang et al., 2020) | – | 65.9 | 83.1 | – | – | – |
| | GAIN (Zeng et al., 2020) | 61.22 | – | – | – | – | – |
| | JEREX (Eberts & Ulges, 2021) | 40.41 | – | – | – | – | – |
| | HeterGSAN (Xu et al., 2021c) | 60.18 | – | – | – | – | – |
| | DRN (Xu et al., 2021b) | 61.39 | – | – | – | – | – |
| | SIRE Zeng et al. (2021) | 61.60 | – | – | – | – | – |
| | SSAN (Xu et al., 2021a) | 65.69 | 68.7 | 83.7 | – | – | – |
| | ATLOP (Zhou et al., 2021) | 63.40 | 69.4 | 83.9 | – | – | – |
| | E2GRE (Huang et al., 2021) | 58.72 | – | – | – | – | – |
| | DocuNet (Zhang et al., 2021) | 64.55 | 76.3 | 85.3 | – | – | – |
| | EIDER (Xie et al., 2022) | 64.79 | – | – | – | – | – |
| | SAIS (Xiao et al., 2022) | 65.17 | **79.0** | **87.1** | – | – | – |
| | DREEAM (Ma et al., 2023) | 67.41 | – | – | – | – | – |
| | DocRE-CLiP (Jain et al., 2024) | **68.13** | – | – | – | – | – |
| Sentence-level methods | Neural Joint (Li et al., 2016a) | – | – | – | – | – | 63.40 |
| | SpERT Eberts & Ulges (2020) | – | – | – | 71.54 | – | 79.22 |
| | Table-sequence Wang & Lu (2020) | – | – | – | 73.58 | – | 80.07 |
| | BILSTM + Att (Geng et al., 2020) | – | – | – | 71.39 | – | – |
| | TANL Paolini et al. (2021) | – | – | – | 71.48 | 90.83 | 80.61 |
| | TriMF (Shen et al., 2021) | – | – | – | 72.35 | – | – |
| | CMAN (Zhao et al., 2021a) | – | – | – | 72.97 | – | 81.14 |
| | CL (Theodoropoulos et al., 2021) | – | – | – | – | – | 79.97 |
| | PFN (Yan et al., 2021) | – | – | – | – | – | 83.20 |
| | REBEL Cabot & Navigli (2021) | 27.52 | – | – | 75.41 | **92.02** | 82.23 |
| | TabERT (Ma et al., 2022) | – | – | – | 72.60 | – | – |
| | BL (Ji et al., 2022) | – | – | – | 72.62 | – | 81.33 |
| | STER (Zhao et al., 2022) | – | – | – | 74.44 | – | 81.24 |
| | FedJ (Wang et al., 2023a) | – | – | – | 72.35 | – | 82.37 |
| | PREFER (Liu et al., 2023) | – | – | – | **75.66** | – | **84.98** |
| | GPT-RE Wan et al. (2023) | – | – | – | 45.84 | – | – |
| | CodeIE Li et al. (2023)[†] | – | – | – | 53.12 | 32.22 | – |
| Our framework | REPLM (GPT-JT) | 35.09 | 55.98 | 66.92 | 72.94 | 81.03 | 82.54 |
| | REPLM (Llama-3.1-8B) | 55.50 | 67.36 | 71.07 | 69.43 | 84.55 | 87.11 |
| | REPLM (Llama-3.1-70B) | 62.31 | 73.29 | 74.10 | 72.00 | 89.33 | 91.45 |
| | REPLM (GPT-3.5) | 59.66 | 65.19 | 72.54 | 80.19 | 86.16 | 84.29 |
| | REPLM (GPT-4o) | **68.35** | **73.62** | **74.11** | **85.22** | **90.12** | **92.17** |

Best values are in bold. [†] CodeIE is the only baseline that does not require any model training.

## 8 ABLATION STUDY

**Are the performance gains of REPLM robust across different datasets and different LM backbones?** We present an ablation study demonstrating the effectiveness of our complete framework on six relation extraction datasets, evaluated across different LM backbones and variants of our own framework (i.e., random context vs. best context vs. complete framework). Our complete framework consistently outperforms retrieving only the best context, which, in turn, performs better than a random context. This pattern holds

Table 5: Ablation of REPLM variants. Shown: micro F1.

| Backbone | Variant | Document-level | | | Sentence-level | | |
|---|---|---|---|---|---|---|---|
| | | DocRED | CDR | GDA | CONLL04 | NYT | ADE |
| GPT-JT | random context | 21.14 | 39.21 | 57.48 | 62.48 | 58.92 | 76.37 |
| | best context | 31.31 | 47.37 | 63.02 | 68.16 | 77.73 | 79.82 |
| | complete framework | **35.09** | **55.98** | **66.92** | **72.94** | **81.03** | **82.54** |
| GPT-3.5 | random context | 46.83 | 58.57 | 66.41 | 72.12 | 68.09 | 78.36 |
| | best context | 54.31 | 62.16 | 68.40 | 74.09 | 83.81 | 81.63 |
| | complete framework | **59.66** | **65.19** | **72.54** | **80.19** | **86.16** | **84.29** |
| GPT-4o | random context | 52.29 | 67.41 | 69.30 | 77.19 | 73.09 | 85.55 |
| | best context | 61.78 | 71.66 | 71.59 | 79.90 | 86.81 | 90.18 |
| | complete framework | **67.47** | **73.62** | **74.11** | **85.22** | **90.12** | **92.17** |
| Llama-3.1-8B | random context | 30.01 | 47.87 | 66.25 | 40.15 | 47.67 | 72.34 |
| | best context | 40.85 | 50.38 | 68.46 | 53.03 | 62.51 | 81.18 |
| | complete framework | **55.50** | **67.36** | **71.07** | **69.43** | **84.55** | **87.11** |
| Llama-3.1-70B | random context | 52.02 | 63.63 | 68.99 | 34.54 | 60.91 | 82.37 |
| | best context | 57.28 | 67.44 | 70.64 | 53.63 | 83.72 | 82.58 |
| | complete framework | **62.31** | **73.29** | **74.10** | **72.00** | **89.33** | **91.45** |

Best values in bold. The std. dev. of random contexts are all below than 0.1 and thus omitted for brevity.

across both document-level and sentence-level datasets and all five backbone models. These findings demonstrate an important implication: *whenever more powerful LMs become available, one can integrate them into our REPLM in a seamless manner and thereby achieve important performance gains for relation extraction tasks.*

**What is the effect of the number of in-context examples?** We repeat the same experiment on CONLL04 when varying the number of in-context examples ($K$). Fig. 4a shows (i) the F1 score for each relation and (ii) the overall score when varying $K$ from 3 to 11. We observe that, in general, more in-context examples yield better F1 scores. Informed by this observation, we used the highest number of in-context examples that fit into the context window for our main experiments, which is $K = 5$ for document-level relation extraction. Detailed results are given in Appendix L.

**Is REPLM actually learning to extract relations? Or does it only retrieve facts from memory?** We design a novel experiment to identify whether our REPLM is learning to extract relations from the input text or it is simply retrieving the facts from its memory. To the best of our knowledge, we are the first to shed more light on the models' learning ability for the relation extraction task. For this experiment, we

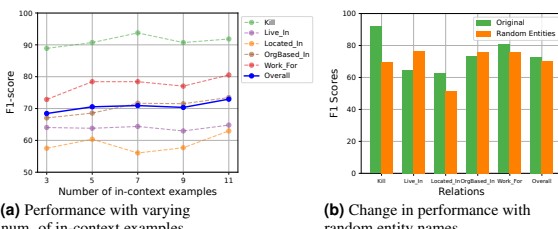

**(a)** Performance with varying num. of in-context examples.

**(b)** Change in performance with random entity names.

Figure 4: Ablation studies on CONLL04.

replaced all the entities with random names in CONLL04 dataset (for both training and test set) that are not mentioned anywhere on the web. Fig. 4b compares the performance against the original dataset. The overall performance decreases only slightly when using the random entities (F1 score of 70.47 vs. 72.9), which is still on par with the state-of-the-art. Therefore, it confirms that our REPLM is an effective method for learning to extract the relations from the context. We provide the experiment details and elaborate on the reasons of the slight performance decrease in Appendix M.

## 9 DISCUSSION

**Benefits:** Our REPLM framework offers many benefits in practice: (1) REPLM eliminates the need for named entity recognition pipelines in our task and thus the error propagated with it; (2) REPLM does not require human annotations but leverages in-context few-shot learning; and (3) REPLM offers great flexibility as it allows to incorporate new relations and new backbone LMs without re-training. Our study further identifies earlier datasets, such as DocRED (Yao et al., 2019), lack comprehensiveness, and thus miss important – but correct – annotations. This may penalize correct methods during benchmarking, suggesting the need of more effective evaluation paradigms.

**Broader Impact:** Our REPLM can help bridge gaps in knowledge bases, particularly for marginalized groups, improving coverage for diverse populations. However, as LM performance can vary, careful and responsible use is necessary when addressing societal, ethical, or sensitive content.

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

# A RELATED WORK

**In-context few-shot learning of LMs:** In-context few-shot learning has been widely adopted in text classification tasks (Holtzman et al., 2021; Liu et al., 2022a; Lu et al., 2022a; Min et al., 2022a;b; Zhao et al., 2021b) and further extended to other tasks such as question answering (Holtzman et al., 2021; Liu et al., 2022a; Min et al., 2022b), fact retrieval (Zhao et al., 2021b), table-to-text generation (Liu et al., 2022a), and mapping utterances to meaning representations (Rubin et al., 2022). However, we are not aware of any earlier work that leveraged the in-context few-shot learning paradigm for *document-level* relation extraction.

**LMs as knowledge bases:** Research has focused on probing the knowledge in LMs. For example, Petroni et al. (2019) introduced the LAMA dataset, a dataset with cloze-style templates for different relations, which allows to probe factual knowledge in LMs. Many works have been introduced to achieve state-of-the-art results via prompt-tuning (Hao et al., 2022; Lester et al., 2021; Li & Liang, 2021; Liu et al., 2021b; 2022b; Newman et al., 2022; Perez et al., 2021; Poerner et al., 2020; Shin et al., 2020; Zhong et al., 2021). Yet, some works further find that, when evaluated as knowledge bases, LMs suffer from inconsistency (AlKhamissi et al., 2022; Elazar et al., 2021), learn shallow heuristics rather than facts (Elazar et al., 2022), have inferior performance in the long tail (Kandpal et al., 2022), and exhibit prompt bias (Cao et al., 2021).

However, we note that the above research stream is different from our work in two salient ways. (1) In the above research stream, LMs are prompted to retrieve factual knowledge from its *memory*, whereas we aim to extract the relational knowledge from the *context*. (2) In the above research stream, LM prompts are structured as "fill-in-the-blank" cloze statement. For example, the task is to output only the correct object, but where both the subject and relation are *given*. Instead of predicting only the object, our goal is to output the entire knowledge triplet. That is, subject, relation, and object must be *inferred* together.

**Relation extraction via pattern-based and statistical methods:** A detailed review of the different methods is provided in Pawar et al. (2017) and Weikum et al. (2021), while we only present a brief summary here. Early works extracted relations from text via pattern-based extraction methods. Specifically, these works introduced automated methods to extract textual patterns corresponding to each relation and each entity type (Carlson et al., 2010; Jiang et al., 2017; Nakashole et al., 2012). Their main limitation is that the automatically constructed patterns involve many mistakes, which, in turn, require human experts to examine and correct them (Han et al., 2020).

Another research stream focused on relation extraction via statistical methods. Examples are crafting custom features for relation classification (Jiang & Zhai, 2007; Nguyen et al., 2007), designing customized kernels for support vector machines(Nguyen et al., 2007; Wang, 2008; Zhang et al., 2006a;b), graphical modeling and inference of relations (Sarawagi & Cohen, 2004; Yu & Lam, 2010), and leveraging knowledge graph embeddings for relation prediction (Lin et al., 2015; Wang et al., 2014).

However, the above methods have only a limited capacity in capturing complex interactions between entities, as compared to state-of-the-art neural networks (Han et al., 2020). On top of that, both pattern-based and statistical methods require large datasets with human annotation for training.

**Relation extraction via neural networks:** Initial methods for relation extraction based on neural network approaches made use of convolutional neural network (CNN) (Zeng et al., 2014) and long short-term memory (LSTM) (Zhou et al., 2016) architectures. These works process the pre-computed word embeddings and then classify the relation for the given named entity pair. Follow-up works proposed joint learning of entity extraction and relation classification, again via CNN (Adel & Schütze, 2017; Zheng et al., 2017) and LSTM (Katiyar & Cardie, 2017; Miwa & Bansal, 2016) architectures. However, these models are not flexible enough to model the complex interactions between named entities to classify the relation, as compared to LMs.

**Relation extraction via LMs:** State-of-the-art methods for relation extraction are based on fine-tuning pre-trained LMs. Specifically, these methods use pre-trained LMs such as BERT (Devlin et al., 2019), RoBERTa (Liu et al., 2019), and SciBERT (Beltagy et al., 2019) and fine-tuned them for relation extraction. For instance, Wang et al. (2019) fine-tuned BERT to classify the relation between each named entity pair in a given sentence. There have been various follow-up works to improve performance by learning complex dependency between named entities. To achieve this,

Wang & Lu (2020) jointly trained LSTM and BERT to get two distinct representations of the entities; Zhang et al. (2021) further incorporate semantic segmentation module into the fine-tuning of BERT; Zhou et al. (2021) propose adaptive thresholding and localized context pooling; Xu et al. (2021a) explicitly model the dependencies between entity mentions; Paolini et al. (2021) augmented the original sentences with the entity and relation types; Tan et al. (2022) use axial attention module for learning the interdependency among named entity pairs; Wang Xu & Zhao (2022) propose sentence importance estimation; Xiao et al. (2022) include additional tasks such as coreference resolution, entity typing, and evidence retrieval; Xu et al. (2023) improves the model performance via synthetic data generation; Hu et al. (2023) incorporates rationale extraction from the sentence; and Zhang et al. (2023a) leverages self-distillation to facilitate relational reasoning. However, all of these works require the named entities to be annotated and provided as input at both training and test time.

# B    DETAILS ON DOCUMENT-LEVEL RELATION EXTRACTION DATASETS

## B.1    DOCRED

For our experiments, we mainly use DocRED (Yao et al., 2019), the largest publicly available dataset for document-level relation extraction. We provide the detailed statistics of each relation type in Tables 6 and 7 (note: the different columns compare the different subsets for distant supervision, human-annotated training, and human-annotated dev).

**Pre-processing.** The original documents in the DocRED dataset are provided only in a tokenized format, e. g., the document is represented as a list of token, where each punctuation mark and word is a different token. We follow the earlier works (Cabot & Navigli, 2021; Yao et al., 2019) and concatenate the tokens with a white space in between to construct the entire document. This approach may introduce typos in the documents; for instance, the original text "`Tarzan's Hidden Jungle is a 1955 black-and-white film ...`" is reconstructed as "`Tarzan 's Hidden Jungle is a 1955 black - and - white film ...`". We initially tried to fix these typos via spelling correction libraries, such as FastPunct[12], but later found that the typos are propagated to the labels, which may impede performance and eventually comparability of our results. Therefore, we decided to follow the same pre-processing as earlier works, as it allows us to operate on the same labels as in earlier work and thus ensures comparability of our results. Sec. G shows some examples of documents after the pre-processing step.

Table 6: DocRED statistics

| Relation ID | Relation Name | # Docs in Dist. Sup. | # Relations in Dist. Sup. | # Docs in Train | # Relations in Train | # Docs in Dev | # Relations in Dev |
|---|---|---|---|---|---|---|---|
| P6 | head of government | 4948 | 6859 | 133 | 210 | 38 | 47 |
| P17 | country | 68402 | 313961 | 1831 | 8921 | 585 | 2817 |
| P19 | place of birth | 21246 | 31232 | 453 | 511 | 135 | 146 |
| P20 | place of death | 15046 | 24937 | 170 | 203 | 50 | 52 |
| P22 | father | 5287 | 9065 | 164 | 273 | 41 | 57 |
| P25 | mother | 1828 | 2826 | 50 | 74 | 10 | 15 |
| P26 | spouse | 4327 | 9723 | 134 | 303 | 34 | 74 |
| P27 | country of citizenship | 45553 | 126360 | 1141 | 2689 | 384 | 808 |
| P30 | continent | 7247 | 18792 | 121 | 356 | 38 | 121 |
| P31 | instance of | 3790 | 5561 | 74 | 103 | 34 | 48 |
| P35 | head of state | 3127 | 4257 | 87 | 140 | 32 | 51 |
| P36 | capital | 27621 | 34047 | 66 | 85 | 24 | 27 |
| P37 | official language | 4040 | 6562 | 82 | 119 | 29 | 47 |
| P39 | position held | 982 | 1692 | 15 | 23 | 6 | 8 |
| P40 | child | 5794 | 11831 | 177 | 360 | 45 | 81 |
| P50 | author | 5265 | 8856 | 162 | 320 | 49 | 93 |
| P54 | member of sports team | 2693 | 12312 | 80 | 379 | 36 | 166 |
| P57 | director | 5891 | 9865 | 153 | 246 | 58 | 90 |
| P58 | screenwriter | 4680 | 7952 | 83 | 156 | 24 | 35 |
| P69 | educated at | 5201 | 8413 | 220 | 316 | 63 | 92 |
| P86 | composer | 2778 | 4249 | 44 | 79 | 21 | 57 |
| P102 | member of political party | 5464 | 11582 | 191 | 406 | 51 | 98 |
| P108 | employer | 4168 | 6775 | 126 | 196 | 30 | 54 |
| P112 | founded by | 5856 | 7700 | 74 | 100 | 20 | 27 |
| P118 | league | 2142 | 6024 | 63 | 185 | 29 | 56 |
| P123 | publisher | 2426 | 4444 | 81 | 172 | 29 | 69 |
| P127 | owned by | 4907 | 7554 | 91 | 208 | 36 | 76 |
| P131 | located in the administrative territorial entity | 44307 | 143006 | 1224 | 4193 | 389 | 1227 |
| P136 | genre | 982 | 1948 | 34 | 111 | 7 | 14 |
| P137 | operator | 1982 | 3011 | 52 | 95 | 18 | 41 |
| P140 | religion | 2515 | 5143 | 60 | 144 | 26 | 82 |
| P150 | contains administrative territorial entity | 34615 | 62646 | 1002 | 2004 | 310 | 603 |
| P155 | follows | 8360 | 12236 | 117 | 188 | 43 | 69 |
| P156 | followed by | 7958 | 11576 | 120 | 192 | 38 | 51 |
| P159 | headquarters location | 12653 | 17089 | 206 | 264 | 57 | 86 |
| P161 | cast member | 6575 | 21139 | 163 | 621 | 62 | 226 |
| P162 | producer | 4434 | 6739 | 77 | 119 | 32 | 50 |
| P166 | award received | 2852 | 6322 | 105 | 173 | 35 | 66 |
| P170 | creator | 3485 | 6036 | 96 | 231 | 25 | 40 |
| P171 | parent taxon | 860 | 2167 | 28 | 75 | 6 | 17 |
| P172 | ethnic group | 6022 | 7563 | 63 | 79 | 24 | 30 |
| P175 | performer | 10783 | 27945 | 344 | 1052 | 101 | 332 |
| P176 | manufacturer | 1260 | 2737 | 27 | 83 | 9 | 40 |
| P178 | developer | 2403 | 6368 | 73 | 238 | 30 | 75 |
| P179 | series | 2404 | 3800 | 72 | 144 | 27 | 63 |
| P190 | sister city | 3388 | 11471 | 2 | 4 | 1 | 2 |
| P194 | legislative body | 2863 | 2989 | 136 | 166 | 36 | 56 |
| P205 | basin country | 2249 | 3299 | 61 | 85 | 21 | 32 |

[12]https://pypi.org/project/fastpunct/

Table 7: DocRED statistics (continued)

| Relation ID | Relation Name | # Docs in Dist. Sup. | # Relations in Dist. Sup. | # Docs in Train | # Relations in Train | # Docs in Dev | # Relations in Dev |
|---|---|---|---|---|---|---|---|
| P206 | located in or next to body of water | 3859 | 6585 | 109 | 194 | 35 | 83 |
| P241 | military branch | 1589 | 2633 | 69 | 108 | 30 | 42 |
| P264 | record label | 4524 | 14804 | 154 | 583 | 49 | 237 |
| P272 | production company | 1417 | 2151 | 49 | 82 | 19 | 36 |
| P276 | location | 5281 | 6654 | 130 | 172 | 55 | 74 |
| P279 | subclass of | 1822 | 2736 | 39 | 77 | 19 | 36 |
| P355 | subsidiary | 1761 | 2436 | 51 | 92 | 18 | 30 |
| P361 | part of | 17335 | 28245 | 382 | 596 | 119 | 194 |
| P364 | original language of work | 1061 | 2274 | 32 | 66 | 11 | 30 |
| P400 | platform | 1565 | 5825 | 52 | 304 | 14 | 69 |
| P403 | mouth of the watercourse | 1700 | 2475 | 49 | 95 | 19 | 38 |
| P449 | original network | 2953 | 4237 | 97 | 152 | 20 | 39 |
| P463 | member of | 7364 | 15272 | 208 | 414 | 55 | 113 |
| P488 | chairperson | 1792 | 2216 | 49 | 63 | 15 | 21 |
| P495 | country of origin | 17160 | 36029 | 300 | 539 | 112 | 212 |
| P527 | has part | 13318 | 22596 | 317 | 632 | 94 | 177 |
| P551 | residence | 2629 | 3197 | 25 | 35 | 5 | 6 |
| P569 | date of birth | 26474 | 33998 | 893 | 1044 | 286 | 343 |
| P570 | date of death | 20905 | 28314 | 587 | 805 | 180 | 255 |
| P571 | inception | 19579 | 26699 | 393 | 475 | 127 | 154 |
| P576 | dissolved, abolished or demolished | 5064 | 7057 | 52 | 79 | 25 | 39 |
| P577 | publication date | 17636 | 37538 | 576 | 1142 | 193 | 406 |
| P580 | start time | 5374 | 6549 | 96 | 110 | 30 | 32 |
| P582 | end time | 4943 | 6144 | 47 | 51 | 18 | 23 |
| P585 | point in time | 2457 | 2920 | 80 | 96 | 29 | 39 |
| P607 | conflict | 4119 | 8056 | 114 | 275 | 46 | 114 |
| P674 | characters | 1594 | 3447 | 62 | 163 | 25 | 74 |
| P676 | lyrics by | 1677 | 2415 | 30 | 36 | 5 | 8 |
| P706 | located on terrain feature | 3157 | 5063 | 74 | 137 | 29 | 60 |
| P710 | participant | 2839 | 4985 | 95 | 191 | 22 | 57 |
| P737 | influenced by | 1166 | 2071 | 9 | 9 | 3 | 10 |
| P740 | location of formation | 3885 | 4531 | 53 | 62 | 12 | 15 |
| P749 | parent organization | 2425 | 3335 | 47 | 92 | 27 | 40 |
| P800 | notable work | 4053 | 5275 | 102 | 150 | 32 | 56 |
| P807 | separated from | 1438 | 2210 | 2 | 2 | 1 | 2 |
| P840 | narrative location | 2026 | 2573 | 38 | 48 | 11 | 15 |
| P937 | work location | 5063 | 7470 | 69 | 104 | 19 | 22 |
| P1001 | applies to jurisdiction | 7471 | 9945 | 204 | 298 | 55 | 83 |
| P1056 | product or material produced | 460 | 624 | 27 | 36 | 6 | 9 |
| P1198 | unemployment rate | 1330 | 1622 | 2 | 2 | 1 | 1 |
| P1336 | territory claimed by | 880 | 1600 | 18 | 33 | 6 | 10 |
| P1344 | participant of | 1707 | 3574 | 87 | 223 | 28 | 57 |
| P1365 | replaces | 1490 | 1811 | 13 | 18 | 9 | 10 |
| P1366 | replaced by | 2214 | 2771 | 25 | 36 | 10 | 10 |
| P1376 | capital of | 25241 | 29816 | 62 | 76 | 20 | 21 |
| P1412 | languages spoken, written or signed | 2781 | 6313 | 91 | 155 | 24 | 46 |
| P1441 | present in work | 2872 | 6763 | 88 | 299 | 34 | 116 |
| P3373 | sibling | 3335 | 11123 | 102 | 335 | 26 | 134 |

## B.2 CDR AND GDA

**CDR** (Li et al., 2016b) contains the abstracts from PubMed (https://pubmed.ncbi.nlm.nih.gov/) but it contains only one relation type, which is the chemical-induced disease. We use the original splits in our work. The dataset statistics can be found in Table 8.

Table 8: CDR statistics

| Relation Name | # Docs in Train | # Relations in Train | # Docs in Validation | # Relations in Validation | # Docs in Test | # Relations in Test |
|---|---|---|---|---|---|---|
| Chemical-Induced Disease | 500 | 1038 | 500 | 1012 | 500 | 1066 |

**GDA** (Wu et al., 2019) is another dataset that offers a collection of documents from the medical domain. The documents are again the abstracts of PubMed. The dataset contains only one relation type, which is gene-disease association. We use the original test split for the evaluation of our work. The statistics of GDA can be found in Table 9.

Table 9: GDA statistics

| Relation Name | # Docs in Train | # Relations in Train | # Docs in Validation | # Relations in Validation | # Docs in Test | # Relations in Test |
|---|---|---|---|---|---|---|
| Gene-Disease Association | 29192 | 44841 | _ | _ | 1000 | 1502 |

# C    DETAILS ON SENTENCE-LEVEL RELATION EXTRACTION

We select three sentence-level relation extraction datasets to show the effectiveness of our REPLM framework against the state-of-the-art supervised methods.

**CONLL04** (Roth & Yih, 2004) is consisting of sentences collected from the news articles. The authors manually annotated the entities and 5 relation types for each sentence. Following the earlier literature, we used the same splits as Eberts & Ulges (2020). The detailed statistics for each relation type are given in Table 10.

Table 10: CONLL04 statistics

| Relation Name | # Sentences in Train | # Relations in Train | # Sentences in Validation | # Relations in Validation | # Sentences in Test | # Relations in Test |
| --- | --- | --- | --- | --- | --- | --- |
| Kill | 160 | 179 | 39 | 42 | 46 | 47 |
| Live_In | 270 | 326 | 68 | 84 | 82 | 98 |
| Located_In | 187 | 245 | 52 | 65 | 58 | 90 |
| OrgBased_In | 213 | 260 | 47 | 71 | 70 | 96 |
| Work_For | 208 | 250 | 57 | 69 | 65 | 76 |
| Overall | 922 | 1260 | 231 | 331 | 288 | 407 |

**NYT** (Riedel et al., 2010) is composed of sentences from New York Times, containing 24 relation types. The detailed statistics are given in Table 11.

It is important to note that the relations in this dataset are annotated via "distant supervision", using the knowledge triplets from FreeBase (Bollacker et al., 2008). As a result, the evaluation on the test set becomes noisy. For instance, the sentence in the test set "`Mr. Abbas, speaking before a meeting in Paris with the French president, Jacques Chirac, said he was sorry for the shootings on Sunday.`" is annotated with the following knowledge triplet "`(place of birth, Jacques Chirac, Paris)`", although the birthplace of Jacques Chirac cannot be inferred from the sentence.

We further found that the overlap of relations between train and test set is high. For the relation type place of birth, 166 out of 260 relations (i. e., the exact (relation, subject, object) triplet) in test set appear in the training set. Therefore, although the evaluation on the test is noisy, the baseline methods leverage the supervised training and they can memorize the relations from the train set at the test time. We hypothesize this as the main reason of the inferior performance of our REPLM framework on this dataset specifically, while achieving the state-of-the-art performance at all other evaluations.

Table 11: NYT statistics

| Relation Name | # Sentences in Train | # Relations in Train | # Sentences in Validation | # Relations in Validation | # Sentences in Test | # Relations in Test |
| --- | --- | --- | --- | --- | --- | --- |
| advisors | 37 | 37 | 5 | 5 | 3 | 3 |
| capital | 6042 | 6121 | 557 | 567 | 649 | 659 |
| child | 407 | 437 | 45 | 46 | 32 | 40 |
| contains_administrative_territorial_entity | 4889 | 5111 | 462 | 497 | 496 | 527 |
| country | 4889 | 5111 | 462 | 497 | 496 | 527 |
| country_of_citizenship | 6136 | 6606 | 545 | 596 | 518 | 548 |
| country_of_origin | 19 | 19 | 1 | 1 | 1 | 1 |
| denonym | 29 | 32 | 3 | 3 | 1 | 1 |
| employer | 4546 | 4734 | 428 | 448 | 401 | 417 |
| ethnicity | 19 | 19 | 1 | 1 | 1 | 1 |
| founded_by | 649 | 682 | 53 | 58 | 58 | 61 |
| headquarters_location | 180 | 186 | 21 | 22 | 17 | 17 |
| industry | 1 | 1 | – | – | – | – |
| location | 37626 | 42961 | 3302 | 3818 | 3296 | 3835 |
| location_of_formation | 344 | 346 | 35 | 35 | 35 | 35 |
| major_shareholder | 229 | 238 | 21 | 21 | 31 | 32 |
| member_of_sports_team | 180 | 186 | 21 | 22 | 17 | 17 |
| neighborhood_of | 4329 | 4682 | 403 | 444 | 338 | 374 |
| occupation | 2 | 2 | – | – | – | – |
| place_of_birth | 2649 | 2703 | 215 | 217 | 256 | 260 |
| place_of_death | 1652 | 1676 | 125 | 128 | 127 | 131 |
| religion | 54 | 56 | 7 | 7 | 5 | 5 |
| residence | 5883 | 6182 | 506 | 531 | 570 | 597 |
| shareholders | 229 | 238 | 21 | 21 | 31 | 32 |
| Overall | 56196 | 88366 | 5000 | 7985 | 5000 | 8120 |

**ADE** (Gurulingappa et al., 2012) contains the sentences from biomedical domain and it has only one relation type, which is adverse effect. The original dataset contains 10 folds of train and test

splits. Following the earlier work (Cabot & Navigli, 2021), we use the test set of the first fold for the evaluation. The statistics are given in Table 12.

Table 12: ADE statistics

| Relation Name | # Sentences in Train | # Relations in Train | # Sentences in Validation | # Relations in Validation | # Sentences in Test | # Relations in Test |
|---|---|---|---|---|---|---|
| Adverse-Effect | 3845 | 5980 | _ | _ | 427 | 653 |

# D IN-CONTEXT FEW-SHOT LEARNING BASED ON DISTANT SUPERVISION VS. HUMAN ANNOTATION

We perform an ablation study to compare the effect of using distantly-supervised vs. human-annotated documents as in-context few-shot examples. We perform such comparison using the four variants of our framework variants, i.e., REPLM (random fixed), REPLM (random all), REPLM (best context⊖), and REPLM (best context⊕). For methods with random in-context examples, the performance may be subject to variability across which seed is picked (whereas the performance is deterministic for the other methods), and, hence, we report the standard deviation for this subset of the methods by averaging the performance across 10 runs. Of note, to directly compare the impact of in-context examples, we deliberately considered our REPLM variants without aggregation over the multiples sets of in-context examples.

Tables 13 to 24 show the comparison between distant supervision vs. human annotation. For random in-context examples (i. e., REPLM (random fixed) and REPLM (random all)), distant supervision and human annotation performs at the same level. For retrieving the semantically most similar context examples (i. e., REPLM (best context⊖) and REPLM (best context⊕)), we observe cases where distant supervision actually improves the result (e. g., P6, P155, P179). However, the overall performance is largely similar. This confirms our choice of using distantly-supervised documents as in-context examples and eliminates the need for human annotation.

Table 13: Ablation study. Comparing the performance across the in-context examples from distant supervision vs. human-annotated train set. Shown are F1 scores on each relation. (Part 1 / 12)

| Method | Context Source | P6 | P17 | P19 | P20 | P22 | P25 | P26 | P27 |
|---|---|---|---|---|---|---|---|---|---|
| REPLM (random fixed) | Train | 20.29 $\pm$ 8.01 | 11.18 $\pm$ 2.44 | 75.68 $\pm$ 6.45 | 71.92 $\pm$ 4.61 | 13.17 $\pm$ 5.66 | 0.00 $\pm$ 0.00 | 24.09 $\pm$ 5.00 | 23.03 $\pm$ 3.25 |
| REPLM (random fixed) | Dist. Sup. | 24.28 $\pm$ 10.53 | 11.34 $\pm$ 4.48 | 66.67 $\pm$ 12.71 | 49.80 $\pm$ 16.07 | 9.76 $\pm$ 4.90 | 0.00 $\pm$ 0.00 | 26.73 $\pm$ 4.38 | 22.41 $\pm$ 2.53 |
| REPLM (random all) | Train | 19.76 $\pm$ 2.59 | 11.03 $\pm$ 0.88 | 77.17 $\pm$ 1.50 | 68.24 $\pm$ 3.54 | 11.94 $\pm$ 3.03 | 0.00 $\pm$ 0.00 | 24.03 $\pm$ 2.73 | 21.94 $\pm$ 1.50 |
| REPLM (random all) | Dist. Sup. | 25.90 $\pm$ 2.44 | 11.05 $\pm$ 0.46 | 68.49 $\pm$ 3.09 | 48.35 $\pm$ 4.82 | 9.51 $\pm$ 2.86 | 0.00 $\pm$ 0.00 | 26.74 $\pm$ 3.83 | 23.26 $\pm$ 1.39 |
| REPLM (best context⊖) | Train | 18.18 | 19.62 | 79.29 | 69.90 | 15.09 | 7.14 | 33.33 | 28.03 |
| REPLM (best context⊖) | Dist. Sup. | 35.96 | 24.60 | 71.38 | 50.39 | 18.18 | 15.38 | 33.33 | 29.14 |
| REPLM (best context⊕) | Train | 18.18 | 19.65 | 74.20 | 72.00 | 20.00 | 5.71 | 30.43 | 28.27 |
| REPLM (best context⊕) | Dist. Sup. | 35.96 | 24.02 | 68.63 | 62.50 | 17.70 | 7.41 | 22.97 | 28.33 |

Table 14: Ablation study. Comparing the performance across the in-context examples from distant supervision vs. human-annotated train set. Shown are F1 scores on each relation. (Part 2 / 12)

| Method | Context Source | P30 | P31 | P35 | P36 | P37 | P39 | P40 | P50 |
|---|---|---|---|---|---|---|---|---|---|
| REPLM (random fixed) | Train | 16.94 $\pm$ 6.32 | 0.00 $\pm$ 0.00 | 26.86 $\pm$ 3.49 | 16.56 $\pm$ 4.57 | 29.69 $\pm$ 10.49 | 0.00 $\pm$ 0.00 | 15.13 $\pm$ 6.04 | 28.38 $\pm$ 4.20 |
| REPLM (random fixed) | Dist. Sup. | 13.58 $\pm$ 1.66 | 5.21 $\pm$ 2.48 | 28.79 $\pm$ 4.96 | 18.67 $\pm$ 5.96 | 22.59 $\pm$ 8.79 | 0.00 $\pm$ 0.00 | 12.97 $\pm$ 3.62 | 27.66 $\pm$ 4.05 |
| REPLM (random all) | Train | 15.43 $\pm$ 2.93 | 6.66 $\pm$ 1.82 | 27.40 $\pm$ 3.48 | 19.92 $\pm$ 8.34 | 27.73 $\pm$ 3.60 | 0.00 $\pm$ 0.00 | 21.05 $\pm$ 3.09 | 28.01 $\pm$ 3.89 |
| REPLM (random all) | Dist. Sup. | 14.75 $\pm$ 2.18 | 6.33 $\pm$ 2.15 | 26.96 $\pm$ 3.17 | 14.85 $\pm$ 5.65 | 24.13 $\pm$ 5.48 | 0.00 $\pm$ 0.00 | 11.63 $\pm$ 3.12 | 26.74 $\pm$ 3.45 |
| REPLM (best context⊖) | Train | 19.14 | 11.90 | 27.27 | 22.22 | 36.36 | 15.38 | 29.14 | 34.18 |
| REPLM (best context⊖) | Dist. Sup. | 31.25 | 6.59 | 31.46 | 47.06 | 31.17 | 25.00 | 20.38 | 38.60 |
| REPLM (best context⊕) | Train | 27.84 | 9.09 | 21.78 | 17.54 | 34.21 | 0.00 | 30.87 | 40.26 |
| REPLM (best context⊕) | Dist. Sup. | 30.85 | 6.82 | 37.21 | 45.28 | 22.78 | 25.00 | 26.42 | 37.66 |

Table 15: Ablation study. Comparing the performance across the in-context examples from distant supervision vs. human-annotated train set. Shown are F1 scores on each relation. (Part 3 / 12)

| Method | Context Source | P54 | P57 | P58 | P69 | P86 | P102 | P108 | P112 |
|---|---|---|---|---|---|---|---|---|---|
| REPLM (random fixed) | Train | 36.25 $\pm$ 12.29 | 30.89 $\pm$ 1.64 | 25.41 $\pm$ 5.63 | 60.96 $\pm$ 4.38 | 17.27 $\pm$ 5.49 | 37.26 $\pm$ 5.68 | 34.50 $\pm$ 3.90 | 24.48 $\pm$ 9.40 |
| REPLM (random fixed) | Dist. Sup. | 42.71 $\pm$ 10.02 | 34.55 $\pm$ 4.59 | 27.65 $\pm$ 6.61 | 53.00 $\pm$ 9.62 | 16.16 $\pm$ 8.57 | 30.82 $\pm$ 9.53 | 33.29 $\pm$ 4.49 | 14.30 $\pm$ 8.81 |
| REPLM (random all) | Train | 39.88 $\pm$ 6.70 | 31.76 $\pm$ 1.75 | 27.32 $\pm$ 6.41 | 61.91 $\pm$ 2.44 | 17.29 $\pm$ 5.43 | 32.78 $\pm$ 2.11 | 32.23 $\pm$ 2.59 | 22.26 $\pm$ 6.95 |
| REPLM (random all) | Dist. Sup. | 43.80 $\pm$ 3.48 | 32.63 $\pm$ 3.59 | 27.32 $\pm$ 3.10 | 57.18 $\pm$ 3.74 | 13.23 $\pm$ 4.60 | 32.85 $\pm$ 3.32 | 32.67 $\pm$ 5.06 | 16.06 $\pm$ 6.68 |
| REPLM (best context⊖) | Train | 48.67 | 33.70 | 35.48 | 67.07 | 30.59 | 37.66 | 41.38 | 12.77 |
| REPLM (best context⊖) | Dist. Sup. | 48.30 | 47.62 | 34.38 | 57.47 | 23.26 | 44.44 | 30.19 | 32.65 |
| REPLM (best context⊕) | Train | 50.57 | 40.72 | 29.41 | 59.63 | 35.16 | 39.74 | 35.96 | 17.02 |
| REPLM (best context⊕) | Dist. Sup. | 40.93 | 43.43 | 39.34 | 58.29 | 40.45 | 40.48 | 34.29 | 34.78 |

Table 16: Ablation study. Comparing the performance across the in-context examples from distant supervision vs. human-annotated train set. Shown are F1 scores on each relation. (Part 4 / 12)

| Method | Context Source | P118 | P123 | P127 | P131 | P136 | P137 | P140 | P150 |
|--------|---------------|------|------|------|------|------|------|------|------|
| REPLM (random fixed) | Train | 33.05 ± 6.31 | 24.29 ± 3.37 | 10.23 ± 3.89 | 15.66 ± 2.99 | 27.79 ± 8.24 | 14.49 ± 4.58 | 8.38 ± 3.52 | 21.52 ± 3.58 |
| REPLM (random fixed) | Dist. Sup. | 32.16 ± 5.80 | 19.81 ± 4.20 | 11.10 ± 4.41 | 14.32 ± 2.81 | 19.70 ± 6.09 | 9.99 ± 3.84 | 0.00 ± 0.00 | 23.36 ± 2.23 |
| REPLM (random all) | Train | 32.13 ± 5.64 | 26.43 ± 3.86 | 13.17 ± 2.31 | 15.58 ± 0.62 | 25.20 ± 7.89 | 11.77 ± 3.69 | 6.73 ± 2.76 | 21.93 ± 1.81 |
| REPLM (random all) | Dist. Sup. | 33.51 ± 5.47 | 19.87 ± 3.15 | 10.50 ± 3.94 | 15.54 ± 0.65 | 21.93 ± 6.65 | 9.83 ± 2.77 | 0.00 ± 0.00 | 21.91 ± 1.52 |
| REPLM (best context⊖) | Train | 33.33 | 20.75 | 13.79 | 22.45 | 22.22 | 21.18 | 11.68 | 0.00 |
| REPLM (best context⊖) | Dist. Sup. | 44.04 | 30.19 | 18.49 | 25.50 | 20.00 | 12.90 | 14.17 | 31.28 |
| REPLM (best context⊕) | Train | 36.36 | 29.41 | 12.90 | 22.95 | 32.00 | 15.58 | 9.66 | 0.00 |
| REPLM (best context⊕) | Dist. Sup. | 37.84 | 19.82 | 18.03 | 26.59 | 26.09 | 9.84 | 13.53 | 30.25 |

Table 17: Ablation study. Comparing the performance across the in-context examples from distant supervision vs. human-annotated train set. Shown are F1 scores on each relation. (Part 5 / 12)

| Method | Context Source | P155 | P156 | P159 | P161 | P162 | P166 | P170 | P171 |
|--------|---------------|------|------|------|------|------|------|------|------|
| REPLM (random fixed) | Train | 5.28 ± 3.31 | 11.22 ± 5.91 | 24.25 ± 8.04 | 27.55 ± 7.43 | 18.40 ± 7.91 | 27.09 ± 2.02 | 5.44 ± 2.44 | 14.23 ± 3.10 |
| REPLM (random fixed) | Dist. Sup. | 6.51 ± 2.64 | 0.00 ± 0.00 | 23.72 ± 8.75 | 30.89 ± 7.21 | 13.31 ± 4.42 | 26.45 ± 3.74 | 6.57 ± 3.01 | 14.38 ± 6.09 |
| REPLM (random all) | Train | 5.45 ± 1.48 | 11.77 ± 3.67 | 25.24 ± 2.93 | 28.60 ± 4.73 | 20.51 ± 3.65 | 28.54 ± 2.88 | 0.00 ± 0.00 | 13.93 ± 4.04 |
| REPLM (random all) | Dist. Sup. | 5.64 ± 1.87 | 11.47 ± 3.78 | 23.76 ± 3.43 | 24.73 ± 4.62 | 14.47 ± 3.10 | 24.09 ± 3.17 | 5.48 ± 2.89 | 13.65 ± 3.79 |
| REPLM (best context⊖) | Train | 0.00 | 25.81 | 39.42 | 36.70 | 17.20 | 32.43 | 13.33 | 8.00 |
| REPLM (best context⊖) | Dist. Sup. | 23.33 | 21.51 | 40.85 | 33.85 | 14.12 | 26.92 | 10.00 | 10.53 |
| REPLM (best context⊕) | Train | 0.00 | 24.18 | 39.42 | 39.63 | 17.82 | 38.46 | 10.67 | 6.67 |
| REPLM (best context⊕) | Dist. Sup. | 22.22 | 29.21 | 37.24 | 30.81 | 14.81 | 31.58 | 11.43 | 10.53 |

Table 18: Ablation study. Comparing the performance across the in-context examples from distant supervision vs. human-annotated train set. Shown are F1 scores on each relation. (Part 6 / 12)

| Method | Context Source | P172 | P175 | P176 | P178 | P179 | P190 | P194 | P205 |
|--------|---------------|------|------|------|------|------|------|------|------|
| REPLM (random fixed) | Train | 13.59 ± 5.05 | 36.38 ± 8.61 | 11.89 ± 5.31 | 22.27 ± 3.21 | 14.80 ± 3.95 | 0.00 ± 0.00 | 12.67 ± 3.37 | 20.50 ± 7.77 |
| REPLM (random fixed) | Dist. Sup. | 19.33 ± 7.93 | 34.03 ± 4.85 | 13.62 ± 4.27 | 17.96 ± 4.72 | 10.19 ± 3.41 | 0.00 ± 0.00 | 11.62 ± 3.92 | 16.28 ± 7.56 |
| REPLM (random all) | Train | 11.62 ± 3.91 | 38.49 ± 2.49 | 12.14 ± 2.80 | 22.09 ± 2.07 | 12.95 ± 1.74 | 0.00 ± 0.00 | 13.66 ± 4.93 | 22.61 ± 7.69 |
| REPLM (random all) | Dist. Sup. | 16.76 ± 6.25 | 34.49 ± 2.53 | 13.54 ± 5.30 | 21.67 ± 4.32 | 8.56 ± 2.64 | 0.00 ± 0.00 | 13.79 ± 4.14 | 12.68 ± 4.19 |
| REPLM (best context⊖) | Train | 10.91 | 41.92 | 22.64 | 21.62 | 15.38 | 66.67 | 21.51 | 39.34 |
| REPLM (best context⊖) | Dist. Sup. | 23.73 | 40.71 | 24.14 | 28.57 | 22.45 | 100.00 | 19.57 | 13.56 |
| REPLM (best context⊕) | Train | 14.29 | 43.43 | 29.09 | 22.81 | 17.31 | 66.67 | 21.28 | 22.64 |
| REPLM (best context⊕) | Dist. Sup. | 34.48 | 46.66 | 23.08 | 28.99 | 23.91 | 66.67 | 25.81 | 14.04 |

Table 19: Ablation study. Comparing the performance across the in-context examples from distant supervision vs. human-annotated train set. Shown are F1 scores on each relation. (Part 7 / 12)

| Method | Context Source | P206 | P241 | P264 | P272 | P276 | P279 | P355 | P361 |
|--------|---------------|------|------|------|------|------|------|------|------|
| REPLM (random fixed) | Train | 7.59 ± 3.03 | 40.15 ± 9.19 | 35.63 ± 6.93 | 27.86 ± 5.07 | 8.40 ± 3.27 | 0.00 ± 0.00 | 0.00 ± 0.00 | 9.31 ± 4.73 |
| REPLM (random fixed) | Dist. Sup. | 6.68 ± 3.53 | 36.97 ± 9.98 | 28.62 ± 9.75 | 28.67 ± 3.68 | 8.32 ± 3.07 | 0.00 ± 0.00 | 14.43 ± 6.32 | 7.36 ± 4.70 |
| REPLM (random all) | Train | 5.84 ± 2.03 | 39.47 ± 3.73 | 32.63 ± 1.86 | 26.46 ± 3.73 | 10.13 ± 2.96 | 0.00 ± 0.00 | 15.48 ± 1.86 | 9.84 ± 1.61 |
| REPLM (random all) | Dist. Sup. | 0.00 ± 0.00 | 33.24 ± 3.41 | 29.66 ± 2.15 | 28.10 ± 4.05 | 9.88 ± 2.77 | 0.00 ± 0.00 | 11.26 ± 5.08 | 7.66 ± 1.82 |
| REPLM (best context⊖) | Train | 11.57 | 46.75 | 32.29 | 24.56 | 17.27 | 6.56 | 21.74 | 18.93 |
| REPLM (best context⊖) | Dist. Sup. | 14.63 | 43.24 | 30.77 | 35.09 | 20.00 | 11.11 | 30.77 | 27.12 |
| REPLM (best context⊕) | Train | 8.20 | 46.58 | 29.91 | 21.43 | 20.44 | 2.94 | 17.78 | 23.10 |
| REPLM (best context⊕) | Dist. Sup. | 18.18 | 45.07 | 28.35 | 37.29 | 22.76 | 10.91 | 33.33 | 25.66 |

Table 20: Ablation study. Comparing the performance across the in-context examples from distant supervision vs. human-annotated train set. Shown are F1 scores on each relation. (Part 8 / 12)

| Method | Context Source | P364 | P400 | P403 | P449 | P463 | P488 | P495 | P527 |
|--------|---------------|------|------|------|------|------|------|------|------|
| REPLM (random fixed) | Train | 24.96 ± 6.35 | 36.17 ± 10.65 | 22.47 ± 3.76 | 29.08 ± 6.31 | 19.08 ± 2.69 | 9.85 ± 2.21 | 15.27 ± 2.74 | 11.89 ± 2.03 |
| REPLM (random fixed) | Dist. Sup. | 22.50 ± 3.05 | 31.76 ± 10.44 | 27.58 ± 6.54 | 20.31 ± 4.25 | 16.47 ± 5.72 | 11.17 ± 1.87 | 12.08 ± 1.52 | 10.20 ± 2.71 |
| REPLM (random all) | Train | 21.75 ± 5.54 | 41.02 ± 4.49 | 21.09 ± 3.61 | 28.86 ± 3.35 | 16.58 ± 4.92 | 12.66 ± 4.46 | 12.95 ± 1.50 | 10.42 ± 2.31 |
| REPLM (random all) | Dist. Sup. | 23.18 ± 5.15 | 30.09 ± 7.83 | 27.12 ± 4.96 | 24.28 ± 5.57 | 15.54 ± 3.49 | 10.01 ± 2.80 | 11.91 ± 2.21 | 10.55 ± 1.76 |
| REPLM (best context⊖) | Train | 35.71 | 31.37 | 17.54 | 30.51 | 32.09 | 11.76 | 18.60 | 15.95 |
| REPLM (best context⊖) | Dist. Sup. | 52.00 | 40.00 | 18.52 | 33.33 | 38.61 | 18.18 | 22.71 | 23.31 |
| REPLM (best context⊕) | Train | 23.26 | 43.40 | 14.04 | 30.51 | 28.11 | 11.11 | 21.05 | 20.45 |
| REPLM (best context⊕) | Dist. Sup. | 43.14 | 29.36 | 25.00 | 27.69 | 33.80 | 17.14 | 21.55 | 27.63 |

Table 21: Ablation study. Comparing the performance across the in-context examples from distant supervision vs. human-annotated train set. Shown are F1 scores on each relation. (Part 9 / 12)

| Method | Context Source | P551 | P569 | P570 | P571 | P576 | P577 | P580 | P582 |
|---|---|---|---|---|---|---|---|---|---|
| REPLM (random fixed) | Train | 0.00 ± 0.00 | 54.19 ± 5.75 | 41.57 ± 3.37 | 30.85 ± 8.75 | 7.82 ± 3.26 | 39.58 ± 4.69 | 19.20 ± 3.61 | 24.98 ± 5.96 |
| REPLM (random fixed) | Dist. Sup. | 29.09 ± 8.91 | 51.90 ± 8.21 | 44.07 ± 3.16 | 24.60 ± 11.18 | 8.85 ± 2.46 | 32.29 ± 5.03 | 18.35 ± 5.51 | 0.00 ± 0.00 |
| REPLM (random all) | Train | 0.00 ± 0.00 | 55.28 ± 1.46 | 42.43 ± 1.70 | 37.54 ± 2.45 | 9.69 ± 2.11 | 36.66 ± 1.80 | 17.65 ± 4.91 | 24.41 ± 5.92 |
| REPLM (random all) | Dist. Sup. | 0.00 ± 0.00 | 55.43 ± 1.43 | 39.35 ± 1.82 | 23.17 ± 2.44 | 5.74 ± 2.03 | 36.61 ± 2.18 | 19.23 ± 3.65 | 11.71 ± 4.98 |
| REPLM (best context⊖) | Train | 18.18 | 57.10 | 44.35 | 39.73 | 9.23 | 43.08 | 25.81 | 35.90 |
| REPLM (best context⊖) | Dist. Sup. | 36.36 | 60.95 | 46.92 | 38.96 | 14.49 | 46.78 | 27.40 | 31.11 |
| REPLM (best context⊕) | Train | 18.18 | 56.56 | 45.55 | 36.49 | 5.88 | 42.28 | 22.95 | 34.15 |
| REPLM (best context⊕) | Dist. Sup. | 36.36 | 62.25 | 46.96 | 40.27 | 20.00 | 45.35 | 18.46 | 28.57 |

Table 22: Ablation study. Comparing the performance across the in-context examples from distant supervision vs. human-annotated train set. Shown are F1 scores on each relation. (Part 10 / 12)

| Method | Context Source | P585 | P607 | P674 | P676 | P706 | P710 | P737 | P740 |
|---|---|---|---|---|---|---|---|---|---|
| REPLM (random fixed) | Train | 16.10 ± 4.22 | 16.02 ± 3.99 | 25.00 ± 5.89 | 54.95 ± 6.90 | 0.00 ± 0.00 | 8.42 ± 6.01 | 0.00 ± 0.00 | 23.41 ± 4.25 |
| REPLM (random fixed) | Dist. Sup. | 15.58 ± 4.51 | 16.23 ± 3.52 | 23.46 ± 7.00 | 0.00 ± 0.00 | 0.00 ± 0.00 | 6.70 ± 6.66 | 0.00 ± 0.00 | 19.36 ± 7.43 |
| REPLM (random all) | Train | 18.76 ± 4.51 | 16.61 ± 1.99 | 22.46 ± 2.71 | 58.90 ± 2.15 | 11.93 ± 2.47 | 9.23 ± 5.93 | 0.00 ± 0.00 | 22.03 ± 8.04 |
| REPLM (random all) | Dist. Sup. | 17.37 ± 5.44 | 15.68 ± 1.45 | 24.97 ± 4.74 | 47.97 ± 7.83 | 6.26 ± 2.60 | 11.34 ± 2.80 | 0.00 ± 0.00 | 20.21 ± 8.17 |
| REPLM (best context⊖) | Train | 31.88 | 17.62 | 21.43 | 46.15 | 13.64 | 24.49 | 0.00 | 23.08 |
| REPLM (best context⊖) | Dist. Sup. | 25.64 | 26.09 | 26.42 | 61.54 | 16.33 | 27.96 | 0.00 | 38.46 |
| REPLM (best context⊕) | Train | 35.82 | 19.79 | 26.17 | 61.54 | 15.05 | 17.20 | 0.00 | 37.04 |
| REPLM (best context⊕) | Dist. Sup. | 24.66 | 26.37 | 34.29 | 61.54 | 12.24 | 23.40 | 0.00 | 37.04 |

Table 23: Ablation study. Comparing the performance across the in-context examples from distant supervision vs. human-annotated train set. Shown are F1 scores on each relation. (Part 11 / 12)

| Method | Context Source | P749 | P800 | P807 | P840 | P937 | P1001 | P1056 | P1198 |
|---|---|---|---|---|---|---|---|---|---|
| REPLM (random fixed) | Train | 11.47 ± 2.41 | 20.02 ± 4.19 | 0.00 ± 0.00 | 35.43 ± 6.25 | 23.90 ± 4.76 | 13.51 ± 5.41 | 0.00 ± 0.00 | 0.00 ± 0.00 |
| REPLM (random fixed) | Dist. Sup. | 10.39 ± 4.08 | 21.38 ± 2.96 | 0.00 ± 0.00 | 26.35 ± 5.02 | 23.37 ± 6.50 | 11.84 ± 5.84 | 0.00 ± 0.00 | 0.00 ± 0.00 |
| REPLM (random all) | Train | 11.32 ± 2.73 | 20.02 ± 2.55 | 0.00 ± 0.00 | 35.41 ± 8.47 | 22.45 ± 5.03 | 12.60 ± 1.68 | 0.00 ± 0.00 | 0.00 ± 0.00 |
| REPLM (random all) | Dist. Sup. | 9.11 ± 4.02 | 22.51 ± 2.95 | 0.00 ± 0.00 | 19.91 ± 3.92 | 24.87 ± 4.55 | 11.30 ± 2.91 | 0.00 ± 0.00 | 0.00 ± 0.00 |
| REPLM (best context⊖) | Train | 22.86 | 25.64 | 0.00 | 30.77 | 20.51 | 17.65 | 14.29 | 0.00 |
| REPLM (best context⊖) | Dist. Sup. | 25.35 | 31.46 | 66.67 | 46.15 | 21.28 | 22.06 | 0.00 | 100.00 |
| REPLM (best context⊕) | Train | 18.46 | 20.93 | 0.00 | 23.08 | 15.00 | 21.58 | 0.00 | 0.00 |
| REPLM (best context⊕) | Dist. Sup. | 25.35 | 29.27 | 66.67 | 46.15 | 22.22 | 29.20 | 0.00 | 100.00 |

Table 24: Ablation study. Comparing the performance across the in-context examples from distant supervision vs. human-annotated train set. Shown are F1 scores on each relation. (Part 12 / 12)

| Method | Context Source | P1336 | P1344 | P1365 | P1366 | P1376 | P1412 | P1441 | P3373 |
|---|---|---|---|---|---|---|---|---|---|
| REPLM (random fixed) | Train | 0.00 ± 0.00 | 19.81 ± 7.60 | 0.00 ± 0.00 | 0.00 ± 0.00 | 30.09 ± 7.72 | 20.64 ± 3.08 | 16.56 ± 4.57 | 22.31 ± 5.88 |
| REPLM (random fixed) | Dist. Sup. | 0.00 ± 0.00 | 20.41 ± 6.54 | 0.00 ± 0.00 | 0.00 ± 0.00 | 34.86 ± 10.12 | 24.65 ± 4.02 | 15.52 ± 5.30 | 22.74 ± 5.03 |
| REPLM (random all) | Train | 0.00 ± 0.00 | 18.93 ± 5.08 | 0.00 ± 0.00 | 0.00 ± 0.00 | 32.08 ± 7.43 | 19.19 ± 2.52 | 15.38 ± 2.60 | 21.81 ± 3.52 |
| REPLM (random all) | Dist. Sup. | 0.00 ± 0.00 | 16.99 ± 3.11 | 0.00 ± 0.00 | 0.00 ± 0.00 | 33.58 ± 3.41 | 25.84 ± 5.77 | 14.65 ± 4.25 | 25.44 ± 4.98 |
| REPLM (best context⊖) | Train | 0.00 | 33.33 | 0.00 | 0.00 | 29.27 | 17.39 | 16.97 | 24.04 |
| REPLM (best context⊖) | Dist. Sup. | 31.58 | 45.54 | 0.00 | 0.00 | 52.38 | 42.35 | 20.00 | 20.11 |
| REPLM (best context⊕) | Train | 0.00 | 33.01 | 0.00 | 0.00 | 34.15 | 33.33 | 18.60 | 30.37 |
| REPLM (best context⊕) | Dist. Sup. | 31.58 | 43.40 | 0.00 | 0.00 | 68.29 | 34.48 | 29.71 | 16.48 |

# E  IMPLEMENTATION DETAILS

We provide the details of our REPLM implementation in this section. We use GPT-JT[13] ($\sim$6B parameters) as our pre-trained LM for in-context few-shot learning. As the number of relations to be extracted is unknown in advance, we generate 200 tokens (for comparison, each extracted triplet consumes roughly 10-15 tokens) to ensure that our pre-trained LM can generate all relations it identifies.

As we use a fixed prefix for each extracted knowledge triplet at each line (e. g., "`Relation:`"), we easily identify if there is no further triplets extracted, simply from the absence of the prefix. We use a special separator token to easily parse the extracted subjects and objects. This separator is "`<==>`" in our experiments (which cannot be found in the original dataset and therefore cannot be confused with a natural text). We additionally inform our pre-trained LM about the task via starting our prompt with the instruction of the task. Here, we note that we have not done any prompt-tuning, since it is not the focus of this paper. For the output generation, we finally note that we use a greedy-decoding, e. g., not any sampling approach applied, which results in deterministic outputs given the input text. Example inputs and outputs can be found in Sec. G.

In our REPLM framework, we retrieve the semantically most-relevant in-context examples for each dev document. For this, we encode the documents via SBERT (Reimers & Gurevych, 2019) to calculate the embeddings and retrieve the most-relevant documents based on the cosine-similarity of the embeddings. We use the following fixed parameters in our framework (if not specified otherwise): $N = 20$, $K = 5$ $L = 5$, $\tau = 0.1$, and $\theta = 0.2$. As the sentences in sentence-level relation extraction datasets (CONLL04, NYT, and ADE) are shorter than the documents in DocRED, we used more in-context examples for these datasets, which is $K = 11$. The other parameters are the same as before.

We run all of the experiments of REPLM on NVIDIA Tesla V100-SXM2 32GB with a batch size of 4. For DocRED, on average, each batch is processed in $\sim$17.80 seconds. As a result, the dev set (998 documents) is processed in $\sim$74.17 minutes for each relation type.

For REPLM variants, we use gpt-3.5-turbo and gpt-4o of OpenAI and Llama-3.1-8B and Llama-3.1-70B from Meta as our backbone LMs. Similar to our design choice with GPT-JT, we opt for deterministic outputs from these LM backbones, which is done by choosing a low temperature such as 0.001. If not specified otherwise, we use exactly the same parameter configuration with our REPLM framework, specifically $N = 20$, $K = 5$ $L = 5$, $\tau = 0.1$, and $\theta = 0.2$.

---

[13]https://huggingface.co/togethercomputer/GPT-JT-6B-v1

## F    EMPIRICAL ANALYSIS OF THE DOCUMENTS THAT LACK THE ANNOTATION

We conjecture that REPLM generates more comprehensive output than REBEL, not because REPLM the probability threshold $\theta$ is too low but because REPLM identifies annotations that are missing in the dataset. To validate this empirically, we thus adopt a simple yet effective strategy to predict if a given document contains information about the given relation. Fig. 5 plots the average cosine similarity between (i) documents in the dev split and (ii) their top-$N$ nearest neighbors in distantly-supervised documents. The histogram shows two modes in distribution, where one corresponds to "known" relation and one to "missing" relations. In the overlap of two distributions, there are semantically similar documents that potentially include the relation but lack the annotation.

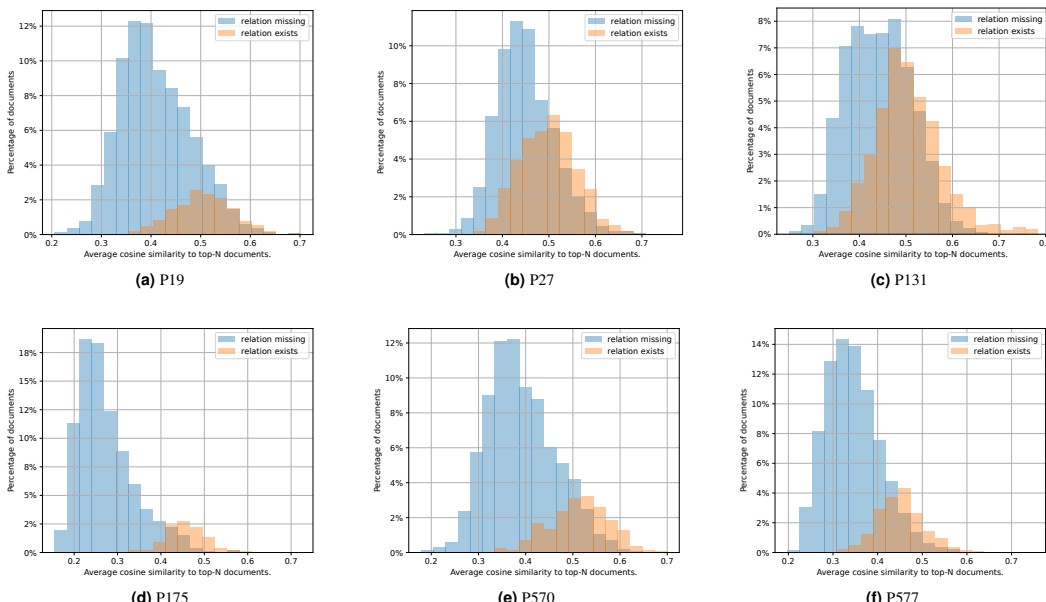

Figure 5: Histogram of average cosine similarity between documents and their top-$N$ neighbors for two example relation types.

## G  EXAMPLE PROMPTS AND OUTPUTS

In the following, we provide examples for different prompts as input (in red) and the corresponding output (in blue).

### G.1  P17 (COUNTRY)

**Input Prompt:**

Your task is to identify all the unique knowledge triplets of 'country' for a given context. Knowledge triplet will be ordered as relation, subject, and object, which are separated by <==>. If there are multiple triplets, list each of them in a new line. Follow the example context-relation pairs for the formatting of your output.

Context: IBM Laboratory Vienna was an IBM research laboratory based in Vienna , Austria . The laboratory started with a group led by Heinz Zemanek that moved from the Technische Hochschule ( now the Technical University of Vienna ) . Initially , the group worked on computer hardware projects . Later a compiler for the ALGOL 60 programming language was produced . The group built on ideas of Calvin C. Elgot , Peter Landin , and John McCarthy , to create an operational semantics that could define the whole of IBM 's PL / I programming language . The meta - language used for this was dubbed by people outside the laboratory as the Vienna Definition Language ( VDL ) . These descriptions were used for compiler design research into compiler design during 1968 - 70 . The formal method VDM ( Vienna Development Method ) was a result of research at the laboratory by Dines Bjørner , Cliff Jones , Peter Lucas , and others .

Relation: (country <==> Vienna <==> Austria)

Relation: (country <==> Technical University of Vienna <==> Austria)

Context: The School of Engineering of Juiz de Fora ( ) was an engineering college in the city of Juiz de Fora , Brazil . It is now the engineering faculty of the Federal University of Juiz de Fora ( UFJF ) . The former president of Brazil Itamar Franco was an alumnus . It was set up in 1914 in the city of Juiz de Fora , Minas Gerais state , Brazil , and taught a five - year course of Civil and Eletrotechnic Engineering . In 1960 , the school joined the Medicine , Pharmacy and Law schools of that city to found the UFJF . Nowadays , the Faculty of Engineering provides courses in civil , production , electrical ( divided into telecommunication , energy , power , electronic , robotic and automation systems ) , mechanical , computer , sanitary and environmental engineering , and architecture . Relation: (country <==> School of Engineering of Juiz de Fora <==> Brazil)

Relation: (country <==> Juiz de Fora <==> Brazil)

Relation: (country <==> UFJF <==> Brazil)

Relation: (country <==> Minas Gerais <==> Brazil)

Context: Bizrate Insights Inc. , doing business as Bizrate Insights , is a market research company , providing consumer ratings information to over 6,000 retailers and publishers across the United States , United Kingdom , France , Germany , and Canada . Bizrate Insights is a Meredith Corporation company based in Los Angeles , CA . Bizrate Insights provides services to both businesses and consumers in two different ways : consumers have access to ratings and reviews from verified buyers that help to inform their purchase decisions . This feedback can be found on the Bizrate website and is syndicated across the web to major search engines such as Google and Bing . Bizrate Insights provides businesses with customer satisfaction insights about consumers , advanced analytics , and competitive benchmarks across all types of online retail industries . Bizrate Insights also provides industry research to analysts at Forrester Research and Internet Retailer for publication and studies . Relation: (country <==> Meredith Corporation <==> the United States)

Relation: (country <==> Los Angeles <==> the United States)

Relation: (country <==> Google <==> the United States)

Relation: (country <==> Bing <==> the United States)

Context: The Universidade Positivo ( abbreviated UP ) is a private universities of the State of Paraná , Brazil . Universidade Positivo 's campus is in the Campo Comprido district of Curitiba and occupies an area of 420,004 m2 . Its 114,000 m2 of installations . Universidade Positivo currently offers 27 undergraduate programs , a doctoral program , three master 's degree programs , certificate of graduate study programs and extension courses . The 6,300 m2 library can hold 864 users at a time . Its collection consists of nearly 115,000 volumes . In a special room the library houses the personal collection of the late cabinet minister , diplomat and professor , Roberto de Oliveira Campos . It includes diplomas , medals and awards , in addition to its 8,426 volumes .

Relation: (country <==> Universidade Positivo <==> Brazil)

Relation: (country <==> State of Paraná <==> Brazil)

Relation: (country <==> Curitiba <==> Brazil)

Context: Softtek is a Mexican - based information technology company , operating in North America , Latin America , Europe and Asia . As of 2010 it was the largest private IT vendor in Latin America . Headquartered in Monterrey , Mexico , the company has 12,000 associates in Mexico and abroad . The company offers application software development , testing , security and support ; business process outsourcing ( BPO ) ; and IT infrastructure management , security and support to more than 400 corporations in more than 20 countries . It also acts as a value added reseller ( VAR ) for SAP SE , Informatica , Cognos , Business Objects and other software products . The company has trademarked the term " nearshoring " to describe the provision of outsourced services to customers in other countries that are in proximity .

Relation: (country <==> Softtek <==> Mexico)

Relation: (country <==> Monterrey <==> Mexico)

Context: IBM Research – Brazil is one of twelve research laboratories comprising IBM Research , its first in South America . It was established in June 2010 , with locations in São Paulo and Rio de Janeiro . Research focuses on Industrial Technology and Science , Systems of Engagement and Insight , Social Data Analytics and Natural Resources Solutions . The new lab , IBM 's ninth at the time of opening and first in 12 years , underscores the growing importance of emerging markets and the globalization of innovation . In collaboration with Brazil 's government , it will help IBM to develop technology systems around natural resource development and large - scale events such as the 2016 Summer Olympics . Engineer and associate lab director Ulisses Mello explains that IBM has four priority areas in Brazil : " The main area is related to natural resources management , involving oil and gas , mining and agricultural sectors . The second is the social data analytics segment that comprises the analysis of data generated from social networking sites [ such as Twitter or Facebook ] , which can be applied , for example , to financial analysis . The third strategic area is nanotechnology applied to the development of the smarter devices for the intermittent production industry . This technology can be applied to , for example , blood testing or recovering oil from existing fields . And the last one is smarter cities .

**Output:**

Relation: (country <==> IBM Research – Brazil <==> Brazil)

Relation: (country <==> São Paulo <==> Brazil)

Relation: (country <==> Rio de Janeiro <==> Brazil)

Relation: (country <==> 2016 Summer Olympics <==> Brazil)

Relation: (country <==> Industrial Technology and Science <==> Brazil)

**Correctness:**

For relation P17 (country), the given dev document is annotated with two knowledge triplets (country, São Paulo, Brazil) and (country, Rio de Janeiro, Brazil). For this document, we find that the annotators did not consider the country information of the event "2016 Summer Olympics" and the organization "IBM Research – Brazil" for their manual annotation. Validating from an external knowledge base, we observe that our REPLM framework identifies two more knowledge triplets correctly, (country, 2016 Summer Olympics, Brazil) and (country, Industrial Technology and Science, Brazil).

## G.2 P118 (LEAGUE)

**Input Prompt:**

Your task is to identify all the unique knowledge triplets of 'league' for a given context. Knowledge triplet will be ordered as relation, subject, and object, which are separated by <==>. If there are multiple triplets, list each of them in a new line. Follow the example context-relation pairs for the formatting of your output.

Context: Richard Ian Wright ( born 5 November 1977 ) is an English retired professional footballer who played as a goalkeeper . He currently works as a coach for Manchester City . Born in Ipswich , he joined Ipswich Town as a trainee , going on to play for the club 298 times between 1995 and 2001 . He then moved to Premier League club Arsenal , before being signed by Everton in 2002 , where he spent five years . A brief spell on loan from West Ham United with Southampton was followed by a transfer back to Ipswich Town . After a short spell at Sheffield United , a third stint at Ipswich and a brief time at Preston North End , he joined Premier League champions Manchester City on a free transfer in 2012 . After four years at City , during which he did not play at all , he announced his retirement in May 2016 . He remained with City as a coach under new manager Pep Guardiola . He was a member of the England squad , earning two caps , and was included in their squad for UEFA Euro 2000 .

Relation: (league <==> Manchester City <==> Premier League)

Relation: (league <==> Arsenal <==> Premier League)

Relation: (league <==> Everton <==> Premier League)

Relation: (league <==> West Ham United <==> Premier League)

Relation: (league <==> Southampton <==> Premier League)

Context: Ashley Renaldo Chambers ( born 1 March 1990 ) is an English professional footballer who plays as a winger or a striker for club Kidderminster Harriers . Chambers started his career with Leicester City , making his first - team debut in 2005 at the age of 15 in a League Cup match against Blackpool , which made him the youngest player in the club 's history . In 2009 , he joined League One club Wycombe Wanderers on loan . This was followed by a loan period with League Two club Grimsby Town . He signed for Conference Premier club York City on loan in November 2010 before signing permanently . He won in the 2012 FA Trophy Final and 2012 Conference Premier play - off Final with York at Wembley Stadium , the latter seeing the club promoted into League Two . Chambers joined Conference Premier club Cambridge United in 2014 .

Relation: (league <==> Kidderminster Harriers <==> Conference Premier)

Relation: (league <==> Leicester City <==> League One)

Context: Roy Eric Carroll ( born 30 September 1977 ) is a Northern Irish professional footballer who plays as a goalkeeper for NIFL Premiership side Linfield . He is best known for his spells at Wigan Athletic , Manchester United ( where he won a Premier League winners medal and the 2004 FA Cup ) and Olympiacos ( where he won the Greek Superleague three times and the Greek Cup twice ) . He has also represented Northern Ireland 45 times at full international level , gaining his first cap in 1997 , aged 19 . Carroll has also had a one - game managerial career , leading Barnet to a 2 - 1 victory in the 2011 Herts Senior Cup final against Stevenage . Therefore , Carroll holds the unusual honour of having won a trophy in his only game as a manager .

Relation: (league <==> Linfield <==> NIFL Premiership)

Relation: (league <==> Manchester United <==> Premier League)

Relation: (league <==> Olympiacos <==> Greek Superleague)

Context: Brian Christopher Deane ( born 7 February 1968 ) is an English football coach and former player whose most recent position was as the manager of the Norwegian side Sarpsborg 08 . During his playing career , he played as forward from 1985 until 2006 . He was the scorer of the first ever goal in the FA Premier League in 1992 , when he was a Sheffield United player . Deane also played in the Premier League for Leeds United and Middlesbrough as well as playing top - flight football in Portugal and Australia for Benfica and Perth Glory respectively . He also played in The Football

League for Doncaster Rovers , Leicester City , West Ham United and Sunderland before finishing his playing career in 2006 with a brief spell back at Sheffield United . Deane was capped three times by England .

Relation: (league <==> Middlesbrough <==> FA Premier League)

Relation: (league <==> Leicester City <==> FA Premier League)

Relation: (league <==> West Ham United <==> FA Premier League)

Context: John Stones ( born 28 May 1994 ) is an English professional footballer who plays for club Manchester City and the English national team . Mainly a centre back , he can also play as a right back . Stones began his career with Barnsley , making his first - team debut in the Championship in March 2012 as a 17-year - old . He joined Premier League club Everton for around £ 3 million in January 2013 and amassed 95 appearances over four seasons . In August 2016 , he signed for Manchester City for an initial £ 47.5 million with add - ons . He won the Premier League and EFL Cup in 2018 . Stones made his senior debut for England in May 2014 after previously being capped by England youth teams at under-19 , under-20 and under-21 levels . He was chosen in England 's squads for UEFA Euro 2016 and the 2018 FIFA World Cup .

Relation: (league <==> Manchester City <==> Premier League)

Relation: (league <==> Manchester City <==> EFL Cup)

Relation: (league <==> Barnsley <==> Championship)

Relation: (league <==> Everton <==> Premier League)

Context: Andrew Alexander Cole ( born 15 October 1971 ) is an English former professional footballer . Playing as a striker , his career lasted from 1988 to 2008 . He is most notably remembered for his time in the Premier League , with Manchester United , where he spent six years of his career , winning numerous trophies in the process . He also played in the top division of English football for Arsenal , Newcastle United , Blackburn Rovers , Fulham , Manchester City , Portsmouth and Sunderland , as well as in the Football League for Bristol City , Birmingham City , Burnley and Nottingham Forest . He is the third - highest goalscorer in Premier League history with 187 goals . Cole has the distinction of being one of the few players in England to have swept all possible honours in the English game , including the PFA Young Player of the Year award , as well as the coveted UEFA Champions League title . Cole was also capped 15 times for the England national team between 1995 and 2001 , scoring once against Albania in a 2002 FIFA World Cup qualifier .

**Output:**

Relation: (league <==> Manchester United <==> Premier League)

Relation: (league <==> Newcastle United <==> Premier League)

Relation: (league <==> Blackburn Rovers <==> Premier League)

Relation: (league <==> Fulham <==> Premier League)

Relation: (league <==> Manchester City <==> Premier League)

Relation: (league <==> Portsmouth <==> Premier League)

Relation: (league <==> Sunderland <==> Premier League)

**Correctness:**

For relation P118 (league), the given dev document is annotated with only one knowledge triplet (league, Manchester United, Premier League). In this example, our pre-trained LM identifies 6 more knowledge triplets via in-context few-shot learning. At the time of writing, the following triplets are also correct (league, Newcastle United, Premier League), (league, Fulham, Premier League), (league, Manchester City, Premier League). We note that although such information is outdated, Blackburn Rovers, Portsmouth, and Sunderland can be inferred as in Premier League from the same passage. This also highlights the need that the relation extraction methods should additionally infer (if possible) for which time period the extracted information is valid.

### G.3 P272 (PRODUCTION COMPANY)

**Input Prompt:**

Your task is to identify all the unique knowledge triplets of 'production company' for a given context. Knowledge triplet will be ordered as relation, subject, and object, which are separated by <==>. If there are multiple triplets, list each of them in a new line. Follow the example context-relation pairs for the formatting of your output.

Context: Ron Moody ( born Ronald Moodnick , 8 January 1924 - 11 June 2015 ) was an English actor , singer , composer and writer best known for his portrayal of Fagin in Oliver ! ( 1968 ) and its 1983 Broadway revival . Moody earned a Golden Globe Award and an Academy Award nomination for the film , as well as a Tony Award nomination for the stage production . Other notable projects include The Mouse on the Moon ( 1963 ) , Mel Brooks ' The Twelve Chairs ( 1970 ) and Flight of the Doves ( 1971 ) , in which Moody shared the screen with Oliver ! co - star Jack Wild . Moody holds the peculiar distinction of having portrayed the wizard Merlin in two Disney films , Unidentified Flying Oddball ( 1979 ) and A Kid in King Arthur 's Court ( 1995 ) .

Relation: (production company <==> Unidentified Flying Oddball <==> Disney)

Context: The Beastmaster is a 1982 sword and sorcery film directed by Don Coscarelli and starring Marc Singer , Tanya Roberts , John Amos and Rip Torn loosely based on the novel The Beast Master by Andre Norton . The film is about a child who is stolen from his mother 's womb by a witch . The child grows into Dar , who has the ability to communicate telepathically with animals . Dar grows up in a village where he learns to do battle . But the village is destroyed by a race of beast - like warriors under the control of the sorcerer Maax . Dar vows revenge and travels with new friends to stop Maax from causing any more problems . Commercially The Beastmaster was not considered a box office success during its original cinematic run ; however later it received extensive television exposure and success on cable in the American market on channels TBS and HBO . The original film spawned two sequels as well as a syndicated television series that chronicled the further adventures of Dar .

Relation: (production company <==> Beastmaster <==> Don Coscarelli)

Context: Monkeybone is a 2001 American black comedy dark fantasy film directed by Henry Selick , written by Sam Hamm , and produced by Selick , Hamm , Mark Radcliffe , Michael Barnathan , and Chris Columbus . The film combines live - action with stop - motion animation . Based on Kaja Blackley 's graphic novel Dark Town , the film stars an ensemble cast led by Brendan Fraser , Bridget Fonda , and Whoopi Goldberg with Rose McGowan , Dave Foley , Giancarlo Esposito , Megan Mullally , Lisa Zane , Chris Kattan , John Turturro , and an uncredited Thomas Haden Church . Theatrically released on February 23 , 2001 by 20th Century Fox , the film was a box office bomb and received generally negative critical reviews .

Relation: (production company <==> Monkeybone <==> 20th Century Fox)

Context: TaleSpin is an American animated television series based in the fictional city of Cape Suzette , that first aired in 1990 as a preview on The Disney Channel and later that year as part of The Disney Afternoon , with characters adapted from Disney 's 1967 animated feature The Jungle Book , which was theatrically rereleased in the summer before this show premiered in the fall . The name of the show is a play on tailspin , the rapid descent of an aircraft in a steep spiral . The two words in the show 's name , tale and spin , are a way to describe telling a story . The show is one of ten Disney Afternoon shows to use established Disney characters as the main characters , with the others being Darkwing Duck , DuckTales , Chip ' n Dale Rescue Rangers , Goof Troop , Bonkers , Quack Pack , Aladdin , Timon & Pumbaa and Jungle Cubs . It is also one of the two animated television series based on The Jungle Book along with Jungle Cubs .

Relation: (production company <==> Aladdin <==> Disney)

Context: Tarzan 's Hidden Jungle is a 1955 black - and - white film from RKO Pictures directed by Harold D. Schuster and starring Gordon Scott in his first film as Tarzan , taking over the role from Lex Barker , who had in turn followed Johnny Weissmuller in the series . The film about Edgar Rice Burroughs ' ape - man also features Vera Miles and Jack Elam . It was the last of twelve Tarzan pictures released by RKO . Tarzan 's mate , Jane , does not appear in the film . Tarzan at first seems to show more than casual interest in Miles ' character , but ultimately there is no romance . In real

life , Scott and Miles were married after the film was completed . Scott eventually played Tarzan in six movies over a five - year span .

Relation: (production company <==> Tarzan 's Hidden Jungle <==> RKO Pictures)

Context: King Louie is a fictional character introduced in Walt Disney 's 1967 animated musical film , The Jungle Book . Unlike the majority of the adapted characters in the film , Louie was not featured in Rudyard Kipling 's original works . King Louie was portrayed as an orangutan who was the leader of the other jungle primates , and who attempted to gain knowledge of fire from Mowgli , in order to become more human . King Louie was voiced by Louis Prima in the original 1967 film . Initially , the filmmakers considered Louis Armstrong for the role , but to avoid the likely controversy that would result from casting a black person to voice an ape , they instead chose Prima , a white singer . Christopher Walken voiced King Louie in the 2016 live - action / CGI The Jungle Book film .

**Output:**

Relation: (production company <==> The Jungle Book <==> Walt Disney)

**Correctness:**

For relation P637 (production company), there was no annotation for the given dev document. However, our framework identifies the knowledge triplet (production company, The Jungle Book, Walt Disney). This can easily be inferred from the first sentence of the document.

## H  FULL RESULTS ON OVERALL PERFORMANCE

Tables 25 to 36 provide the detailed results of relation extraction of all methods on all relation types. For methods with random in-context examples, the performance may be subject to variability across which seed is picked (whereas the performance is deterministic for the other methods), and, hence, we report the standard deviation for this subset of the methods by averaging the performance across 10 runs. The evaluation is done based on the human annotations of the dev documents.

When we compare the different variants of our framework, the results confirm our choice in the complete REPLM framework. Specifically, we find that (1) retrieving the best in-context examples improves the performance compared to random examples (i. e., REPLM (best context⊖) and REPLM (best context⊕) vs. REPLM (random fixed) and REPLM (random all)) and (2) our complete REPLM framework brings a significant improvement over REPLM (best context⊖) and REPLM (best context⊕) by aggregating multiple sets of most relevant in-context examples, thus establishing the importance of using multiple sets. We further compare our framework against the state-of-the-art relation extraction method REBEL and find that, overall, our REPLM and REPLM (params adj) perform the best.

Table 25: Full results in overall performance (Section 6.1). Shown are F1 scores on each relation. (Part 1 / 12)

| Method | P6 | P17 | P19 | P20 | P22 | P25 | P26 | P27 |
|---|---|---|---|---|---|---|---|---|
| REBEL | 0.00 | 20.66 | 50.89 | 29.41 | 40.86 | 38.10 | 41.32 | 5.69 |
| REBEL-sent | 4.08 | 25.73 | 45.95 | 29.85 | 33.71 | 30.00 | 41.84 | 11.37 |
| REPLM (random fixed) | 24.28 ± 10.53 | 11.34 ± 4.48 | 66.67 ± 12.71 | 49.80 ± 16.07 | 9.76 ± 4.90 | 0.00 ± 0.00 | 26.73 ± 4.38 | 22.41 ± 2.53 |
| REPLM (random all) | 25.90 ± 2.44 | 11.05 ± 0.46 | 68.49 ± 3.09 | 48.35 ± 4.82 | 9.51 ± 2.86 | 0.00 ± 0.00 | 26.74 ± 3.83 | 23.26 ± 1.39 |
| REPLM (best context⊖) | 35.96 | 24.60 | 71.38 | 50.39 | 18.18 | 15.38 | 33.33 | 29.14 |
| REPLM (best context⊕) | 35.96 | 24.02 | 68.63 | 62.50 | 17.70 | 7.41 | 22.97 | 28.33 |
| REPLM | 34.78 | 27.74 | 59.06 | 40.41 | 17.17 | 28.00 | 37.17 | 30.71 |
| REPLM (params adj) | 29.79 | 27.85 | 77.49 | 59.77 | 15.54 | 22.54 | 40.24 | 34.64 |

Table 26: Full results in overall performance (Section 6.1). Shown are F1 scores on each relation. (Part 2 / 12)

| Method | P30 | P31 | P35 | P36 | P37 | P39 | P40 | P50 |
|---|---|---|---|---|---|---|---|---|
| REBEL | 6.40 | 9.84 | 0.00 | 57.89 | 0.00 | 46.15 | 39.02 | 42.76 |
| REBEL-sent | 10.29 | 16.16 | 3.70 | 55.81 | 0.00 | 33.33 | 35.00 | 51.81 |
| REPLM (random fixed) | 13.58 ± 1.66 | 5.21 ± 2.48 | 28.79 ± 4.96 | 18.67 ± 5.96 | 22.59 ± 8.79 | 0.00 ± 0.00 | 12.97 ± 3.62 | 27.66 ± 4.05 |
| REPLM (random all) | 14.75 ± 2.18 | 6.33 ± 2.15 | 26.96 ± 3.17 | 14.85 ± 5.65 | 24.13 ± 5.48 | 0.00 ± 0.00 | 11.63 ± 3.12 | 26.74 ± 3.45 |
| REPLM (best context⊖) | 31.25 | 6.59 | 31.46 | 47.06 | 31.17 | 25.00 | 20.38 | 38.60 |
| REPLM (best context⊕) | 30.85 | 6.82 | 37.21 | 45.28 | 22.78 | 25.00 | 26.42 | 37.66 |
| REPLM | 39.11 | 14.29 | 32.56 | 52.46 | 27.91 | 8.70 | 17.56 | 41.12 |
| REPLM (params adj) | 38.94 | 14.04 | 29.58 | 52.83 | 29.73 | 12.50 | 20.38 | 41.12 |

Table 27: Full results in overall performance (Section 6.1). Shown are F1 scores on each relation. (Part 3 / 12)

| Method | P54 | P57 | P58 | P69 | P86 | P102 | P108 | P112 |
|---|---|---|---|---|---|---|---|---|
| REBEL | 38.06 | 48.53 | 19.05 | 41.38 | 26.67 | 49.32 | 26.87 | 27.78 |
| REBEL-sent | 25.22 | 40.60 | 5.26 | 5.31 | 16.90 | 60.61 | 26.09 | 20.51 |
| REPLM (random fixed) | 42.71 ± 10.02 | 34.55 ± 4.59 | 27.65 ± 6.61 | 53.00 ± 9.62 | 16.16 ± 8.57 | 30.82 ± 9.53 | 33.29 ± 4.49 | 14.30 ± 8.81 |
| REPLM (random all) | 43.80 ± 3.48 | 32.63 ± 3.59 | 27.32 ± 3.10 | 57.18 ± 3.74 | 13.23 ± 4.60 | 32.85 ± 3.32 | 32.67 ± 5.06 | 16.06 ± 6.68 |
| REPLM (best context⊖) | 48.30 | 47.62 | 34.38 | 57.47 | 23.26 | 44.44 | 30.19 | 32.65 |
| REPLM (best context⊕) | 40.93 | 43.43 | 39.34 | 58.29 | 40.45 | 40.48 | 34.29 | 34.78 |
| REPLM | 45.78 | 43.70 | 30.77 | 56.25 | 51.33 | 42.26 | 31.88 | 31.43 |
| REPLM (params adj) | 43.42 | 45.99 | 35.82 | 64.41 | 51.33 | 45.90 | 34.00 | 37.21 |

Table 28: Full results in overall performance (Section 6.1). Shown are F1 scores on each relation. (Part 4 / 12)

| Method | P118 | P123 | P127 | P131 | P136 | P137 | P140 | P150 |
|---|---|---|---|---|---|---|---|---|
| REBEL | 34.21 | 27.59 | 13.48 | 31.01 | 19.05 | 16.67 | 4.65 | 33.49 |
| REBEL-sent | 40.74 | 30.11 | 18.69 | 28.61 | 16.67 | 8.16 | 14.81 | 39.53 |
| REPLM (random fixed) | 32.16 ± 5.80 | 19.81 ± 4.20 | 11.10 ± 4.41 | 14.32 ± 2.81 | 19.70 ± 6.09 | 9.99 ± 3.84 | 0.00 ± 0.00 | 23.36 ± 2.23 |
| REPLM (random all) | 33.51 ± 5.47 | 19.87 ± 3.15 | 10.50 ± 3.94 | 15.54 ± 0.65 | 21.93 ± 6.65 | 9.83 ± 2.77 | 0.00 ± 0.00 | 21.91 ± 1.52 |
| REPLM (best context⊖) | 44.04 | 30.19 | 18.49 | 25.50 | 20.00 | 12.90 | 14.17 | 31.28 |
| REPLM (best context⊕) | 37.84 | 19.82 | 18.03 | 26.59 | 26.09 | 9.84 | 13.53 | 30.25 |
| REPLM | 39.75 | 32.47 | 21.52 | 29.89 | 18.18 | 17.72 | 14.29 | 34.36 |
| REPLM (params adj) | 46.28 | 35.77 | 21.79 | 29.79 | 18.18 | 17.95 | 14.55 | 33.75 |

Table 29: Full results in overall performance (Section 6.1). Shown are F1 scores on each relation. (Part 5 / 12)

| Method | P155 | P156 | P159 | P161 | P162 | P166 | P170 | P171 |
|---|---|---|---|---|---|---|---|---|
| REBEL | 2.70 | 7.02 | 22.02 | 37.96 | 10.91 | 28.92 | 28.07 | 16.00 |
| REBEL-sent | 16.33 | 24.39 | 35.90 | 17.74 | 7.27 | 38.64 | 31.58 | 13.79 |
| REPLM (random fixed) | 6.51 ± 2.64 | 0.00 ± 0.00 | 23.72 ± 8.75 | 30.89 ± 7.21 | 13.31 ± 4.42 | 26.45 ± 3.74 | 6.57 ± 3.01 | 14.38 ± 6.09 |
| REPLM (random all) | 5.64 ± 1.87 | 11.47 ± 3.78 | 23.76 ± 3.43 | 24.73 ± 4.62 | 14.47 ± 3.10 | 24.09 ± 3.17 | 5.48 ± 2.89 | 13.65 ± 3.79 |
| REPLM (best context⊖) | 23.33 | 21.51 | 40.85 | 33.85 | 14.12 | 26.92 | 10.00 | 10.53 |
| REPLM (best context⊕) | 22.22 | 29.21 | 37.24 | 30.81 | 14.81 | 31.58 | 11.43 | 10.53 |
| REPLM | 16.09 | 29.93 | 45.03 | 45.23 | 23.81 | 26.42 | 15.52 | 14.63 |
| REPLM (params adj) | 16.47 | 30.07 | 42.11 | 46.21 | 23.64 | 27.59 | 17.50 | 10.81 |

Table 30: Full results in overall performance (Section 6.1). Shown are F1 scores on each relation. (Part 6 / 12)

| Method | P172 | P175 | P176 | P178 | P179 | P190 | P194 | P205 |
|---|---|---|---|---|---|---|---|---|
| REBEL | 0.00 | 47.90 | 13.04 | 28.32 | 0.00 | 0.00 | 10.00 | 11.76 |
| REBEL-sent | 0.00 | 45.88 | 32.79 | 28.04 | 0.00 | 0.00 | 17.39 | 16.22 |
| REPLM (random fixed) | 19.33 ± 7.93 | 34.03 ± 4.85 | 13.62 ± 4.27 | 17.96 ± 4.72 | 10.19 ± 3.41 | 0.00 ± 0.00 | 11.62 ± 3.92 | 16.28 ± 7.56 |
| REPLM (random all) | 16.76 ± 6.25 | 34.49 ± 2.53 | 13.54 ± 5.30 | 21.67 ± 4.32 | 8.56 ± 2.64 | 0.00 ± 0.00 | 13.79 ± 4.14 | 12.68 ± 4.19 |
| REPLM (best context⊖) | 23.73 | 40.71 | 24.14 | 28.57 | 22.45 | 100.00 | 19.57 | 13.56 |
| REPLM (best context⊕) | 34.48 | 46.66 | 23.08 | 28.99 | 23.91 | 66.67 | 25.81 | 14.04 |
| REPLM | 32.91 | 52.19 | 28.17 | 31.75 | 27.27 | 66.67 | 30.65 | 25.35 |
| REPLM (params adj) | 30.77 | 53.27 | 29.73 | 31.75 | 23.30 | 66.67 | 21.51 | 27.12 |

Table 31: Full results in overall performance (Section 6.1). Shown are F1 scores on each relation. (Part 7 / 12)

| Method | P206 | P241 | P264 | P272 | P276 | P279 | P355 | P361 |
|---|---|---|---|---|---|---|---|---|
| REBEL | 34.11 | 26.42 | 42.61 | 44.90 | 19.05 | 4.55 | 6.06 | 37.45 |
| REBEL-sent | 31.25 | 12.24 | 27.39 | 48.00 | 26.42 | 15.15 | 23.53 | 26.09 |
| REPLM (random fixed) | 6.68 ± 3.53 | 36.97 ± 9.98 | 28.62 ± 9.75 | 28.67 ± 3.68 | 8.32 ± 3.07 | 0.00 ± 0.00 | 14.43 ± 6.32 | 7.36 ± 4.70 |
| REPLM (random all) | 0.00 ± 0.00 | 33.24 ± 3.41 | 29.66 ± 2.15 | 28.10 ± 4.05 | 9.88 ± 2.77 | 0.00 ± 0.00 | 11.26 ± 5.08 | 7.66 ± 1.82 |
| REPLM (best context⊖) | 14.63 | 43.24 | 30.77 | 35.09 | 20.00 | 11.11 | 30.77 | 27.12 |
| REPLM (best context⊕) | 18.18 | 45.07 | 28.35 | 37.29 | 22.76 | 10.91 | 33.33 | 25.66 |
| REPLM | 20.51 | 46.00 | 45.54 | 36.36 | 30.12 | 9.09 | 22.22 | 25.39 |
| REPLM (params adj) | 11.57 | 53.12 | 46.75 | 36.36 | 32.79 | 8.96 | 28.07 | 25.33 |

Table 32: Full results in overall performance (Section 6.1). Shown are F1 scores on each relation. (Part 8 / 12)

| Method | P364 | P400 | P403 | P449 | P463 | P488 | P495 | P527 |
|---|---|---|---|---|---|---|---|---|
| REBEL | 0.00 | 36.54 | 38.46 | 0.00 | 43.90 | 17.39 | 3.54 | 37.04 |
| REBEL-sent | 0.00 | 36.17 | 44.12 | 0.00 | 40.23 | 8.33 | 5.69 | 34.45 |
| REPLM (random fixed) | 22.50 ± 3.05 | 31.76 ± 10.44 | 27.58 ± 6.54 | 20.31 ± 4.25 | 16.47 ± 5.72 | 11.17 ± 1.87 | 12.08 ± 1.52 | 10.20 ± 2.71 |
| REPLM (random all) | 23.18 ± 5.15 | 30.09 ± 7.83 | 27.12 ± 4.96 | 24.28 ± 5.57 | 15.54 ± 3.49 | 10.01 ± 2.80 | 11.91 ± 2.21 | 10.55 ± 1.76 |
| REPLM (best context⊖) | 52.00 | 40.00 | 18.52 | 33.33 | 38.61 | 18.18 | 22.71 | 23.31 |
| REPLM (best context⊕) | 43.14 | 29.36 | 25.00 | 27.69 | 33.80 | 17.14 | 21.55 | 27.63 |
| REPLM | 39.34 | 34.01 | 27.03 | 38.10 | 42.07 | 14.81 | 22.41 | 21.74 |
| REPLM (params adj) | 38.10 | 35.21 | 21.28 | 39.39 | 41.05 | 20.00 | 21.82 | 23.49 |

Table 33: Full results in overall performance (Section 6.1). Shown are F1 scores on each relation. (Part 9 / 12)

| Method | P551 | P569 | P570 | P571 | P576 | P577 | P580 | P582 |
|---|---|---|---|---|---|---|---|---|
| REBEL | 0.00 | 50.51 | 39.19 | 50.93 | 0.00 | 35.32 | 21.05 | 0.00 |
| REBEL-sent | 25.00 | 51.32 | 44.44 | 42.59 | 0.00 | 42.79 | 23.81 | 0.00 |
| REPLM (random fixed) | 29.09 ± 8.91 | 51.90 ± 8.21 | 44.07 ± 3.16 | 24.60 ± 11.18 | 8.85 ± 2.46 | 32.29 ± 5.03 | 18.35 ± 5.51 | 0.00 ± 0.00 |
| REPLM (random all) | 0.00 ± 0.00 | 55.43 ± 1.43 | 39.35 ± 1.82 | 23.17 ± 2.44 | 5.74 ± 2.03 | 36.61 ± 2.18 | 19.23 ± 3.65 | 11.71 ± 4.98 |
| REPLM (best context⊖) | 36.36 | 60.95 | 46.92 | 38.96 | 14.49 | 46.78 | 27.40 | 31.11 |
| REPLM (best context⊕) | 36.36 | 62.25 | 46.96 | 40.27 | 20.00 | 45.35 | 18.46 | 28.57 |
| REPLM | 28.57 | 55.73 | 44.56 | 39.71 | 17.78 | 52.39 | 29.17 | 43.14 |
| REPLM (params adj) | 33.33 | 61.29 | 50.66 | 46.78 | 19.35 | 54.40 | 35.48 | 46.15 |

Table 34: Full results in overall performance (Section 6.1). Shown are F1 scores on each relation. (Part 10 / 12)

| Method | P585 | P607 | P674 | P676 | P706 | P710 | P737 | P740 |
|---|---|---|---|---|---|---|---|---|
| REBEL | 11.32 | 42.17 | 28.26 | 0.00 | 5.88 | 25.32 | 0.00 | 12.50 |
| REBEL-sent | 28.57 | 38.64 | 21.15 | 0.00 | 2.94 | 20.22 | 0.00 | 10.53 |
| REPLM (random fixed) | 15.58 ± 4.51 | 16.23 ± 3.52 | 23.46 ± 7.00 | 0.00 ± 0.00 | 0.00 ± 0.00 | 6.70 ± 6.66 | 0.00 ± 0.00 | 19.36 ± 7.43 |
| REPLM (random all) | 17.37 ± 5.44 | 15.68 ± 1.45 | 24.97 ± 4.74 | 47.97 ± 7.83 | 6.26 ± 2.60 | 11.34 ± 2.80 | 0.00 ± 0.00 | 20.21 ± 8.17 |
| REPLM (best context⊖) | 25.64 | 26.09 | 26.42 | 61.54 | 16.33 | 27.96 | 0.00 | 38.46 |
| REPLM (best context⊕) | 24.66 | 26.37 | 34.29 | 61.54 | 12.24 | 23.40 | 0.00 | 37.04 |
| REPLM | 30.11 | 33.20 | 31.25 | 47.06 | 23.02 | 30.19 | 0.00 | 31.58 |
| REPLM (params adj) | 28.26 | 27.69 | 23.30 | 46.15 | 17.39 | 23.66 | 0.00 | 31.25 |

Table 35: Full results in overall performance (Section 6.1). Shown are F1 scores on each relation. (Part 11 / 12)

| Method | P749 | P800 | P807 | P840 | P937 | P1001 | P1056 | P1198 |
|---|---|---|---|---|---|---|---|---|
| REBEL | 4.26 | 39.47 | 0.00 | 0.00 | 0.00 | 15.05 | 0.00 | 0.00 |
| REBEL-sent | 32.35 | 43.90 | 0.00 | 0.00 | 0.00 | 1.94 | 0.00 | 0.00 |
| REPLM (random fixed) | 10.39 ± 4.08 | 21.38 ± 2.96 | 0.00 ± 0.00 | 26.35 ± 5.02 | 23.37 ± 6.50 | 11.84 ± 5.84 | 0.00 ± 0.00 | 0.00 ± 0.00 |
| REPLM (random all) | 9.11 ± 4.02 | 22.51 ± 2.95 | 0.00 ± 0.00 | 19.91 ± 3.92 | 24.87 ± 4.55 | 11.30 ± 2.91 | 0.00 ± 0.00 | 0.00 ± 0.00 |
| REPLM (best context⊖) | 25.35 | 31.46 | 66.67 | 46.15 | 21.28 | 22.06 | 0.00 | 100.00 |
| REPLM (best context⊕) | 25.35 | 29.27 | 66.67 | 46.15 | 22.22 | 29.20 | 0.00 | 100.00 |
| REPLM | 24.49 | 29.27 | 50.00 | 34.29 | 28.57 | 28.57 | 0.00 | 100.00 |
| REPLM (params adj) | 21.05 | 32.56 | 50.00 | 30.77 | 21.74 | 28.57 | 0.00 | 66.67 |

Table 36: Full results in overall performance (Section 6.1). Shown are F1 scores on each relation. (Part 12 / 12)

| Method | P1336 | P1344 | P1365 | P1366 | P1376 | P1412 | P1441 | P3373 |
|---|---|---|---|---|---|---|---|---|
| REBEL | 0.00 | 0.00 | 18.18 | 18.18 | 25.00 | 0.00 | 27.78 | 41.81 |
| REBEL-sent | 0.00 | 0.00 | 33.33 | 23.53 | 24.00 | 0.00 | 18.54 | 46.35 |
| REPLM (random fixed) | 0.00 ± 0.00 | 20.41 ± 6.54 | 0.00 ± 0.00 | 0.00 ± 0.00 | 34.86 ± 10.12 | 24.65 ± 4.02 | 15.52 ± 5.30 | 22.74 ± 5.03 |
| REPLM (random all) | 0.00 ± 0.00 | 16.99 ± 3.11 | 0.00 ± 0.00 | 0.00 ± 0.00 | 33.58 ± 3.41 | 25.84 ± 5.77 | 14.65 ± 4.25 | 25.44 ± 4.98 |
| REPLM (best context⊖) | 31.58 | 45.54 | 0.00 | 0.00 | 52.38 | 42.35 | 20.00 | 20.11 |
| REPLM (best context⊕) | 31.58 | 43.40 | 0.00 | 0.00 | 68.29 | 34.48 | 29.71 | 16.48 |
| REPLM | 26.09 | 35.80 | 0.00 | 0.00 | 55.17 | 28.81 | 30.15 | 28.22 |
| REPLM (params adj) | 26.09 | 35.58 | 0.00 | 0.00 | 60.87 | 28.81 | 30.61 | 29.27 |

# I   FULL RESULTS ON COMPARISON AGAINST EXTERNAL KNOWLEDGE

Tables 37 to 44 provide the detailed results of relation extraction of all methods on all relation types. The evaluation is done based on checking the correctness of extracted relations against both the human annotations of the dev documents *and* an external knowledge base.

Overall, we see that our REPLM and REPLM (params adj) outperform REBEL in most relations, when further including the relations from an external knowledge base. The performance improvement becomes more striking for the relation types having a large number of knowledge triplets. For instance, for the relation P17 (country), our REPLM achieves an F1 score of 56.14, whereas REBEL-sent can achieve less than half of the performance, that is, an F1 score of 21.17. We want to highlight that, these methods performed at a similar level when compared against only human-annotations as ground-truth (REPLM with 27.74 F1 score vs. REBEL-sent with 25.73 F1 score, see Sec. H). Therefore, it shows the importance of evaluating against an external knowledge base. Even a larger performance gap can be found in relation P27 (country of citizenship), our REPLM and REPLM (params adj) achieve the F1 scores 39.52 and 46.15, respectively, whereas REBEL and REBEL-sent can achieve only F1 scores 3.47 and 8.24. These results further confirm the effectiveness of our REPLM framework, agreeing with the results from the earlier sections.

Table 37: Full results in comparison against external knowledge (Section 6.2). Shown are F1 scores on each relation. (Part 1 / 8)

| Method | P6 | P17 | P19 | P20 | P22 | P25 | P26 | P27 | P30 | P31 | P35 | P36 |
|---|---|---|---|---|---|---|---|---|---|---|---|---|
| REBEL | 0.00 | 17.08 | 37.46 | 13.92 | 35.09 | 30.77 | 35.86 | 3.47 | 3.22 | 4.80 | 0.00 | 15.34 |
| REBEL-sent | 2.13 | 21.17 | 34.55 | 14.29 | 27.74 | 26.32 | 30.66 | 7.72 | 8.24 | 2.81 | 2.78 | 16.90 |
| REPLM | 14.81 | 56.14 | 33.83 | 10.90 | 3.86 | 9.16 | 6.40 | 39.52 | 49.16 | 9.78 | 5.85 | 14.63 |
| REPLM (params adj) | 14.04 | 55.32 | 53.67 | 37.84 | 12.68 | 10.31 | 4.08 | 46.15 | 39.22 | 13.89 | 13.86 | 26.94 |

Table 38: Full results in comparison against external knowledge (Section 6.2). Shown are F1 scores on each relation. (Part 2 / 8)

| Method | P37 | P39 | P40 | P50 | P54 | P57 | P58 | P69 | P86 | P102 | P108 | P112 |
|---|---|---|---|---|---|---|---|---|---|---|---|---|
| REBEL | 0.00 | 30.11 | 34.97 | 43.27 | 37.84 | 48.05 | 16.90 | 35.79 | 31.25 | 47.13 | 27.91 | 15.38 |
| REBEL-sent | 0.00 | 13.39 | 28.07 | 40.89 | 24.00 | 41.56 | 2.82 | 5.23 | 15.38 | 57.58 | 20.00 | 10.31 |
| REPLM | 8.79 | 2.50 | 2.75 | 14.15 | 45.04 | 38.82 | 24.78 | 37.76 | 19.48 | 37.04 | 6.01 | 3.12 |
| REPLM (params adj) | 18.60 | 15.00 | 4.97 | 30.84 | 45.04 | 38.32 | 25.23 | 35.23 | 25.64 | 37.78 | 22.22 | 12.96 |

Table 39: Full results in comparison against external knowledge (Section 6.2). Shown are F1 scores on each relation. (Part 3 / 8)

| Method | P118 | P123 | P127 | P131 | P136 | P137 | P140 | P150 | P155 | P156 | P159 | P161 |
|---|---|---|---|---|---|---|---|---|---|---|---|---|
| REBEL | 29.09 | 26.67 | 12.28 | 28.54 | 4.29 | 12.50 | 4.12 | 29.01 | 6.67 | 8.06 | 13.86 | 42.52 |
| REBEL-sent | 36.99 | 23.44 | 12.31 | 26.03 | 2.37 | 5.06 | 12.70 | 33.80 | 7.91 | 12.31 | 21.67 | 21.20 |
| REPLM | 34.55 | 25.24 | 6.61 | 38.54 | 19.40 | 9.09 | 0.00 | 43.21 | 9.52 | 4.88 | 21.88 | 24.62 |
| REPLM (params adj) | 33.63 | 26.92 | 11.03 | 42.39 | 13.33 | 9.52 | 8.55 | 47.81 | 10.98 | 11.70 | 23.85 | 32.56 |

Table 40: Full results in comparison against external knowledge (Section 6.2). Shown are F1 scores on each relation. (Part 4 / 8)

| Method | P162 | P166 | P170 | P171 | P172 | P175 | P176 | P178 | P179 | P190 | P194 | P205 |
|---|---|---|---|---|---|---|---|---|---|---|---|---|
| REBEL | 10.20 | 31.37 | 25.00 | 10.81 | 0.00 | 49.91 | 15.62 | 30.43 | 0.00 | 0.00 | 9.09 | 5.56 |
| REBEL-sent | 6.38 | 36.21 | 24.44 | 6.45 | 0.00 | 46.49 | 32.18 | 32.06 | 0.00 | 0.00 | 14.14 | 15.58 |
| REPLM | 11.04 | 5.94 | 6.90 | 27.91 | 12.28 | 42.04 | 24.76 | 1.06 | 0.62 | 6.85 | 9.30 | 12.50 |
| REPLM (params adj) | 10.48 | 26.79 | 10.34 | 26.32 | 21.97 | 42.16 | 24.72 | 16.36 | 19.51 | 6.18 | 14.43 | 14.46 |

Table 41: Full results in comparison against external knowledge (Section 6.2). Shown are F1 scores on each relation. (Part 5 / 8)

| Method | P206 | P241 | P264 | P272 | P276 | P279 | P355 | P361 | P364 | P400 | P403 | P449 |
|---|---|---|---|---|---|---|---|---|---|---|---|---|
| REBEL | 28.88 | 27.59 | 44.39 | 38.71 | 18.03 | 5.00 | 10.67 | 27.42 | 0.00 | 37.38 | 36.11 | 0.00 |
| REBEL-sent | 24.91 | 12.90 | 27.37 | 40.58 | 18.00 | 2.28 | 14.37 | 12.41 | 0.00 | 34.00 | 36.89 | 0.00 |
| REPLM | 4.41 | 3.96 | 0.00 | 4.61 | 9.01 | 19.72 | 6.90 | 16.35 | 16.25 | 32.18 | 23.73 | 48.65 |
| REPLM (params adj) | 4.32 | 58.97 | 29.06 | 27.35 | 9.35 | 15.24 | 9.52 | 13.30 | 37.91 | 32.18 | 23.08 | 44.44 |

Table 42: Full results in comparison against external knowledge (Section 6.2). Shown are F1 scores on each relation. (Part 6 / 8)

| Method | P463 | P488 | P495 | P527 | P551 | P569 | P570 | P571 | P576 | P577 | P580 | P582 |
|---|---|---|---|---|---|---|---|---|---|---|---|---|
| REBEL | 32.03 | 15.38 | 3.47 | 29.70 | 7.14 | 50.08 | 38.79 | 44.90 | 0.00 | 34.01 | 19.51 | 0.00 |
| REBEL-sent | 24.58 | 5.88 | 4.50 | 16.55 | 10.81 | 50.82 | 43.56 | 27.59 | 0.00 | 39.40 | 14.71 | 0.00 |
| REPLM | 9.33 | 0.00 | 12.52 | 8.43 | 5.56 | 57.76 | 38.81 | 29.67 | 0.97 | 21.96 | 6.90 | 0.86 |
| REPLM (params adj) | 15.92 | 16.95 | 19.80 | 20.00 | 5.00 | 62.15 | 46.31 | 31.20 | 4.00 | 42.14 | 7.48 | 5.61 |

Table 43: Full results in comparison against external knowledge (Section 6.2). Shown are F1 scores on each relation. (Part 7 / 8)

| Method | P585 | P607 | P674 | P676 | P706 | P710 | P737 | P740 | P749 | P800 | P807 | P840 |
|---|---|---|---|---|---|---|---|---|---|---|---|---|
| REBEL | 5.50 | 38.78 | 25.64 | 0.00 | 11.43 | 16.53 | 0.00 | 11.76 | 5.26 | 29.31 | 0.00 | 0.00 |
| REBEL-sent | 3.35 | 34.36 | 19.86 | 0.00 | 8.62 | 13.79 | 0.00 | 12.90 | 14.77 | 24.71 | 0.00 | 0.00 |
| REPLM | 21.43 | 26.14 | 18.02 | 3.62 | 7.62 | 2.55 | 0.00 | 25.93 | 0.00 | 8.89 | 2.98 | 3.95 |
| REPLM (params adj) | 20.00 | 24.86 | 15.24 | 14.71 | 7.62 | 26.67 | 6.67 | 27.40 | 6.32 | 17.35 | 15.24 | 4.94 |

Table 44: Full results in comparison against external knowledge (Section 6.2). Shown are F1 scores on each relation. (Part 8 / 8)

| Method | P937 | P1001 | P1056 | P1198 | P1336 | P1344 | P1365 | P1366 | P1376 | P1412 | P1441 | P3373 |
|---|---|---|---|---|---|---|---|---|---|---|---|---|
| REBEL | 0.00 | 11.03 | 0.00 | 0.00 | 0.00 | 0.00 | 6.25 | 3.70 | 7.06 | 0.00 | 28.07 | 39.58 |
| REBEL-sent | 7.06 | 3.37 | 0.00 | 0.00 | 0.00 | 0.00 | 10.81 | 4.12 | 4.61 | 0.00 | 19.29 | 32.93 |
| REPLM | 4.42 | 12.56 | 0.00 | 0.00 | 0.00 | 49.52 | 0.00 | 0.00 | 36.36 | 34.22 | 15.71 | 7.41 |
| REPLM (params adj) | 12.35 | 17.83 | 0.00 | 1.19 | 13.95 | 47.27 | 0.00 | 0.00 | 37.85 | 12.03 | 15.83 | 8.78 |

## J  SENSITIVITY ANALYSIS OF OUR REPLM FRAMEWORK

In our original REPLM framework, we used a fixed temperature $\tau$ and a fixed probability threshold $\theta$ as our REPLM does not require a human-annotated training documents to tune the hyperparameters. Tables 45 to 50 show the sensitivity of REPLM on various hyperparameter selections for all relation types individually. Overall, we observe that our REPLM framework is robust to different hyperparameter: the performance remains at the same level for most variations. (Only exception is $\theta = 0.5$, where such high probability threshold results in discarding the correctly extracted relations. This leads to much lower recall, and therefore, F1 score.) This also explains why the performance gap between REPLM and REPLM (params adj) is small.

Table 45: Sensitivity of our REPLM framework predictions w.r.t $\tau$ and $\theta$. Shown are F1 scores on each relation. (Part 1 / 6)

| $\tau$ | $\theta$ | P6 | P17 | P19 | P20 | P22 | P25 | P26 | P27 | P30 | P31 | P35 | P36 | P37 | P39 | P40 | P50 |
|---|---|---|---|---|---|---|---|---|---|---|---|---|---|---|---|---|---|
| 0.01 | 0.01 | 33.11 | 25.88 | 57.28 | 41.05 | 16.11 | 22.58 | 35.96 | 29.08 | 36.69 | 11.51 | 30.88 | 45.95 | 22.22 | 8.70 | 17.87 | 40.00 |
| 0.01 | 0.05 | 34.07 | 26.13 | 60.31 | 43.43 | 17.30 | 20.83 | 36.89 | 30.39 | 38.43 | 12.61 | 29.03 | 51.61 | 27.37 | 9.52 | 16.22 | 42.05 |
| 0.01 | 0.10 | 36.67 | 25.36 | 64.09 | 47.50 | 12.82 | 14.63 | 36.27 | 31.77 | 32.69 | 12.77 | 32.73 | 53.33 | 30.95 | 0.00 | 16.67 | 38.67 |
| 0.01 | 0.20 | 40.40 | 24.55 | 71.03 | 54.96 | 12.60 | 20.00 | 38.32 | 32.39 | 31.46 | 14.63 | 34.69 | 48.00 | 27.40 | 0.00 | 16.18 | 37.50 |
| 0.01 | 0.50 | 24.62 | 17.25 | 76.26 | 57.78 | 14.12 | 10.00 | 35.48 | 25.56 | 26.49 | 12.70 | 30.14 | 42.86 | 10.91 | 0.00 | 17.09 | 33.57 |
| 0.05 | 0.01 | 31.06 | 25.78 | 55.94 | 40.20 | 17.65 | 22.54 | 36.89 | 28.29 | 33.87 | 10.39 | 29.17 | 45.00 | 21.85 | 7.14 | 17.09 | 41.41 |
| 0.05 | 0.05 | 31.25 | 26.86 | 56.34 | 40.20 | 17.94 | 22.95 | 36.51 | 28.81 | 38.20 | 11.94 | 29.37 | 46.58 | 25.00 | 8.33 | 17.65 | 40.00 |
| 0.05 | 0.10 | 35.56 | 27.63 | 60.10 | 41.05 | 15.54 | 28.00 | 38.01 | 30.75 | 39.46 | 14.16 | 33.33 | 52.46 | 27.91 | 9.09 | 17.83 | 41.18 |
| 0.05 | 0.20 | 32.65 | 26.12 | 69.91 | 52.24 | 20.80 | 16.67 | 40.24 | 34.45 | 34.09 | 15.58 | 36.00 | 52.83 | 29.73 | 12.50 | 17.75 | 41.98 |
| 0.05 | 0.50 | 16.95 | 13.68 | 77.94 | 57.47 | 13.33 | 11.11 | 31.78 | 24.81 | 22.07 | 7.02 | 31.43 | 46.15 | 11.32 | 0.00 | 12.24 | 33.59 |
| 0.10 | 0.01 | 31.06 | 25.75 | 55.94 | 40.20 | 17.65 | 22.54 | 36.89 | 28.29 | 33.65 | 10.32 | 29.17 | 45.00 | 21.85 | 7.14 | 17.03 | 41.41 |
| 0.10 | 0.05 | 31.25 | 26.85 | 56.34 | 40.20 | 17.47 | 22.22 | 37.04 | 28.68 | 36.36 | 11.76 | 29.58 | 45.95 | 25.24 | 8.00 | 17.59 | 41.78 |
| 0.10 | 0.10 | 32.99 | 26.16 | 70.34 | 53.03 | 19.83 | 16.67 | 40.25 | 34.65 | 34.48 | 15.79 | 35.05 | 52.83 | 29.73 | 12.50 | 18.63 | 41.25 |
| 0.10 | 0.20 | 34.78 | 27.74 | 59.06 | 40.41 | 17.17 | 28.00 | 37.17 | 30.71 | 39.11 | 14.29 | 32.56 | 52.46 | 27.91 | 8.70 | 17.56 | 41.12 |
| 0.10 | 0.50 | 16.95 | 13.63 | 77.49 | 58.14 | 13.33 | 11.11 | 32.08 | 24.84 | 22.07 | 7.02 | 30.99 | 46.15 | 11.32 | 0.00 | 10.31 | 33.59 |
| 0.20 | 0.01 | 31.06 | 25.72 | 55.94 | 40.20 | 17.65 | 22.54 | 36.89 | 28.29 | 33.65 | 10.32 | 29.17 | 45.00 | 21.85 | 7.14 | 17.03 | 41.41 |
| 0.20 | 0.05 | 31.25 | 26.78 | 56.21 | 40.20 | 18.26 | 22.22 | 37.19 | 28.66 | 34.91 | 11.76 | 29.58 | 45.33 | 25.24 | 8.00 | 17.53 | 41.59 |
| 0.20 | 0.10 | 34.53 | 27.68 | 59.20 | 40.41 | 18.00 | 28.00 | 37.00 | 30.56 | 38.94 | 14.29 | 31.82 | 51.61 | 28.24 | 9.09 | 17.16 | 43.52 |
| 0.20 | 0.20 | 33.33 | 26.39 | 70.99 | 53.12 | 18.64 | 11.43 | 41.56 | 34.56 | 34.48 | 16.00 | 35.42 | 52.83 | 30.14 | 13.33 | 20.13 | 41.51 |
| 0.20 | 0.50 | 20.00 | 13.71 | 77.49 | 58.14 | 13.33 | 11.11 | 33.64 | 24.86 | 20.83 | 7.02 | 30.99 | 46.15 | 11.32 | 0.00 | 10.20 | 33.59 |
| 0.50 | 0.01 | 31.06 | 25.73 | 55.94 | 40.20 | 17.65 | 22.54 | 36.89 | 28.28 | 33.65 | 10.32 | 29.17 | 45.00 | 21.85 | 7.14 | 17.03 | 41.41 |
| 0.50 | 0.05 | 31.25 | 26.73 | 56.21 | 40.20 | 18.26 | 22.58 | 37.19 | 28.65 | 34.78 | 11.76 | 29.58 | 45.95 | 25.49 | 8.00 | 17.53 | 41.59 |
| 0.50 | 0.10 | 34.04 | 27.77 | 59.35 | 40.21 | 17.91 | 28.00 | 37.00 | 30.54 | 38.94 | 14.04 | 31.58 | 51.61 | 27.91 | 9.09 | 17.23 | 43.12 |
| 0.50 | 0.20 | 29.79 | 26.23 | 70.99 | 53.97 | 18.49 | 11.43 | 39.74 | 34.55 | 33.33 | 15.79 | 35.42 | 52.83 | 30.14 | 13.33 | 20.25 | 41.77 |
| 0.50 | 0.50 | 20.00 | 13.96 | 77.49 | 59.77 | 13.33 | 11.11 | 33.64 | 24.98 | 20.83 | 7.02 | 23.19 | 46.15 | 11.32 | 0.00 | 10.20 | 33.59 |
| 1.00 | 0.01 | 31.06 | 25.73 | 55.94 | 40.20 | 17.65 | 22.54 | 36.89 | 28.28 | 33.65 | 10.32 | 29.17 | 45.00 | 21.85 | 7.14 | 17.03 | 41.41 |
| 1.00 | 0.05 | 31.25 | 26.73 | 56.21 | 40.20 | 18.26 | 22.22 | 37.19 | 28.65 | 34.91 | 11.76 | 29.58 | 45.95 | 25.24 | 8.00 | 17.53 | 41.59 |
| 1.00 | 0.10 | 33.09 | 27.85 | 59.35 | 40.21 | 18.72 | 28.00 | 37.00 | 30.54 | 38.77 | 14.04 | 31.34 | 51.61 | 27.91 | 9.09 | 17.10 | 43.12 |
| 1.00 | 0.20 | 29.79 | 26.18 | 71.21 | 53.97 | 18.80 | 11.43 | 39.74 | 34.64 | 33.33 | 15.79 | 35.42 | 52.83 | 30.56 | 13.33 | 20.38 | 41.77 |
| 1.00 | 0.50 | 20.00 | 13.97 | 77.49 | 59.77 | 13.33 | 11.11 | 32.08 | 24.81 | 20.83 | 7.02 | 23.19 | 46.15 | 11.32 | 0.00 | 10.20 | 33.59 |

Table 46: Sensitivity of our REPLM framework predictions w.r.t $\tau$ and $\theta$. Shown are F1 scores on each relation. (Part 2 / 6)

| $\tau$ | $\theta$ | P54 | P57 | P58 | P69 | P86 | P102 | P108 | P112 | P118 | P123 | P127 | P131 | P136 | P137 | P140 | P150 |
|---|---|---|---|---|---|---|---|---|---|---|---|---|---|---|---|---|---|
| 0.01 | 0.01 | 44.71 | 42.55 | 30.00 | 56.39 | 52.54 | 39.43 | 29.73 | 30.14 | 35.16 | 32.47 | 19.28 | 29.07 | 21.05 | 15.38 | 16.17 | 32.25 |
| 0.01 | 0.05 | 45.26 | 43.17 | 31.58 | 60.29 | 50.45 | 38.71 | 31.34 | 28.57 | 39.47 | 32.17 | 20.13 | 28.64 | 25.81 | 17.95 | 15.03 | 32.92 |
| 0.01 | 0.10 | 42.81 | 43.48 | 33.33 | 58.59 | 49.06 | 42.92 | 31.15 | 33.33 | 40.88 | 31.34 | 20.00 | 27.52 | 20.69 | 13.51 | 13.70 | 34.45 |
| 0.01 | 0.20 | 39.86 | 45.41 | 35.82 | 59.67 | 39.58 | 45.60 | 31.48 | 35.56 | 42.28 | 29.75 | 19.67 | 24.54 | 17.39 | 16.13 | 13.79 | 33.10 |
| 0.01 | 0.50 | 41.53 | 47.20 | 37.93 | 61.44 | 19.18 | 41.77 | 27.78 | 36.84 | 37.65 | 30.43 | 18.56 | 15.82 | 11.11 | 8.16 | 6.38 | 24.12 |
| 0.05 | 0.01 | 43.06 | 42.28 | 29.63 | 56.77 | 52.54 | 39.04 | 30.07 | 26.83 | 34.97 | 32.34 | 19.69 | 28.97 | 24.39 | 14.00 | 15.44 | 32.13 |
| 0.05 | 0.05 | 44.06 | 42.98 | 29.63 | 56.77 | 52.54 | 39.72 | 30.46 | 29.33 | 35.96 | 31.90 | 20.11 | 29.35 | 27.03 | 14.58 | 15.02 | 32.86 |
| 0.05 | 0.10 | 45.45 | 44.44 | 31.58 | 55.86 | 51.79 | 41.38 | 31.65 | 31.25 | 40.25 | 32.65 | 22.08 | 30.02 | 18.75 | 17.95 | 14.55 | 34.27 |
| 0.05 | 0.20 | 43.42 | 47.42 | 35.82 | 64.41 | 38.71 | 46.07 | 34.00 | 40.91 | 45.90 | 35.20 | 20.87 | 26.60 | 16.00 | 17.54 | 13.45 | 33.77 |
| 0.05 | 0.50 | 35.71 | 48.05 | 38.60 | 60.00 | 17.14 | 35.71 | 25.71 | 24.24 | 37.04 | 30.23 | 17.78 | 13.51 | 12.50 | 4.35 | 4.55 | 20.25 |
| 0.10 | 0.01 | 43.06 | 42.28 | 29.63 | 56.77 | 52.54 | 39.04 | 30.07 | 26.51 | 34.97 | 32.34 | 19.59 | 28.95 | 24.39 | 14.00 | 15.50 | 32.11 |
| 0.10 | 0.05 | 43.68 | 42.98 | 29.63 | 56.77 | 52.54 | 39.86 | 30.46 | 28.57 | 35.96 | 32.73 | 20.65 | 29.39 | 27.03 | 14.43 | 14.88 | 32.72 |
| 0.10 | 0.10 | 43.42 | 47.12 | 36.36 | 64.74 | 38.71 | 45.90 | 35.42 | 37.21 | 46.28 | 35.77 | 22.41 | 26.68 | 16.67 | 17.24 | 13.68 | 33.74 |
| 0.10 | 0.20 | 45.78 | 43.70 | 30.77 | 56.25 | 51.33 | 42.26 | 31.88 | 31.43 | 39.75 | 32.47 | 21.52 | 29.89 | 18.18 | 17.72 | 14.29 | 34.36 |
| 0.10 | 0.50 | 36.28 | 47.06 | 39.29 | 59.06 | 17.14 | 36.23 | 25.71 | 24.24 | 37.04 | 30.59 | 19.78 | 13.55 | 12.50 | 4.35 | 4.60 | 20.08 |
| 0.20 | 0.01 | 43.06 | 42.28 | 29.63 | 56.77 | 52.54 | 39.04 | 30.07 | 26.51 | 34.97 | 32.34 | 19.59 | 28.95 | 24.39 | 14.00 | 15.44 | 32.11 |
| 0.20 | 0.05 | 43.68 | 42.80 | 29.63 | 56.77 | 52.54 | 39.86 | 30.46 | 28.57 | 35.96 | 32.53 | 20.54 | 29.27 | 27.03 | 14.58 | 14.75 | 32.77 |
| 0.20 | 0.10 | 45.78 | 43.51 | 30.38 | 56.25 | 49.12 | 42.38 | 32.17 | 31.43 | 39.75 | 33.99 | 21.79 | 30.07 | 18.75 | 17.72 | 13.25 | 34.15 |
| 0.20 | 0.20 | 43.42 | 45.74 | 36.92 | 65.12 | 35.16 | 43.43 | 36.56 | 37.21 | 47.06 | 37.61 | 22.61 | 26.65 | 17.39 | 17.24 | 13.68 | 34.10 |
| 0.20 | 0.50 | 35.96 | 47.06 | 40.74 | 58.11 | 17.14 | 36.50 | 25.71 | 24.24 | 39.02 | 30.59 | 20.00 | 13.31 | 12.50 | 4.44 | 4.60 | 19.58 |
| 0.50 | 0.01 | 43.06 | 42.28 | 29.63 | 56.77 | 52.54 | 39.04 | 30.07 | 26.51 | 34.97 | 32.34 | 19.59 | 28.95 | 24.39 | 14.00 | 15.44 | 32.11 |
| 0.50 | 0.05 | 43.68 | 42.62 | 29.63 | 56.77 | 52.54 | 39.58 | 30.46 | 28.57 | 35.96 | 32.53 | 20.54 | 29.29 | 27.03 | 14.58 | 14.75 | 32.73 |
| 0.50 | 0.10 | 45.65 | 43.15 | 30.38 | 56.50 | 47.79 | 42.07 | 32.17 | 31.43 | 39.51 | 33.99 | 21.79 | 29.89 | 24.24 | 17.50 | 13.10 | 33.90 |
| 0.50 | 0.20 | 44.04 | 45.99 | 36.92 | 65.12 | 35.56 | 43.43 | 34.78 | 37.21 | 47.46 | 36.84 | 23.01 | 26.47 | 16.67 | 17.54 | 13.56 | 34.02 |
| 0.50 | 0.50 | 35.96 | 48.05 | 40.74 | 58.11 | 17.14 | 36.50 | 25.71 | 29.41 | 39.02 | 30.59 | 20.00 | 13.32 | 12.50 | 4.44 | 4.60 | 19.89 |
| 1.00 | 0.01 | 43.06 | 42.28 | 29.63 | 56.77 | 52.54 | 39.04 | 30.07 | 26.19 | 34.97 | 32.34 | 19.59 | 28.95 | 24.39 | 14.00 | 15.44 | 32.11 |
| 1.00 | 0.05 | 43.68 | 42.62 | 29.63 | 56.77 | 52.54 | 39.58 | 30.46 | 28.57 | 35.96 | 32.53 | 20.54 | 29.30 | 27.03 | 14.58 | 14.61 | 32.71 |
| 1.00 | 0.10 | 45.51 | 43.15 | 30.38 | 57.14 | 47.79 | 42.07 | 32.17 | 31.43 | 39.26 | 33.99 | 21.79 | 29.79 | 24.24 | 17.50 | 13.10 | 33.92 |
| 1.00 | 0.20 | 44.20 | 45.99 | 36.92 | 65.50 | 35.56 | 42.53 | 34.78 | 37.21 | 47.46 | 36.84 | 23.21 | 26.37 | 16.67 | 17.54 | 13.56 | 33.75 |
| 1.00 | 0.50 | 35.96 | 48.05 | 40.74 | 58.11 | 17.14 | 35.29 | 25.71 | 29.41 | 39.02 | 30.59 | 20.00 | 13.32 | 12.50 | 8.70 | 4.60 | 19.89 |

Table 47: Sensitivity of our REPLM framework predictions w.r.t $\tau$ and $\theta$. Shown are F1 scores on each relation. (Part 3 / 6)

| $\tau$ | $\theta$ | P155 | P156 | P159 | P161 | P162 | P166 | P170 | P171 | P172 | P175 | P176 | P178 | P179 | P190 | P194 | P205 |
|---|---|---|---|---|---|---|---|---|---|---|---|---|---|---|---|---|---|
| 0.01 | 0.01 | 15.84 | 26.44 | 41.58 | 45.04 | 24.06 | 27.50 | 16.81 | 14.63 | 31.37 | 50.90 | 27.03 | 32.51 | 25.00 | 66.67 | 26.21 | 23.38 |
| 0.01 | 0.05 | 15.57 | 28.17 | 44.09 | 46.60 | 24.59 | 27.40 | 17.48 | 10.81 | 35.44 | 51.75 | 28.57 | 32.46 | 27.64 | 66.67 | 27.87 | 24.62 |
| 0.01 | 0.10 | 17.39 | 26.23 | 43.68 | 45.96 | 23.64 | 27.48 | 14.74 | 11.43 | 37.68 | 52.24 | 28.99 | 31.82 | 26.55 | 66.67 | 25.69 | 27.12 |
| 0.01 | 0.20 | 13.79 | 27.08 | 43.87 | 43.17 | 16.28 | 30.36 | 13.33 | 12.12 | 30.51 | 50.17 | 25.00 | 32.26 | 26.42 | 66.67 | 24.74 | 24.49 |
| 0.01 | 0.50 | 14.12 | 24.24 | 35.59 | 35.53 | 8.96 | 30.77 | 11.76 | 14.81 | 26.09 | 43.15 | 9.09 | 19.64 | 23.26 | 66.67 | 17.39 | 5.13 |
| 0.05 | 0.01 | 14.29 | 25.26 | 39.09 | 43.73 | 24.64 | 26.51 | 17.14 | 14.63 | 28.32 | 49.94 | 28.57 | 31.78 | 24.83 | 66.67 | 24.20 | 23.53 |
| 0.05 | 0.05 | 14.29 | 26.82 | 40.76 | 44.18 | 24.82 | 27.16 | 18.18 | 14.63 | 31.58 | 50.86 | 28.95 | 32.23 | 25.00 | 66.67 | 25.50 | 25.32 |
| 0.05 | 0.10 | 16.47 | 31.21 | 46.24 | 46.21 | 24.00 | 27.27 | 14.55 | 14.63 | 32.91 | 51.99 | 28.17 | 31.02 | 26.98 | 66.67 | 29.27 | 23.88 |
| 0.05 | 0.20 | 11.97 | 27.66 | 44.30 | 47.86 | 26.37 | 34.23 | 17.72 | 11.43 | 30.77 | 53.27 | 24.89 | 32.89 | 22.43 | 66.67 | 21.51 | 16.00 |
| 0.05 | 0.50 | 14.81 | 22.58 | 35.09 | 36.80 | 9.38 | 27.59 | 8.16 | 15.38 | 15.00 | 39.39 | 4.76 | 18.37 | 20.25 | 66.67 | 12.70 | 5.26 |
| 0.10 | 0.01 | 14.29 | 25.26 | 39.09 | 43.73 | 24.64 | 26.51 | 17.14 | 14.63 | 28.32 | 49.94 | 28.57 | 31.78 | 24.83 | 66.67 | 24.20 | 23.53 |
| 0.10 | 0.05 | 14.08 | 26.52 | 40.57 | 44.03 | 24.82 | 27.16 | 17.52 | 14.63 | 30.30 | 50.55 | 28.95 | 31.92 | 24.83 | 66.67 | 25.50 | 25.00 |
| 0.10 | 0.10 | 12.28 | 27.37 | 42.31 | 47.95 | 27.59 | 34.55 | 17.50 | 11.43 | 29.63 | 53.00 | 31.58 | 32.43 | 23.30 | 66.67 | 23.91 | 16.33 |
| 0.10 | 0.20 | 16.09 | 29.93 | 45.03 | 45.23 | 23.81 | 26.42 | 15.00 | 14.63 | 32.91 | 51.91 | 28.17 | 31.75 | 27.27 | 66.67 | 30.65 | 25.35 |
| 0.10 | 0.50 | 14.81 | 19.67 | 37.93 | 36.90 | 9.38 | 27.59 | 12.50 | 16.00 | 10.26 | 38.94 | 9.30 | 19.57 | 20.00 | 66.67 | 15.62 | 5.13 |
| 0.20 | 0.01 | 14.29 | 25.26 | 39.09 | 43.73 | 24.64 | 26.35 | 17.14 | 14.63 | 28.32 | 49.94 | 28.57 | 31.78 | 24.83 | 66.67 | 24.20 | 23.53 |
| 0.20 | 0.05 | 13.95 | 26.52 | 40.57 | 44.03 | 24.82 | 27.16 | 17.52 | 14.63 | 30.30 | 50.61 | 28.95 | 31.92 | 24.83 | 66.67 | 25.50 | 25.00 |
| 0.20 | 0.10 | 15.56 | 30.07 | 44.33 | 45.23 | 23.62 | 26.42 | 15.00 | 14.63 | 32.50 | 51.91 | 30.14 | 32.46 | 27.07 | 66.67 | 30.40 | 25.35 |
| 0.20 | 0.20 | 12.28 | 28.57 | 41.83 | 47.71 | 30.23 | 33.64 | 17.07 | 11.76 | 26.92 | 52.88 | 32.14 | 30.99 | 22.00 | 66.67 | 21.98 | 16.00 |
| 0.20 | 0.50 | 14.63 | 20.00 | 36.52 | 35.82 | 9.38 | 27.59 | 12.50 | 16.00 | 10.26 | 38.22 | 9.30 | 19.35 | 20.00 | 66.67 | 15.62 | 5.13 |
| 0.50 | 0.01 | 14.29 | 25.26 | 39.09 | 43.73 | 24.64 | 26.35 | 17.14 | 14.63 | 28.32 | 49.94 | 28.57 | 31.78 | 24.83 | 66.67 | 24.20 | 23.53 |
| 0.50 | 0.05 | 13.95 | 26.52 | 40.57 | 44.03 | 24.82 | 26.99 | 17.52 | 14.63 | 30.30 | 50.49 | 28.95 | 31.78 | 24.83 | 66.67 | 25.68 | 24.69 |
| 0.50 | 0.10 | 15.38 | 30.07 | 43.88 | 45.23 | 23.81 | 27.50 | 14.75 | 14.63 | 32.10 | 51.84 | 29.73 | 32.12 | 27.27 | 66.67 | 30.16 | 25.35 |
| 0.50 | 0.20 | 12.28 | 28.89 | 42.11 | 48.37 | 30.23 | 32.08 | 17.50 | 11.76 | 26.92 | 52.76 | 32.14 | 31.88 | 20.62 | 66.67 | 21.98 | 12.24 |
| 0.50 | 0.50 | 14.63 | 20.00 | 36.52 | 35.71 | 9.38 | 27.59 | 13.04 | 16.00 | 10.26 | 37.58 | 9.30 | 19.35 | 20.00 | 66.67 | 15.62 | 5.13 |
| 1.00 | 0.01 | 14.29 | 25.26 | 39.09 | 43.73 | 24.64 | 26.35 | 17.14 | 14.63 | 28.32 | 49.94 | 28.57 | 31.78 | 24.83 | 66.67 | 24.20 | 23.53 |
| 1.00 | 0.05 | 13.95 | 26.52 | 40.76 | 44.10 | 24.82 | 26.99 | 17.52 | 14.63 | 30.30 | 50.49 | 28.95 | 31.78 | 24.83 | 66.67 | 25.68 | 24.69 |
| 1.00 | 0.10 | 15.47 | 30.97 | 44.10 | 45.23 | 23.81 | 27.50 | 14.52 | 14.63 | 32.50 | 51.78 | 29.73 | 31.96 | 27.27 | 66.67 | 30.16 | 25.00 |
| 1.00 | 0.20 | 12.28 | 28.89 | 41.83 | 48.26 | 30.23 | 32.08 | 17.50 | 11.76 | 26.92 | 52.94 | 32.14 | 31.88 | 20.83 | 66.67 | 21.98 | 12.24 |
| 1.00 | 0.50 | 14.81 | 20.00 | 36.52 | 36.20 | 9.38 | 31.11 | 12.77 | 16.00 | 10.26 | 37.58 | 9.30 | 19.35 | 20.00 | 66.67 | 15.62 | 5.13 |

Table 48: Sensitivity of our REPLM framework predictions w.r.t $\tau$ and $\theta$. Shown are F1 scores on each relation. (Part 4 / 6)

| $\tau$ | $\theta$ | P206 | P241 | P264 | P272 | P276 | P279 | P355 | P361 | P364 | P400 | P403 | P449 | P463 | P488 | P495 | P527 |
|---|---|---|---|---|---|---|---|---|---|---|---|---|---|---|---|---|---|
| 0.01 | 0.01 | 19.78 | 41.90 | 43.53 | 37.04 | 26.60 | 11.11 | 24.66 | 22.50 | 34.29 | 32.26 | 28.57 | 33.71 | 38.26 | 14.29 | 21.37 | 20.21 |
| 0.01 | 0.05 | 19.74 | 42.86 | 43.43 | 35.62 | 27.16 | 8.57 | 28.07 | 25.66 | 36.36 | 33.33 | 28.95 | 34.88 | 39.68 | 16.33 | 22.54 | 21.41 |
| 0.01 | 0.10 | 16.78 | 45.45 | 43.21 | 37.14 | 29.58 | 6.90 | 26.92 | 26.53 | 28.07 | 32.00 | 23.53 | 31.71 | 38.94 | 13.95 | 21.82 | 21.97 |
| 0.01 | 0.20 | 11.57 | 48.19 | 39.71 | 37.29 | 29.03 | 8.33 | 26.67 | 26.38 | 23.53 | 35.09 | 26.23 | 33.77 | 38.38 | 16.22 | 23.73 | 20.88 |
| 0.01 | 0.50 | 4.12 | 47.62 | 34.12 | 34.62 | 22.45 | 9.52 | 27.03 | 21.05 | 22.22 | 27.08 | 20.41 | 33.33 | 34.62 | 19.35 | 18.57 | 21.19 |
| 0.05 | 0.01 | 19.79 | 44.04 | 44.10 | 37.65 | 23.58 | 12.61 | 20.00 | 21.53 | 33.80 | 32.05 | 26.97 | 35.16 | 35.29 | 12.90 | 21.68 | 17.44 |
| 0.05 | 0.05 | 21.11 | 42.99 | 44.33 | 36.59 | 25.71 | 13.48 | 22.22 | 22.80 | 34.29 | 32.26 | 27.27 | 35.56 | 37.62 | 13.33 | 21.76 | 19.68 |
| 0.05 | 0.10 | 20.38 | 45.36 | 45.10 | 36.36 | 30.12 | 9.38 | 26.67 | 25.23 | 38.10 | 35.21 | 27.40 | 36.14 | 43.02 | 14.81 | 21.75 | 22.17 |
| 0.05 | 0.20 | 17.24 | 52.63 | 44.66 | 40.68 | 35.48 | 8.51 | 23.81 | 30.72 | 23.53 | 34.19 | 27.12 | 36.11 | 43.52 | 20.00 | 23.76 | 23.49 |
| 0.05 | 0.50 | 6.45 | 53.12 | 33.44 | 36.73 | 20.45 | 0.00 | 18.18 | 20.54 | 22.22 | 29.21 | 21.28 | 33.33 | 31.72 | 15.38 | 22.99 | 18.96 |
| 0.10 | 0.01 | 19.79 | 44.04 | 44.10 | 37.65 | 23.58 | 12.61 | 20.00 | 21.43 | 33.80 | 32.05 | 26.97 | 35.16 | 35.29 | 12.90 | 21.68 | 17.73 |
| 0.10 | 0.05 | 20.77 | 44.04 | 44.41 | 36.14 | 25.00 | 13.64 | 21.43 | 22.50 | 34.29 | 32.26 | 27.27 | 35.16 | 36.77 | 13.11 | 21.96 | 19.68 |
| 0.10 | 0.10 | 17.09 | 52.63 | 45.93 | 42.11 | 32.79 | 8.89 | 23.26 | 30.19 | 24.00 | 35.40 | 28.07 | 39.39 | 42.71 | 20.00 | 24.30 | 24.66 |
| 0.10 | 0.20 | 20.51 | 46.00 | 45.54 | 36.36 | 30.12 | 9.09 | 22.22 | 25.39 | 39.34 | 34.01 | 27.03 | 38.10 | 42.07 | 14.81 | 22.41 | 21.74 |
| 0.10 | 0.50 | 4.35 | 53.97 | 33.54 | 37.50 | 20.69 | 0.00 | 18.18 | 20.54 | 22.22 | 29.55 | 21.28 | 33.33 | 32.39 | 8.33 | 22.22 | 18.54 |
| 0.20 | 0.01 | 19.79 | 44.04 | 44.10 | 37.65 | 23.58 | 12.61 | 20.00 | 21.39 | 33.80 | 32.05 | 26.97 | 35.16 | 35.29 | 12.90 | 21.68 | 17.73 |
| 0.20 | 0.05 | 20.88 | 44.04 | 44.41 | 35.71 | 25.12 | 13.64 | 21.43 | 21.98 | 34.78 | 32.26 | 27.27 | 35.16 | 36.77 | 12.90 | 22.06 | 19.56 |
| 0.20 | 0.10 | 19.74 | 45.54 | 45.54 | 35.90 | 29.94 | 8.82 | 22.22 | 25.11 | 39.34 | 33.56 | 29.33 | 37.65 | 41.18 | 15.09 | 22.67 | 21.85 |
| 0.20 | 0.20 | 17.24 | 52.63 | 46.75 | 42.11 | 32.52 | 0.00 | 23.81 | 29.30 | 24.00 | 37.04 | 28.07 | 37.50 | 41.88 | 15.79 | 24.68 | 24.74 |
| 0.20 | 0.50 | 4.35 | 53.97 | 32.81 | 37.50 | 20.69 | 0.00 | 18.18 | 20.54 | 22.22 | 25.58 | 21.28 | 33.33 | 32.39 | 8.33 | 22.05 | 18.45 |
| 0.50 | 0.01 | 19.79 | 44.04 | 44.10 | 37.65 | 23.58 | 12.73 | 20.00 | 21.43 | 33.80 | 32.05 | 26.97 | 35.16 | 35.29 | 12.90 | 21.68 | 17.73 |
| 0.50 | 0.05 | 20.88 | 44.04 | 44.41 | 35.71 | 25.12 | 13.64 | 21.43 | 21.66 | 34.78 | 32.26 | 27.27 | 35.16 | 36.77 | 12.90 | 22.06 | 19.60 |
| 0.50 | 0.10 | 19.74 | 45.54 | 45.37 | 35.90 | 30.49 | 8.96 | 21.54 | 25.17 | 39.34 | 33.33 | 28.95 | 38.10 | 41.33 | 15.09 | 22.67 | 21.80 |
| 0.50 | 0.20 | 17.24 | 52.63 | 46.63 | 42.11 | 32.52 | 0.00 | 23.81 | 29.39 | 24.49 | 37.38 | 28.07 | 38.10 | 41.05 | 15.79 | 24.68 | 24.74 |
| 0.50 | 0.50 | 4.35 | 53.97 | 31.85 | 40.82 | 20.69 | 0.00 | 18.18 | 21.33 | 22.22 | 23.53 | 21.28 | 33.33 | 32.39 | 7.69 | 22.05 | 18.54 |
| 1.00 | 0.01 | 19.79 | 44.04 | 44.10 | 37.65 | 23.58 | 12.73 | 20.00 | 21.43 | 33.80 | 32.05 | 26.97 | 35.16 | 35.29 | 12.90 | 21.68 | 17.73 |
| 1.00 | 0.05 | 20.88 | 44.04 | 44.41 | 35.71 | 25.12 | 13.79 | 21.43 | 21.66 | 34.78 | 32.26 | 27.27 | 35.16 | 36.77 | 12.90 | 22.06 | 19.64 |
| 1.00 | 0.10 | 20.00 | 45.54 | 45.45 | 35.90 | 30.67 | 8.96 | 21.21 | 25.33 | 39.34 | 32.89 | 28.95 | 38.10 | 41.48 | 14.81 | 22.63 | 21.70 |
| 1.00 | 0.20 | 17.24 | 52.63 | 46.75 | 42.11 | 32.26 | 0.00 | 23.81 | 28.85 | 24.49 | 37.74 | 28.07 | 38.10 | 42.71 | 15.79 | 24.81 | 24.91 |
| 1.00 | 0.50 | 4.35 | 53.97 | 31.85 | 40.82 | 20.69 | 0.00 | 18.18 | 21.33 | 22.22 | 23.53 | 21.28 | 33.33 | 32.39 | 7.69 | 22.05 | 18.54 |

Table 49: Sensitivity of our REPLM framework predictions w.r.t $\tau$ and $\theta$. Shown are F1 scores on each relation. (Part 5 / 6)

| $\tau$ | $\theta$ | P551 | P569 | P570 | P571 | P576 | P577 | P580 | P582 | P585 | P607 | P674 | P676 | P706 | P710 | P737 | P740 |
|---|---|---|---|---|---|---|---|---|---|---|---|---|---|---|---|---|---|
| 0.01 | 0.01 | 26.67 | 55.59 | 42.86 | 37.04 | 14.55 | 49.90 | 23.33 | 38.10 | 25.93 | 31.23 | 30.08 | 47.06 | 21.77 | 26.02 | 0.00 | 33.33 |
| 0.01 | 0.05 | 30.77 | 58.03 | 44.24 | 40.31 | 15.05 | 52.24 | 26.42 | 40.74 | 25.81 | 31.93 | 24.79 | 57.14 | 24.81 | 29.09 | 0.00 | 30.77 |
| 0.01 | 0.10 | 36.36 | 59.68 | 45.71 | 44.13 | 15.38 | 52.85 | 27.59 | 42.55 | 28.92 | 31.63 | 22.61 | 57.14 | 22.81 | 29.70 | 0.00 | 28.57 |
| 0.01 | 0.20 | 36.36 | 61.61 | 49.16 | 46.25 | 9.84 | 51.57 | 32.84 | 43.90 | 28.57 | 31.31 | 24.53 | 46.15 | 16.67 | 27.37 | 0.00 | 31.25 |
| 0.01 | 0.50 | 22.22 | 62.01 | 50.36 | 43.98 | 12.00 | 46.79 | 35.56 | 36.36 | 22.22 | 25.00 | 23.66 | 46.15 | 13.51 | 24.69 | 0.00 | 41.67 |
| 0.05 | 0.01 | 25.00 | 54.43 | 41.94 | 34.61 | 12.60 | 47.98 | 21.37 | 36.36 | 27.03 | 29.89 | 30.00 | 47.06 | 19.51 | 24.43 | 0.00 | 31.11 |
| 0.05 | 0.05 | 25.00 | 54.78 | 42.48 | 36.99 | 14.16 | 50.35 | 23.53 | 38.71 | 29.13 | 30.77 | 30.66 | 47.06 | 20.65 | 25.81 | 0.00 | 34.15 |
| 0.05 | 0.10 | 28.57 | 56.29 | 44.87 | 40.89 | 17.98 | 52.46 | 28.28 | 46.15 | 28.26 | 33.87 | 30.40 | 50.00 | 24.43 | 30.19 | 0.00 | 32.43 |
| 0.05 | 0.20 | 33.33 | 61.47 | 49.79 | 45.66 | 19.05 | 54.40 | 34.92 | 43.90 | 27.78 | 27.69 | 23.64 | 46.15 | 17.39 | 23.66 | 0.00 | 32.26 |
| 0.05 | 0.50 | 28.57 | 61.18 | 50.36 | 44.55 | 8.33 | 46.69 | 28.57 | 26.67 | 26.92 | 23.45 | 18.18 | 46.15 | 14.08 | 18.42 | 0.00 | 38.10 |
| 0.10 | 0.01 | 25.00 | 54.43 | 41.94 | 34.61 | 12.60 | 47.94 | 21.37 | 36.36 | 27.03 | 29.89 | 30.00 | 47.06 | 19.51 | 24.06 | 0.00 | 31.11 |
| 0.10 | 0.05 | 26.67 | 54.71 | 42.41 | 36.75 | 13.56 | 50.05 | 23.33 | 37.50 | 29.13 | 30.22 | 30.22 | 47.06 | 20.25 | 25.60 | 0.00 | 34.15 |
| 0.10 | 0.10 | 33.33 | 62.10 | 50.21 | 46.15 | 19.35 | 53.90 | 35.48 | 45.00 | 26.47 | 26.04 | 23.85 | 46.15 | 17.78 | 23.66 | 0.00 | 33.33 |
| 0.10 | 0.20 | 28.57 | 55.73 | 44.56 | 39.71 | 17.78 | 52.39 | 29.17 | 43.14 | 30.11 | 32.94 | 31.01 | 47.06 | 23.02 | 30.19 | 0.00 | 31.58 |
| 0.10 | 0.50 | 28.57 | 61.29 | 50.36 | 43.44 | 8.51 | 46.96 | 28.57 | 26.67 | 27.45 | 23.29 | 17.78 | 46.15 | 14.08 | 18.67 | 0.00 | 40.00 |
| 0.20 | 0.01 | 25.00 | 54.43 | 41.94 | 34.61 | 12.60 | 47.98 | 21.37 | 36.36 | 27.03 | 29.89 | 30.00 | 47.06 | 19.51 | 24.06 | 0.00 | 31.11 |
| 0.20 | 0.05 | 26.67 | 54.64 | 42.35 | 36.83 | 13.68 | 49.85 | 23.14 | 36.92 | 29.13 | 30.22 | 30.22 | 47.06 | 20.25 | 25.60 | 0.00 | 34.15 |
| 0.20 | 0.10 | 26.67 | 55.51 | 44.48 | 39.42 | 17.58 | 52.61 | 28.87 | 44.00 | 30.11 | 32.94 | 31.01 | 47.06 | 22.86 | 30.19 | 0.00 | 32.43 |
| 0.20 | 0.20 | 33.33 | 61.77 | 50.54 | 45.95 | 19.35 | 53.42 | 35.48 | 45.00 | 26.47 | 26.18 | 23.30 | 46.15 | 17.78 | 23.66 | 0.00 | 33.33 |
| 0.20 | 0.50 | 28.57 | 61.18 | 50.00 | 43.44 | 8.51 | 47.02 | 28.57 | 25.81 | 27.45 | 21.92 | 17.78 | 46.15 | 14.08 | 18.67 | 0.00 | 40.00 |
| 0.50 | 0.01 | 25.00 | 54.43 | 41.94 | 34.61 | 12.60 | 47.94 | 21.37 | 36.36 | 27.03 | 29.89 | 30.00 | 47.06 | 19.51 | 24.06 | 0.00 | 31.11 |
| 0.50 | 0.05 | 26.67 | 54.64 | 42.35 | 36.91 | 13.56 | 49.75 | 22.95 | 36.92 | 29.13 | 30.22 | 30.43 | 47.06 | 20.13 | 25.60 | 0.00 | 34.15 |
| 0.50 | 0.10 | 26.67 | 55.44 | 44.41 | 38.94 | 17.39 | 52.38 | 26.53 | 43.14 | 30.11 | 32.81 | 31.01 | 47.06 | 23.02 | 30.19 | 0.00 | 32.43 |
| 0.50 | 0.20 | 33.33 | 61.11 | 50.66 | 46.10 | 19.05 | 53.37 | 35.48 | 45.00 | 26.47 | 25.26 | 23.30 | 46.15 | 17.98 | 23.91 | 0.00 | 33.33 |
| 0.50 | 0.50 | 28.57 | 61.29 | 50.85 | 44.14 | 8.70 | 46.94 | 28.57 | 25.81 | 26.92 | 21.77 | 17.78 | 46.15 | 14.08 | 18.67 | 0.00 | 40.00 |
| 1.00 | 0.01 | 25.00 | 54.43 | 41.94 | 34.61 | 12.60 | 47.94 | 21.37 | 36.36 | 27.03 | 29.89 | 30.00 | 47.06 | 19.51 | 24.06 | 0.00 | 31.11 |
| 1.00 | 0.05 | 26.67 | 54.57 | 42.35 | 36.83 | 13.56 | 49.75 | 22.95 | 36.92 | 29.13 | 30.22 | 30.43 | 47.06 | 20.00 | 25.60 | 0.00 | 34.15 |
| 1.00 | 0.10 | 26.67 | 55.44 | 44.33 | 38.94 | 17.58 | 52.22 | 26.53 | 43.14 | 30.11 | 32.81 | 30.77 | 47.06 | 23.02 | 30.19 | 0.00 | 32.43 |
| 1.00 | 0.20 | 33.33 | 61.21 | 50.66 | 46.78 | 19.05 | 53.19 | 35.48 | 41.03 | 26.47 | 25.26 | 23.30 | 46.15 | 17.98 | 23.91 | 0.00 | 33.33 |
| 1.00 | 0.50 | 28.57 | 61.29 | 50.85 | 44.14 | 8.70 | 46.94 | 28.57 | 25.81 | 26.92 | 21.77 | 17.78 | 46.15 | 14.08 | 18.67 | 0.00 | 40.00 |

Table 50: Sensitivity of our REPLM framework predictions w.r.t $\tau$ and $\theta$. Shown are F1 scores on each relation. (Part 6 / 6)

| $\tau$ | $\theta$ | P749 | P800 | P807 | P840 | P937 | P1001 | P1056 | P1198 | P1336 | P1344 | P1365 | P1366 | P1376 | P1412 | P1441 | P3373 |
|------|------|-------|-------|-------|-------|-------|-------|------|--------|-------|-------|------|-------|-------|-------|-------|-------|
| 0.01 | 0.01 | 22.02 | 27.94 | 50.00 | 43.24 | 27.27 | 22.75 | 0.00 | 66.67 | 23.08 | 34.15 | 0.00 | 6.25 | 56.67 | 28.35 | 30.41 | 29.27 |
| 0.01 | 0.05 | 20.22 | 28.80 | 50.00 | 41.18 | 30.99 | 24.42 | 0.00 | 100.00 | 25.00 | 36.36 | 0.00 | 8.33 | 57.14 | 28.83 | 27.41 | 28.96 |
| 0.01 | 0.10 | 21.05 | 28.57 | 66.67 | 40.00 | 33.33 | 25.85 | 0.00 | 100.00 | 27.27 | 34.59 | 0.00 | 9.09 | 52.00 | 28.00 | 27.12 | 25.96 |
| 0.01 | 0.20 | 24.62 | 25.81 | 66.67 | 48.00 | 35.29 | 29.03 | 0.00 | 100.00 | 21.05 | 37.04 | 0.00 | 13.33 | 54.55 | 31.58 | 27.71 | 24.74 |
| 0.01 | 0.50 | 20.00 | 26.67 | 0.00 | 30.00 | 27.78 | 23.08 | 0.00 | 0.00 | 14.29 | 41.38 | 0.00 | 0.00 | 60.61 | 32.79 | 17.81 | 16.57 |
| 0.05 | 0.01 | 21.43 | 27.74 | 50.00 | 46.15 | 26.67 | 21.10 | 0.00 | 66.67 | 19.35 | 34.12 | 0.00 | 4.35 | 53.97 | 26.09 | 31.49 | 29.60 |
| 0.05 | 0.05 | 20.97 | 28.15 | 50.00 | 42.11 | 25.29 | 23.30 | 0.00 | 100.00 | 23.08 | 34.12 | 0.00 | 5.88 | 54.84 | 27.07 | 29.36 | 29.60 |
| 0.05 | 0.10 | 19.35 | 27.87 | 50.00 | 36.36 | 29.33 | 28.05 | 0.00 | 100.00 | 26.09 | 36.02 | 0.00 | 10.00 | 56.14 | 30.09 | 30.93 | 27.85 |
| 0.05 | 0.20 | 25.81 | 30.77 | 66.67 | 30.77 | 26.92 | 30.65 | 0.00 | 100.00 | 12.50 | 37.50 | 0.00 | 0.00 | 60.87 | 32.50 | 24.54 | 26.26 |
| 0.05 | 0.50 | 17.39 | 27.40 | 0.00 | 21.05 | 21.43 | 16.84 | 0.00 | 0.00 | 15.38 | 41.86 | 0.00 | 0.00 | 58.06 | 24.14 | 18.84 | 19.63 |
| 0.10 | 0.01 | 21.43 | 27.74 | 50.00 | 46.15 | 26.67 | 21.10 | 0.00 | 66.67 | 19.35 | 34.12 | 0.00 | 4.35 | 53.97 | 26.09 | 31.49 | 29.60 |
| 0.10 | 0.05 | 21.37 | 27.94 | 50.00 | 42.11 | 25.29 | 23.81 | 0.00 | 100.00 | 21.43 | 34.12 | 0.00 | 5.26 | 54.84 | 26.87 | 30.00 | 29.60 |
| 0.10 | 0.10 | 22.58 | 31.82 | 66.67 | 30.77 | 21.74 | 30.40 | 0.00 | 100.00 | 12.50 | 37.84 | 0.00 | 0.00 | 62.22 | 33.33 | 23.75 | 25.13 |
| 0.10 | 0.20 | 24.49 | 29.27 | 50.00 | 34.29 | 28.57 | 28.57 | 0.00 | 100.00 | 26.09 | 35.80 | 0.00 | 0.00 | 55.17 | 28.81 | 30.15 | 28.22 |
| 0.10 | 0.50 | 17.39 | 27.03 | 0.00 | 21.05 | 20.69 | 16.84 | 0.00 | 0.00 | 15.38 | 41.38 | 0.00 | 0.00 | 56.25 | 24.56 | 17.65 | 19.75 |
| 0.20 | 0.01 | 21.43 | 27.74 | 50.00 | 46.15 | 26.67 | 21.10 | 0.00 | 66.67 | 19.35 | 34.12 | 0.00 | 4.35 | 53.97 | 26.09 | 31.49 | 29.60 |
| 0.20 | 0.05 | 22.73 | 27.94 | 50.00 | 42.11 | 25.29 | 23.81 | 0.00 | 100.00 | 21.43 | 34.12 | 0.00 | 5.00 | 54.84 | 26.87 | 30.49 | 29.60 |
| 0.20 | 0.10 | 24.24 | 29.03 | 50.00 | 34.29 | 27.85 | 28.57 | 0.00 | 100.00 | 27.27 | 35.58 | 0.00 | 0.00 | 55.17 | 28.81 | 29.44 | 27.98 |
| 0.20 | 0.20 | 22.95 | 32.18 | 66.67 | 30.77 | 21.74 | 30.40 | 0.00 | 100.00 | 13.33 | 38.18 | 0.00 | 0.00 | 62.22 | 33.33 | 23.60 | 24.08 |
| 0.20 | 0.50 | 13.33 | 27.03 | 0.00 | 22.22 | 20.69 | 16.84 | 0.00 | 0.00 | 15.38 | 41.38 | 0.00 | 0.00 | 51.61 | 24.56 | 17.65 | 20.86 |
| 0.50 | 0.01 | 21.43 | 27.74 | 50.00 | 46.15 | 26.67 | 21.10 | 0.00 | 66.67 | 19.35 | 34.12 | 0.00 | 4.35 | 53.97 | 26.09 | 31.49 | 29.60 |
| 0.50 | 0.05 | 22.56 | 27.94 | 50.00 | 42.11 | 25.29 | 23.81 | 0.00 | 100.00 | 22.22 | 34.12 | 0.00 | 5.00 | 54.84 | 27.07 | 30.49 | 29.60 |
| 0.50 | 0.10 | 25.74 | 29.27 | 50.00 | 33.33 | 27.85 | 28.93 | 0.00 | 100.00 | 19.05 | 35.58 | 0.00 | 0.00 | 55.17 | 28.33 | 30.61 | 28.69 |
| 0.50 | 0.20 | 23.33 | 32.18 | 66.67 | 30.77 | 21.74 | 29.03 | 0.00 | 100.00 | 13.33 | 38.53 | 0.00 | 0.00 | 62.22 | 31.17 | 24.84 | 24.21 |
| 0.50 | 0.50 | 9.09 | 27.03 | 0.00 | 23.53 | 20.69 | 17.02 | 0.00 | 0.00 | 15.38 | 41.38 | 0.00 | 0.00 | 51.61 | 24.56 | 17.65 | 20.86 |
| 1.00 | 0.01 | 21.43 | 27.74 | 50.00 | 46.15 | 26.67 | 21.10 | 0.00 | 66.67 | 19.35 | 34.12 | 0.00 | 4.35 | 53.97 | 26.09 | 31.49 | 29.60 |
| 1.00 | 0.05 | 22.56 | 27.94 | 50.00 | 42.11 | 25.29 | 23.92 | 0.00 | 100.00 | 22.22 | 34.12 | 0.00 | 5.00 | 54.84 | 27.07 | 30.49 | 29.60 |
| 1.00 | 0.10 | 26.00 | 29.27 | 50.00 | 33.33 | 27.85 | 28.75 | 0.00 | 100.00 | 19.05 | 35.58 | 0.00 | 0.00 | 54.24 | 28.33 | 30.61 | 28.69 |
| 1.00 | 0.20 | 23.33 | 32.56 | 66.67 | 30.77 | 21.74 | 29.03 | 0.00 | 100.00 | 13.33 | 38.53 | 0.00 | 0.00 | 62.22 | 31.17 | 24.69 | 24.34 |
| 1.00 | 0.50 | 9.09 | 27.03 | 0.00 | 23.53 | 20.69 | 17.02 | 0.00 | 0.00 | 15.38 | 41.38 | 0.00 | 0.00 | 51.61 | 24.56 | 17.65 | 20.86 |

## K  FINDING THE BEST IN-CONTEXT EXAMPLES FOR ALL DOCUMENTS

We performed an ablation study to find the global top-$K$ in-context examples for all dev documents, where $K = 5$ as in our default REPLM configuration. That is, we search a *fixed* set of $K$ documents that would yield the best overall performance for a given relation. For this, we leverage the parallel feature selection via group testing (Zhou et al., 2014), where we treat each document in the distantly-supervised set as a feature from the original method and then search for top-$K$ documents.

In essence, this method requires running many experiments, each of which evaluates a random subset of documents, where the number of experiments grows quadratically with respect to the number of documents available. To reduce the number of experiments to a computationally feasible level, we performed this ablation study for only one relation (P118), and we performed our search within the 100 documents that have the highest average cosine similarity to the training documents. This results in 2,659 experiments with different set of in-context examples evaluated on training set.

Table 51 shows the performance of top-$K$ documents selected via group testing (named as REPLM (group testing)). For a better comparison, we include the performance of all methods implemented. For methods with random in-context examples, the performance may be subject to variability across which seed is picked (whereas the performance is deterministic for the other methods), and, hence, we report the standard deviation for this subset of the methods by averaging the performance across 10 runs.

It is shown that the selected documents perform better than the random document selection, e.g., compared to REPLM (random fixed) and REPLM (random all). However, it performs worse than retrieving the most relevant in-context examples for each document, e.g., compared to REPLM (best context⊖) and REPLM (best context⊕). Therefore, it justifies our original REPLM framework in retrieving the semantically most relevant documents for each dev document.

Table 51: Performance of finding the best in-context examples for all documents via group-testing theory (for relation P118). Shown are F1 scores for each method.

| Method | F1 |
| --- | --- |
| REBEL | 34.21 |
| REBEL-sent | 40.74 |
| REPLM (random fixed) | 32.16 ± 5.80 |
| REPLM (random all) | 33.51 ± 5.47 |
| REPLM (best context⊖) | 44.04 |
| REPLM (best context⊕) | 37.84 |
| REPLM | 39.75 |
| REPLM (params adj) | 46.28 |
| REPLM (group testing) | 36.02 |

## L    PERFORMANCE WITH VARYING NUMBER OF IN-CONTEXT EXAMPLES

We performed an ablation study to show how the performance of REPLM framework changes with different number of in-context examples (i. e., varying $K$). For this, we run the same experiment on CONLL04 dataset with $K = 3, 5, 7, 9, 11$ while keeping all the other parameters fixed. Table 52 and Table 53 show the performance of REPLM and REPLM+GPT3.5, respectively, for each relation type and the overall performance. We observe that more in-context examples yield better F1 scores in general. Informed by this observation, we used the highest number of in-context examples ($K$) in our experiments. For the document-level relation extraction dataset DocRED, we use $K = 5$ as more than 5 documents do not fit into the context window of our REPLM framework.

Table 52: Comparing the performance of REPLM across varying number of in-context examples $K$ for the dataset CONLL04. Shown are F1 scores on each relation and overall (Micro) F1 score.

| Num. examples ($K$) | Kill | Live_In | Located_In | OrgBased_In | Work_For | Overall |
|---|---|---|---|---|---|---|
| 3 | 88.89 | 64.04 | 57.52 | 67.05 | 72.85 | 68.44 |
| 5 | 90.72 | 63.80 | 60.36 | 68.57 | 78.43 | 70.54 |
| 7 | **93.75** | 64.37 | 56.05 | 71.66 | 78.43 | 70.93 |
| 9 | 90.72 | 62.96 | 57.69 | 71.51 | 77.03 | 70.35 |
| 11 | 91.84 | **64.80** | **62.96** | **73.45** | **80.54** | **72.94** |

Table 53: Comparing the performance of REPLM+GPT3.5 across varying number of in-context examples $K$ for the dataset CONLL04. Shown are F1 scores on each relation and overall (Micro) F1 score.

| Num. examples ($K$) | Kill | Live_In | Located_In | OrgBased_In | Work_For | Overall |
|---|---|---|---|---|---|---|
| 3 | 91.63 | 74.45 | 71.08 | 82.41 | 77.01 | 77.94 |
| 5 | 93.85 | 75.58 | 69.81 | 81.29 | 79.56 | 78.60 |
| 7 | 91.93 | 75.78 | 73.00 | 80.52 | 79.85 | 78.81 |
| 9 | **94.92** | **76.36** | 72.45 | 81.11 | 80.12 | 79.49 |
| 11 | 93.75 | 76.06 | **73.32** | **84.31** | **80.98** | **80.19** |

# M    IS REPLM ACTUALLY LEARNING TO EXTRACT RELATIONS?

We test whether our REPLM framework is actually learning to extract the relation from the document or simply retrieving the facts from its own memory in the specified format. For this, we re-construct the sentences from CONLL04 (Roth & Yih, 2004) with fake entity names that are not mentioned anywhere in the web. As a result, we push the limits of our REPLM and test whether the model correctly extracts the relations about the entities that appear only in the input sentence. To the best of our knowledge, we are the first to design such an experiment to shed light on the learning abilities of a LM in the context of relation extraction.

Table 54 compares the performance of our REPLM based on the original dataset vs. the same dataset with fake entities. There is only a slight decrease in the overall performance with the fake entities (F1 score of 70.47 vs. 72.94), confirming that our REPLM is actually learning to extract the relations from the context. Further, Table 55 shows that our findings from REPLM are transferable to REPLM+GPT3.5, i.e., to more advanced LMs. After this confirmation, we further investigate "how" our REPLM learns to extract relations, by contrasting its behavior against the presence of adversarial in-context examples (see Appendix N).

We conjecture that the slight decrease in the performance can be attributed to mainly two factors: (1) In some cases, the relation between the entities becomes unclear without knowing what these entities actually are. As can be seen from the in-context examples at Sec. M.1, it is hard to infer "work for" relation between "Entity62" and "Entity22" from " ... , said Lt. Entity62 of the Entity22", which mitigates the model performance. (2) The random entity names (e. g., "Entity10", "Entity55" etc.) are not seen during the pre-training of the model. Therefore, the likelihood of generating the fake entities from the context is not the same as generating the real names that appeared during the pre-training, which can impact the performance of a LM.

Table 54: Comparing the performance of REPLM on the original dataset vs. the dataset with random entity names. Shown are F1 scores on each relation and overall (Micro) F1 score.

| Dataset | Kill | Live_In | Located_In | OrgBased_In | Work_For | Overall |
|---|---|---|---|---|---|---|
| Original | **91.84** | 64.80 | **62.96** | 73.45 | **80.54** | **72.94** |
| Random Entities | 69.66 | **76.68** | 51.53 | **75.96** | 75.82 | 70.47 |

Table 55: Comparing the performance of REPLM+GPT3.5 on the original dataset vs. the dataset with random entity names. Shown are F1 scores on each relation and overall (Micro) F1 score.

| Dataset | Kill | Live_In | Located_In | OrgBased_In | Work_For | Overall |
|---|---|---|---|---|---|---|
| Original | **93.75** | **76.06** | **72.32** | 84.31 | 80.98 | **80.19** |
| Random Entities | 88.89 | 74.65 | 68.93 | **85.59** | **85.71** | 79.65 |

In the following, we provide an example prompt and the generation of the output based on the re-constructed CONLL04 with the fake entity names.

## M.1    EXAMPLE PROMPT AND GENERATION FOR THE RELATION WORK_FOR

**Input prompt:**

Your task is to identify all the unique knowledge triplets of 'Work for' for a given context. Knowledge triplet will be ordered as relation, subject, and object, which are separated by <==>. If there are multiple triplets, list each of them in a new line. Follow the example context-relation pairs for the formatting of your output.

Context: Entity93 , an Entity79 employee , saw the plane go down as he was getting off work at the airport .

Relation: (Work for <==> Entity93 <==> Entity79)

Context: Entity39 , chief of waste disposal for the Entity97 , said the shafts would be ' ' the first major holes to be dug into the Entity93 formation. ' '

Relation: (Work for <==> Entity39 <==> Entity97)

...

Context: The fire at the Entity58 Entity97 's ammunition dump , which resulted from today 's explosion , is being localized , according to the fleet 's senior duty officer Entity55 .

Relation: (Work for <==> Entity55 <==> Entity58)

Context: Entity16 president Entity74 said he did nt know how any criminal wrongdoing could be found in the accident , which developed out of unanticipated conditions , not out of any willful acts.

Relation: (Work for <==> Entity74 <==> Entity16)

Context: The explosion set fire to about Entity55 of wooded area , said Lt. Entity62 of the Entity22 .

Relation: (Work for <==> Entity62 <==> Entity22)

Context: Entity55 's Entity10 said he believes the crater was created Entity41 ago when a 50-mile-wide meteorite slammed into the Entity95 .

**Output:**

Relation: (Work for <==> Entity10 <==> Entity55)

# N   LEARNING BEHAVIOR OF REPLM WITH ADVERSARIAL IN-CONTEXT EXAMPLES

To further investigate how our REPLM learns from the in-context examples, we evaluate the effect of adversarial in-context examples. To this end, we created the adversarial examples with the following strategy: (1) we randomly selected the half of the documents from the distantly-supervised set, and (2) for the selected documents, we replaced the subject-object pairs of the existing knowledge triplets with random named entities of the same document, which do not share the corresponding relation. We refer to this setup as "50 % clean + 50 % adversarial". Hence, half of the in-context examples should be adversarial for each dev document. As we do not change the content of the documents, this results in retrieving the same set of in-context examples as in our original framework (i. e., the similarity scores remain the same).

For comparison, we include two other experimental setups: (1) "100 % clean" refers to our original work and (2) "50 % clean" refers to experiments with remaining distantly-supervised documents after discarding the adversarial ones. Of note, we perform all the experiments in this section with our REPLM (best context⊖) framework variation. This enables us to directly quantify the impact of the adversarial in-context examples, without any aggregation over the multiple sets of in-context examples.

Table 56 shows the overall results. Furthermore, Tables 57 to 62 reports the performance across each relation type. The performance clearly drops as a result of including the adversarial in-context examples. This has two crucial implications: (1) The quality of the labels is important. Overall, we perform our experiments with distantly-supervised documents that are automatically annotated with certain heuristics to ensure quality (as explained in the main paper). If these are ignored, the resulting in-context examples can mislead the LM. (2) Our framework *actually* learns from the in-context examples. When it learns the relations with noisy labels, this also reflects into its performance. Further, comparing the performance between "100 % clean" and "50 % clean", we see that the relation extractions stays at the same level. This informs us that, if some adversarial documents exist in the dataset, filtering them out is a viable option and there is no need to find another method to fix their labels.

Table 56: Document-level relation extraction performance with adversarial in-context examples. Shown: Micro F1 scores.

| Data Source | F1-Score |
|---|---|
| 100 % clean | 31.31 |
| 50 % clean | 31.16 |
| 50 % clean + 50 % adversarial | 18.33 |

Table 57: Performance with adversarial in-context examples. Shown are F1 scores on each relation. (Part 1 / 6)

| Data Source | P6 | P17 | P19 | P20 | P22 | P25 | P26 | P27 | P30 | P31 | P35 | P36 | P37 | P39 | P40 | P50 |
|---|---|---|---|---|---|---|---|---|---|---|---|---|---|---|---|---|
| 100 % clean | 35.96 | 24.60 | 71.38 | 50.39 | 18.18 | 15.38 | 33.33 | 29.14 | 31.25 | 6.59 | 31.46 | 47.06 | 31.17 | 25.00 | 20.38 | 38.60 |
| 50 % clean | 40.91 | 24.41 | 72.05 | 43.17 | 18.02 | 20.69 | 31.94 | 28.61 | 39.15 | 9.20 | 36.36 | 50.98 | 28.21 | 25.00 | 18.67 | 40.24 |
| 50 % clean + 50 % adversarial | 20.93 | 11.28 | 44.91 | 26.55 | 8.85 | 7.41 | 21.62 | 16.12 | 17.73 | 15.73 | 9.30 | 20.00 | 12.66 | 14.29 | 11.25 | 20.73 |

Table 58: Performance with adversarial in-context examples. Shown are F1 scores on each relation. (Part 2 / 6)

| Data Source | P54 | P57 | P58 | P69 | P86 | P102 | P108 | P112 | P118 | P123 | P127 | P131 | P136 | P137 | P140 | P150 |
|---|---|---|---|---|---|---|---|---|---|---|---|---|---|---|---|---|
| 100 % clean | 48.30 | 47.62 | 34.38 | 57.47 | 23.26 | 44.44 | 30.19 | 32.65 | 44.04 | 30.19 | 18.49 | 25.50 | 20.00 | 12.90 | 14.17 | 31.28 |
| 50 % clean | 50.77 | 47.06 | 33.33 | 56.98 | 21.43 | 41.18 | 31.25 | 34.04 | 42.99 | 32.38 | 17.09 | 23.94 | 22.22 | 16.67 | 9.52 | 32.14 |
| 50 % clean + 50 % adversarial | 35.34 | 25.37 | 12.50 | 23.86 | 11.24 | 31.14 | 18.35 | 17.02 | 17.39 | 26.55 | 17.24 | 16.12 | 23.08 | 13.33 | 7.52 | 13.15 |

Table 59: Performance with adversarial in-context examples. Shown are F1 scores on each relation. (Part 3 / 6)

| Data Source | P155 | P156 | P159 | P161 | P162 | P166 | P170 | P171 | P172 | P175 | P176 | P178 | P179 | P190 | P194 | P205 |
|---|---|---|---|---|---|---|---|---|---|---|---|---|---|---|---|---|
| 100 % clean | 23.33 | 21.51 | 40.85 | 33.85 | 14.12 | 26.92 | 10.00 | 10.53 | 23.73 | 40.71 | 24.14 | 28.57 | 22.45 | 100.00 | 19.57 | 13.56 |
| 50 % clean | 17.24 | 27.08 | 37.76 | 31.89 | 16.47 | 30.77 | 10.67 | 10.00 | 29.51 | 41.77 | 25.45 | 30.88 | 26.26 | 66.67 | 15.22 | 17.86 |
| 50 % clean + 50 % adversarial | 6.96 | 16.16 | 27.78 | 19.20 | 16.28 | 20.18 | 14.08 | 17.39 | 6.78 | 29.57 | 14.81 | 28.79 | 12.37 | 50.00 | 6.59 | 3.77 |

Table 60: Performance with adversarial in-context examples. Shown are F1 scores on each relation. (Part 4 / 6)

| Data Source | P206 | P241 | P264 | P272 | P276 | P279 | P355 | P361 | P364 | P400 | P403 | P449 | P463 | P488 | P495 | P527 |
|---|---|---|---|---|---|---|---|---|---|---|---|---|---|---|---|---|
| 100 % clean | 14.63 | 43.24 | 30.77 | 35.09 | 20.00 | 11.11 | 30.77 | 27.12 | 52.00 | 40.00 | 18.52 | 33.33 | 38.61 | 18.18 | 22.71 | 23.31 |
| 50 % clean | 14.52 | 49.32 | 34.45 | 35.71 | 26.36 | 10.91 | 29.63 | 24.85 | 34.78 | 41.90 | 25.45 | 35.62 | 34.20 | 17.65 | 22.06 | 20.19 |
| 50 % clean + 50 % adversarial | 8.13 | 25.97 | 29.03 | 21.82 | 9.68 | 0.00 | 3.92 | 15.30 | 8.16 | 36.70 | 18.18 | 22.54 | 20.51 | 12.12 | 10.13 | 9.88 |

Table 61: Performance with adversarial in-context examples. Shown are F1 scores on each relation. (Part 5 / 6)

| Data Source | P551 | P569 | P570 | P571 | P576 | P577 | P580 | P582 | P585 | P607 | P674 | P676 | P706 | P710 | P737 | P740 |
|---|---|---|---|---|---|---|---|---|---|---|---|---|---|---|---|---|
| 100 % clean | 36.36 | 60.95 | 46.92 | 38.96 | 14.49 | 46.78 | 27.40 | 31.11 | 25.64 | 26.09 | 26.42 | 61.54 | 16.33 | 27.96 | 0.00 | 38.46 |
| 50 % clean | 36.36 | 61.22 | 47.93 | 37.58 | 12.12 | 43.87 | 30.56 | 37.21 | 23.38 | 25.41 | 22.22 | 46.15 | 16.33 | 26.37 | 0.00 | 38.46 |
| 50 % clean + 50 % adversarial | 0.00 | 33.22 | 22.38 | 20.53 | 5.97 | 27.52 | 15.38 | 18.60 | 10.39 | 16.30 | 16.16 | 15.38 | 13.04 | 8.70 | 0.00 | 16.00 |

Table 62: Performance with adversarial in-context examples. Shown are F1 scores on each relation. (Part 6 / 6)

| Data Source | P749 | P800 | P807 | P840 | P937 | P1001 | P1056 | P1198 | P1336 | P1344 | P1365 | P1366 | P1376 | P1412 | P1441 | P3373 |
|---|---|---|---|---|---|---|---|---|---|---|---|---|---|---|---|---|
| 100 % clean | 25.35 | 31.46 | 66.67 | 46.15 | 21.28 | 22.06 | 0.00 | 100.00 | 31.58 | 45.54 | 0.00 | 0.00 | 52.38 | 42.35 | 20.00 | 20.11 |
| 50 % clean | 19.72 | 36.78 | 66.67 | 46.15 | 27.91 | 23.94 | 0.00 | 100.00 | 31.58 | 42.86 | 0.00 | 0.00 | 58.54 | 42.86 | 20.69 | 24.86 |
| 50 % clean + 50 % adversarial | 14.71 | 14.89 | 0.00 | 23.08 | 21.28 | 8.63 | 0.00 | 0.00 | 9.52 | 15.38 | 0.00 | 0.00 | 50.00 | 34.41 | 13.02 | 15.96 |

## O   DETAILED PERFORMANCE COMPARISON AGAINST OTHER IN-CONTEXT LEARNING METHODS

In this section, we compare the performance of our REPLM against the in-context learning methods developed for *sentence-level* relation extraction task (Wadhwa et al., 2023; Wan et al., 2023). As explained in Sec. 2, these models are not scalable to *document-level*. Further, at the time of writing, the implementation of (Wadhwa et al., 2023) is not publicly available, therefore, we compare our REPLM against GPT-RE (Wan et al., 2023).

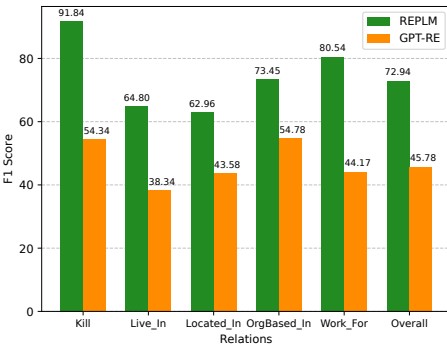

Figure 6: Comparing the performance of our REPLM against GPT-RE (Wan et al., 2023) on CONLL04 dataset.

Figure 6 shows the performance of our REPLM and GPT-RE (Wan et al., 2023) for each relation, along with the overall performance (Micro F1). Our REPLM consistently outperforms GPT-RE on each relation and it achieves roughly 60% F1 score improvement over GPT-RE (72.94 vs. 45.78). We attribute the inferior performance of GPT-RE to mainly two reasons: (1) GPT-RE introduces the in-context examples of all relations into the same context, therefore, the model is forced to learn the classification of all relation types at one inference. With more relation types, the task becomes more difficult. (2) In most cases, GPT-RE outputs one of the relation types, even when there is no relation between the given entity pairs. Although the in-context examples include "no relation" instances, it is not possible to cover all variations of "no relation" cases.

On top of the inferior performance of GPT-RE, we note that it is much more costly to run. (1) Our REPLM leverages a 7B-parameter model GPT-JT, which fits into a GPU with 32 GB memory. On the other hand, GPT-RE framework relies on commercial products, such as OpenAI's GPT models. (2) GPT-RE framework runs the inference for each entity pair in a sentence. As a result, when there are $N$ named entities in a sentence, GPT-RE runs the inference $\sim N^2$ for the same sentence. This results in costly computations. For instance, we spent roughly 100 USD to complete the experiments on CONLL04, which is the smallest sentence-level relation extraction dataset in our setup.

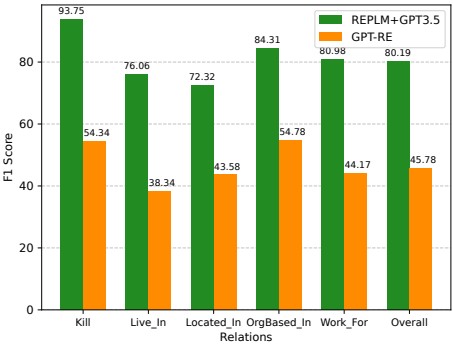

Figure 7: Comparing the performance of our REPLM+GPT3.5 against GPT-RE (Wan et al., 2023) on CONLL04 dataset.

To better demonstrate the strength of our framework, we further compared GPT-RE against our REPLM+GPT3.5, in which the backbone LM is replaced by GPT-3.5, so that we use the same GPT model that GPT-RE leverages. Figure 7 shows that our REPLM+GPT3.5 outperforms GPT-RE by even more larger margins. Specifically, it achieves more than 75% F1 score improvement over GPT-RE (80.19 vs. 45.78). Further, our REPLM+GPT3.5 requires much less API calls than GPT-RE, therefore, it costs around 15 USD in comparison to 100 USD spending required from GPT-RE.

