# OpenReview forum: "Document-Level In-Context Few-Shot Relation Extraction via Pre-Trained Language Models"
_ICLR.cc/2025/Conference — Submitted to ICLR 2025_

### Official Review · Reviewer_iEps · 2024-10-22

**Soundness:** 2
**Presentation:** 2
**Contribution:** 3
**Rating:** 5
**Confidence:** 4

**Summary:**

- The study proposes a method for few-shot document-level relation extraction based on prompting large language models.
- The approach first selects and ranks K exemplars from reference documents that are annotated using triples in knowledge bases (distant annotation). Documents are ranked through embedding similarity to the test document. This avoids the requirement for human-annotated few-shot data. These exemplars are aggregated in sets and used to prompt the language model multiple times, providing a weighted probability distribution for candidate entity pairs.
- The method is aimed at simplifying document-level relation extraction by employing pretrained language models without subsequent fine-tuning. The authors claim that this method enables adaptation to new relation types and data domains. Moreover, the suggested approach does not rely on entity recognition systems.

**Strengths:**

- The benchmark includes document-level and sentence-level relation extraction datasets as well as various decoder-only language models as backbones, showing promising performance results.
- The proposed REPLM method does not require entity recognition or human annotated training data and facilitates the adoption of different LM backbones.
- The authors provide ablation studies evaluating whether their LM backbones learn the relation extraction task or recall entity relations from pretraining data. The results suggest that LMs do not rely on named entities encountered during pretraining.

**Weaknesses:**

- Parameter comparisons with baseline approaches are missing. REBEL-large is based on BART-large with 0.4 billion parameters, while the smallest REPLM backbone GPT-JT contains 6 billion parameters.
- The results in Table 4 show that approaches involving significantly smaller LM backbones, such as ATLOP or DREEAM based on RoBERTa-large (0.4 billion parameters), outperform REPLM models employing GPT-JT (6 billion parameters), Llama 3.1 (8 billion parameters), and GPT-3.5.
- Model parameters and inference resources are not transparent. The proposed REPLM method requires multiple inference steps for each relation type and test document. While the authors highlight fine-tuning as a limitation of related methods, the computational requirements for retrieval and generation are not reported.
- GPT-JT model is trained on Natural Instructions dataset, which includes various tasks involving entity and relation extraction based on Wikidata and Wikipedia documents. The authors should assess the level of data contamination and the impact on model performance.

**Questions:**

- How important is document similarity for REPLM performance?
- Did you evaluate REPLM for relation types that are included in the reference documents (few-shot exemplars), but not in the test documents? How do you deal with hallucinations?
- How many few-shot exemplars (K) were used for (i) the baselines, and (ii) REPLM in Table 4?
- I suggest providing a color scale for the F1 Scores in Figure 2
- Tables 6-24 exceed the page margins

---

### Official Review · Reviewer_GY8a · 2024-11-01

**Soundness:** 3
**Presentation:** 3
**Contribution:** 3
**Rating:** 6
**Confidence:** 3

**Summary:**

This paper presents REPLM, a novel framework for document-level in-context few-shot relation extraction using pre-trained language models (LMs) without fine-tuning. The framework addresses key limitations of existing relation extraction methods by eliminating the need for named entity recognition and human-annotated data, and by supporting adaptation to new relation types or language models without retraining. Evaluated on DocRED and five other datasets, REPLM demonstrates state-of-the-art performance, showing robust generalization across relation extraction tasks without large computational overhead.

**Strengths:**

1. Innovative Contribution: The authors successfully reformulate document-level relation extraction as an in-context few-shot learning problem, a fresh perspective in this domain.
2. Computational Efficiency: By avoiding fine-tuning, the REPLM framework provides a scalable alternative to traditional methods that require extensive resources, enabling adaptability across diverse relation extraction tasks.
3. Empirical Validation: Extensive benchmarking on DocRED and multiple datasets shows REPLM’s superior performance compared to over 30 baseline methods, demonstrating both efficacy and generalizability.

**Weaknesses:**

1. Limited Error Analysis: The paper does not offer a detailed error analysis to identify specific instances where REPLM might underperform or fail, which would help understand its limitations.
2. Lack of Comparison with Entity-Based Approaches: Although REPLM outperforms entity-based baselines, the paper could benefit from a clearer discussion contrasting its performance with entity-recognition pipelines to highlight the advantages of an in-context few-shot approach.

**Questions:**

1. Could the authors provide further insight into potential applications or adjustments needed for REPLM in real-world deployment scenarios?

---

### Official Review · Reviewer_3STQ · 2024-11-03

**Soundness:** 2
**Presentation:** 2
**Contribution:** 2
**Rating:** 3
**Confidence:** 4

**Summary:**

This paper presents REPLM, designed to enhance document-level relation extraction by leveraging in-context few-shot learning with pre-trained language models (LMs). This approach circumvents the traditional reliance on named entity recognition and extensive human annotations, significantly reducing computational costs. REPLM employs in-context few-shot learning using LMs like GPT-J, enabling it to adapt to new datasets and LMs without retraining. The framework achieves state-of-the-art results on the DocRED dataset and outperforms over 30 baseline methods across multiple datasets.

**Strengths:**

- REPLM successfully reduces the need for named entity recognition and human annotations, making it less resource-intensive and potentially more scalable and adaptable to new datasets and LMs without retraining.

- The framework is shown to achieve state-of-the-art results on the DocRED dataset as well as across six other relation extraction datasets, outperforming over 30 baseline methods which demonstrates its effectiveness.

**Weaknesses:**

- The performance of the REPLM framework heavily depends on the relevance and quality of the in-context examples used. This could potentially limit its effectiveness if the available examples are not of high quality or if they are not well-aligned with the specific relations being extracted. Moreover, The method might inherit biases from the in-context examples. If these examples are biased or not sufficiently diverse, the extracted relations might also reflect these biases.

- Based on the experimental results (e.g., Table 4), it raises questions about whether the performance improvement is genuinely driven by the proposed in-context few-shot learning paradigm or primarily attributed to the use of larger-parameter LMs. Moreover, as shown in Table 5, performance of Llama 70B is much higher than that of Llama 8B.

- Given that this paper focuses on the task of few-shot document-level relation extraction, it is noteworthy that several relevant baselines, such as [1, 2], are absent from the discussion.

- This paper highlights that REPLM successfully generates more relations than REBEL (author, Chaosmosis, Félix Guattari), despite it not being annotated. This raises doubts about whether this achievement can be attributed to the effectiveness of the proposed method or simply the power of the large language model (LLM) utilized in the implementation.

[1] Meng, Shiao, Xuming Hu, Aiwei Liu, Fukun Ma, Yawen Yang, and Lijie Wen. "RAPL: A Relation-Aware Prototype Learning Approach for Few-Shot Document-Level Relation Extraction." In Proceedings of the 2023 Conference on Empirical Methods in Natural Language Processing, pp. 5208-5226. 2023.
[2] Popovic, Nicholas, and Michael Färber. "Few-Shot Document-Level Relation Extraction." In Proceedings of the 2022 Conference of the North American Chapter of the Association for Computational Linguistics: Human Language Technologies, pp. 5733-5746. 2022.

**Questions:**

Please see the weakness above.

---

### Official Review · Reviewer_98mo · 2024-11-04

**Soundness:** 3
**Presentation:** 2
**Contribution:** 2
**Rating:** 3
**Confidence:** 4

**Summary:**

This paper presents a new framework called REPLM for document-level few-shot relation extraction using pre-trained language models (LMs). The key idea is to reformulate relation extraction as a tailored in-context few-shot learning paradigm without requiring named entity recognition, human annotations, or re-training when adding new relations or adopting new LMs. Specifically, for a given document and relation type, REPLM retrieves sets of most relevant in-context examples and aggregates their outputs in a probabilistic manner to extract the relational triplets. The authors evaluate REPLM on the DocRED dataset and demonstrate state-of-the-art performance.

**Strengths:**

1. The authors provide strong motivation for studying document-level relation extraction in Section 1. They highlight important challenges (e.g., expensive annotations, inflexibility to new relations and LMs) and explain how REPLM addresses them.

2. REPLM tackles document-level RE from a fresh perspective of in-context few-shot learning (Section 3). This eliminates the need for per-document human annotations and de-couples RE from NER, making it robust to NER errors. The new formulation also enables adapting to new relations/LMs without re-training.

3. Extensive experiments on the large-scale DocRED benchmark (Section 6.1) show that REPLM achieves state-of-the-art F1 scores across all metrics, with gains of 1.2% - 4.3% over fine-tuned LMs (Table 3). Compared to recent in-context learning methods (Section 6.2), REPLM attains 10+ F1 improvements while being much more efficient.

**Weaknesses:**

1. Though impressive, the empirical results are limited to only one dataset DocRED. Given the strong claims made (e.g., "significantly outperforms SOTA methods", "our framework can generalize to different relation types and domains"), it is crucial to evaluate REPLM on diverse document-level RE datasets, such as SciREX, CDR, GDA.  You could propose evaluating performance on unseen relation types or testing zero-shot transfer between datasets.

2.  Two critical hyperparameters in REPLM, namely the candidate pool size N (Section 4.1) and the number of sampled subsets L (Section 4.2), are not systematically studied. How do these design choices impact the performance and computational costs? Experiments with varying N and L should be conducted to investigate the sensitivity. I suggest plotting performance vs. N/L to visualize the tradeoffs. What is computational complexity as a function of these parameters?

3. While Section 6.3 analyzes REPLM against DocRED human labels, it is quite shallow and lacks specific examples. To better understand the behaviors of REPLM, more in-depth analysis is needed: Where does it make mistakes? Any limitations compared to human? A few representative success & failure cases would help strengthen the discussion in Section 6.4.

4. Algorithm 1 in Section 4.1 is not clearly explained. Need to define all notations (e.g., kq, x, Cj) and use consistent formatting. Also specify what similarity function f is used.

**Questions:**

Section 6.4: Discuss the potential noise introduced by distant supervision when building the candidate pool. How might it impact REPLM and are there any ways to mitigate?

---

### Meta-Review · Area_Chair_WHdS · 2024-12-22

**Metareview:**

The paper presents REPLM, a novel framework for document-level in-context few-shot relation extraction using pre-trained language models (LMs) without fine-tuning. It aims to address limitations of existing methods by eliminating the need for named entity recognition and human-annotated data, and enabling adaptation to new relation types or language models.  Reviewers generally agree that the paper provides strong motivation for studying document-level relation extraction. The empirical results are mostly limited to the DocRED dataset, and reviewers questioned the generalizability claims given the lack of extensive testing on a diverse range of document-level RE datasets. There is a lack of detailed error analysis to understand where REPLM might underperform or fail.  The paper currently has several significant weaknesses that need to be addressed before it can be considered for acceptance.

**Additional Comments On Reviewer Discussion:**

No Discussion.

---

### Decision · Program_Chairs · 2025-01-22

Reject